# CAUSAL TIME SERIES GENERATION VIA DIFFUSION MODELS

## ABSTRACT

Time series generation (TSG) synthesizes realistic sequences and has achieved remarkable success. Among TSG, conditional models generate sequences given observed covariates, however, such models learn observational correlations without considering unobserved confounding. In this work, we propose a causal perspective on conditional TSG and introduce *causal time series generation* as a new TSG task family, formalized within Pearl's causal ladder, extending beyond *observational* generation to include *interventional* and *counterfactual* settings. To instantiate these tasks, we develop `CaTSG`, a unified diffusion-based framework with backdoor-adjusted guidance that causally steers sampling toward desired interventions and individual counterfactuals while preserving observational fidelity. Specifically, our method derives causal score functions via backdoor adjustment and the abduction–action–prediction procedure, thus enabling principled support for all three levels of TSG. Extensive experiments on both synthetic and real-world datasets show that `CaTSG` achieves superior fidelity while also supporting interventional and counterfactual generation that existing baselines cannot handle. Overall, we propose the causal TSG family and instantiate it with `CaTSG`, providing an initial proof-of-concept and opening a promising direction toward more reliable simulation under interventions and counterfactual generation. Our source code is available at https://anonymous.4open.science/r/CaTSG_iclr26.

.

## 1 INTRODUCTION

Time Series Generation (TSG) aims to synthesize realistic sequences that preserve temporal dependencies and inter-dimensional correlations, and is crucial for many applications, e.g., data augmentation (Ramponi et al., 2018), privacy preservation (Yoon et al., 2020) and simulation for downstream tasks including classification and forecasting (Ang et al., 2023). A significant extension, conditional TSG, generates sequences under given conditions, enabling controlled synthesis (Yoon et al., 2019; Coletta et al., 2023; Narasimhan et al., 2024). In most implementations, conditional TSG aligns generated data with the observational conditional $P(X \mid C)$ using correlation-fitting objectives, e.g., adversarial training in conditional Generative Adversarial Networks(GANs) (Yoon et al., 2019) and conditional diffusion with denoising score-matching objectives (Narasimhan et al., 2024). However, these approaches remain fundamentally *correlation-driven*: they learn statistical associations between conditions and outcomes without modeling the underlying data-generating mechanisms.

However, real-world time series systems are rarely governed by observed variables alone. Consider a transportation example: a dataset contains observational pairs $(X, C)$, where $X$ is school-neighborhood traffic volume and $C$ is temperature (Figure 1a). In reality, unobserved variables $E$ usually simultaneously affect both $X$ and $C$ (Figure 1b). For instance, summer holidays can be one of the $E$, which often coincide with higher temperatures while also reducing commuting demand, thereby acting as a latent common cause. Given such dataset, a typical conditional TSG model $p_\theta(X \mid C)$ (Figure 1c) cannot account for $E$ and may thus unknowingly capture *spurious correlations*, e.g., learning that "higher temperature causes lower traffic". This limits their applicability in real-world decision-making, where the goal may not be merely to replicate what happened, but to answer counterfactual queries such as: *"What would happen if we changed the condition within the same environment (i.e., with the latent factors held fixed)?"*

Moving beyond purely observational modeling, we adopt a *causal view* that makes the mechanisms linking conditions and outcomes explicit. Pearl's causal ladder (Pearl, 2009) offers a precise vocabulary for this extension, which organizes causal tasks into three levels: association, intervention, and counterfactual. Within this framing, classical conditional TSG seeks to approximate the observational conditional $P(X \mid C)$, and we refer to this setting as observational TSG (Obs-TSG), which occupies the association level. We then formulate *causal time series generation* as an extension with two higher-level tasks: interventional (Int-TSG) and counterfactual TSG (CF-TSG), shown in Figure 1d. Specifically, Int-TSG generates samples from $P(X \mid \mathrm{do}(C))$, where $\mathrm{do}(\cdot)$ denotes an intervention oper-

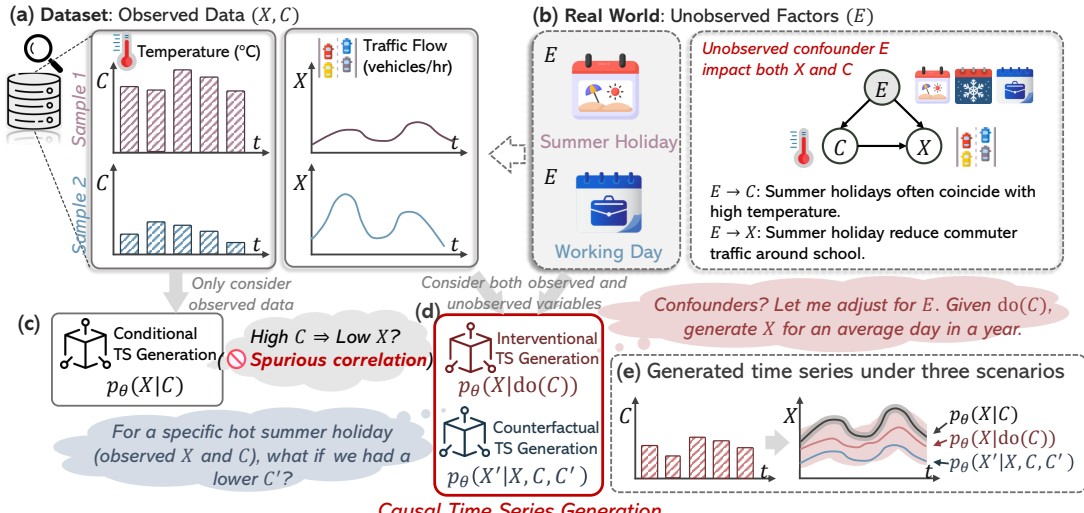

Figure 1: Motivation for causal time series generation. (a) We observe time series $X$ and $C$, (b) yet unobserved factors $E$ may affect both. (c) Standard conditional TSG risk learning spurious correlations. (d) We extend this paradigm into a causal TSG family with Int-TSG and CF-TSG. (e) Three tasks yield distinct outputs given same $C$.

ation that sets $C$ independent of its causes, thereby removing confounding effects from latent variables to simulate outcomes. CF-TSG generates samples from $P(X' \mid X, C, C')$, producing personalized alternative trajectories by replacing $C'$ for $C$ while holding the realized environment fixed. Conceptually, the outcomes under these tasks differ (Figure 1e): Obs-TSG (gray) may reflect spurious correlations with $C$; Int-TSG (red) shows adjusted outcomes free from latent-factor influences, yielding more natural and diverse generations; CF-TSG (blue) depicts an individual's specific alternative trajectory.

While introducing Int- and CF-TSG expands conditional TSG toward decision-oriented synthesis, it also raises several challenges: *ground truth dilemma*, *invisible confounders*, and *evaluation difficulty*. First, real-world time series often originate from human systems (e.g., mobility, health), where running interventions is ethically and practically infeasible (Xia et al., 2025b), thus causal supervision is rare and most datasets are observational (Matthay & Glymour, 2022). Second, confounders that simultaneously influence conditions and outcomes (e.g., individual difference and latent randomness) are typically unobserved and complex, making it non-trivial to model or adjust for them. Third, evaluating the quality of generated time series under these two tasks is difficult. Unlike images, time series lack reliable visual inspection, and their quality depends on preserving subtle statistical patterns (Narasimhan et al., 2024).

To address these challenges, motivated by the growing adoption of diffusion models in time series practice (Ho et al., 2020; Yang et al., 2024), we introduce a diffusion-based framework **CaTSG** (**Ca**usality-grounded **T**ime **S**eries **G**eneration). **CaTSG** employs backdoor-adjusted guidance to steer sampling toward interventional targets without ground-truth interventional labels and a learnable latent environment bank with an EnvInfer module to flexibly model latent confounders (*Challenges 1 and 2*). For *Challenge 3*, we complement real-world evaluations with SCM-grounded synthetic datasets that expose observable counterfactuals, enabling quantitative assessment of CF-TSG. Our contributions are summarized as follows:

- **Causal Expansion of Conditional TSG Paradigm.** We formalize *causal time series generation* as an extension of conditional TSG along Pearl's ladder, introducing two tasks beyond association, i.e., interventional (Int-TSG) and counterfactual (CF-TSG), to open up richer generative capabilities aligned with real-world decision-making needs.

- **A Unified Causality-Guided Diffusion Framework.** We derive *causal score functions* via backdoor adjustment and abduction–action–prediction, and instantiate **CaTSG** to embed these principles into diffusion sampling. Through backdoor-adjusted guidance and a learnable latent environment bank, **CaTSG** supports observational, interventional, and counterfactual generation within a single framework.

- **Comprehensive Empirical Evaluation.** Across four datasets, **CaTSG** consistently improves observational fidelity and achieves competitive interventional performance, outperforming the second-best baseline with relative gains of 2.4% - 98.7% across four metrics. For counterfactuals, **CaTSG** yields accurate CF generations on synthetic datasets, supported by environment analyses, and reasonable CF results on real-world datasets via case-study visualizations.

## 2 BACKGROUND

**Conditional Time Series Generation.** Conditional TSG aims to synthesize realistic time series that conform to specified auxiliary information. Early methods like conditional GANs (Esteban et al., 2017; Smith & Smith, 2020)

conditioned on discrete labels, but suffered from instability and mode collapse (Chen, 2021). Recent works adopt diffusion-based models (Sohl-Dickstein et al., 2015; Ho et al., 2020) with side information injected into the reverse process, enabling stable conditional generation (Tashiro et al., 2021). To handle heterogeneous metadata, TimeWeaver (Narasimhan et al., 2024) extends diffusion to categorical, continuous, and time-varying inputs. Other methods leverage structured state-space models (Alcaraz & Strodthoff, 2023), score-based transformers (Yuan & Qiao, 2024), or constrained objectives (Coletta et al., 2023). To improve adaptability across domains, recent efforts explore prompt-based control (Huang et al., 2025), disentanglement of dynamics and conditions (Bao et al., 2024), and textual guidance (Li et al., 2025). Consequently, diffusion models are currently a prevalent choice for time series generation (Yang et al., 2024; Narasimhan et al., 2024).

**Diffusion Models & Score Function.** Diffusion probabilistic models (Sohl-Dickstein et al., 2015; Ho et al., 2020; Song & Ermon, 2019) define a generative process by reversing a fixed forward noising process. Diffusion models can also be interpreted as score-based generative models (Song & Ermon, 2019), The score function estimates the gradient of the log-density of noisy samples: $s_t(x_t) = \nabla_{x_t} \log p(x_t)$. Under the Gaussian forward process, the data score satisfies $\nabla_{x_t} \log q(x_t \mid x_0) = -\frac{1}{\sigma_t} \epsilon$, $\sigma_t = \sqrt{1 - \bar{\alpha}_t}$. Accordingly, learning to predict the noise is equivalent to estimating the score up to a time-dependent scaling factor. In the conditional setting, the score becomes

$$s_t(x_t, c) = \nabla_{x_t} \log p(x_t \mid c) \approx -\frac{1}{\sigma_t} \varepsilon_\theta(x_t, t, c), \tag{1}$$

which links the denoising network directly to the conditional score function at each timestep. Note that due to the space limitation, we put preliminaries with more detail in Appendix B.

**Causal Modeling for Generation.** Causal views recently integrated into deep learning tasks e.g., representation learning (Yang et al., 2021), vision-language grounding (Wang et al., 2025), and anomaly detection (Xia et al., 2025c). In generative modeling (Komanduri et al., 2023), it has been explored across several modalities. In vision, models like CausalGAN (Kocaoglu et al., 2018), CausalDiffAE (Komanduri et al., 2024) and Counterfactual Generative Networks (Sauer & Geiger, 2021) enable interventions and recombinations of disentangled factors for counterfactual image synthesis. In addition, it has also been used for time-varying treatment effects (Wu et al., 2024), and decoupled classifier-free guidance further reduces spurious shifts during counterfactual sampling (Xia et al., 2025a). In text generation, structural causal models guide interventions on latent narrative factors (Hu & Li, 2021), while in knowledge graphs, they have been leveraged to generate hypothetical relations that enhance link prediction (Liu et al., 2021). Despite these advances, causal modeling for generation remains largely unexplored in time series.

## 3 A CAUSAL VIEW FOR CONDITIONAL TIME SERIES GENERATION

Traditional approaches to conditional TSG focus solely on observational correlations. To move beyond, we adopt a causal perspective and situate this generative task within Pearl's causal ladder (Pearl, 2009), thereby motivating a structured view across different causal levels.

**Causal Ladder for TSG.** Pearl's Causal Ladder organizes reasoning into three levels: association, intervention, and counterfactual. We adapt this hierarchy to the context of TSG, summarized in Table 1. *Level 1 (association)* captures statistical correlations between observed contextual variables $C$ (e.g., temperature) and the outcome $X$ (e.g., traffic); *Level 2 (intervention)* models the outcome under controlled changes to $C$ while adjusting for potential confounding from unobserved common causes; and *Level 3 (counterfactual)* simulates alternate outcomes $X'$ had $C$ been different, given the observed sample $X$ and $C$. This structured perspective enables us to unify and extend TSG tasks along the causal hierarchy.

**General Problem Statement.** Let $\mathcal{D} = \{(\mathbf{x}, \mathbf{c})\}_{i=1}^N$ denote a dataset consisting of $N$ samples of multivariate time series $\mathbf{x} \in \mathbb{R}^{T \times D}$ and paired contextual sequences $\mathbf{c} \in \mathbb{R}^{T \times D_c}$. Each $\mathbf{x} = [\boldsymbol{x}_1, \ldots, \boldsymbol{x}_T]$ represents a length-$T$ time series over $D$ dimensions, and $\mathbf{c}$ contains auxiliary features in the same time span with dimension $D_c$. **Our goal** is to learn a generative model $\mathcal{F}_\theta$ such that the samples generated from $\mathcal{F}_\theta(\mathbf{c})$ match a target distribution of the form $P(\mathbf{x}^\star \mid \mathcal{I})$, where the form of the conditioning variable $\mathcal{I}$ determines the underlying generation task. Specifically, we consider:

- **Observational TSG (Obs-TSG):** $P(\mathbf{x} \mid \mathbf{c})$, the standard conditional time series generation task;

- **Interventional TSG (Int-TSG):** $P(\mathbf{x} \mid \mathrm{do}(\mathbf{c}))$, which simulates outcomes under an intervention on $\mathbf{c}$, where the $\mathrm{do}(\cdot)$ operator sets $\mathbf{c}$ independent of its causes;

- **Counterfactual TSG (CF-TSG):** $P(\mathbf{x}' \mid \mathbf{x}, \mathbf{c}, \mathbf{c}')$, which simulates an alternative outcome under a counterfactual context $\mathbf{c}'$ given an observed realization $(\mathbf{x}, \mathbf{c})$.

This causal view elevates conditional TSG from a purely observational setting to a principled hierarchy of generative tasks. We refer to the latter two settings as *causal time series generation*, as they go beyond observational associations and involve reasoning over interventions and structured counterfactuals.

| Causal Level | Description | Example in Transportation[1] | Time Series Generation Task |
|---|---|---|---|
| *Level 1:* Association | **Seeing**: Observe statistical correlations | High temperature $\Rightarrow$ Low traffic flow (! May capture **spurious correlations**, because both may be caused by summer holiday) | **Obs-TSG**: $P(X \mid C)$ Generate time series $X$ under condition $C$ |
| *Level 2:* Intervention | **Doing**: Generate under controlled change of variables | Generate traffic flow when we set temperature = high, regardless of summer or not | **Int-TSG**: $P(X \mid \mathrm{do}(C))$ Generate time series $X$ given $C$, free from unobserved common causes |
| *Level 3:* Counterfactual | **Imagining**: Ask "what if" on a real sample | Observed traffic flow on a hot summer holiday $\Rightarrow$ what would the counterfactual traffic flow have been if the same holiday had been cool instead? | **CF-TSG**: $P(X' \mid X, C, C')$ Given observed $(X, C)$, generate what $X'$ would look like under $C'$ |

Table 1: Three levels of TSG following the causal ladder (Pearl, 2009). Color indicates semantic roles of variables: blue for generated outcomes $X$, red for observed condition $C$, green for counterfactual condition $C'$, yellow for counterfactual outcomes $X'$, and gray for latent or unobserved confounders (e.g., summer holiday).

## 4 METHODOLOGY

To instantiate $\mathcal{F}_\theta$ for the causal generative tasks, we first introduce a unified Structural Causal Model (SCM) (Pearl et al., 2000) as the theoretical foundation (Figure 2). We then derive the interventional and counterfactual objectives using two classical tools (Pearl et al., 2016): *backdoor adjustment* for Int-TSG and *abduction–action–prediction* procedure for CF-TSG. Given that diffusion models (Ho et al., 2020) have become a powerful technique for time series modeling with strong performance and stable training (Yang et al., 2024), we adopt diffusion as the backbone. Guided by the causal principles, within the diffusion framework, we derive a *causal score function* that guides the sampling process, and introduce **CaTSG** (Figure 3), a generative model that instantiates causal objectives in practice.

### 4.1 SCM & CAUSAL TREATMENTS

Beyond the no-confounding assumption of Obs-TSG (Figure 2a), we consider an SCM that captures the causation among observed sequences $X$, contextual variables $C$, and unobserved environment factors $E$ (Figure 2b), that supports the identification of interventional and counterfactual distributions.

**Backdoor Adjustment for Int-TSG** (Figure 2c). To estimate the interventional distribution $P(X \mid \mathrm{do}(C))$, we apply the *backdoor adjustment* (Pearl et al., 2016), which controls for confounding by conditioning on the appropriate set of variables that block all backdoor paths from $C$ to $X$, i.e., $C \leftarrow E \rightarrow X$. The interventional distribution can thus be approximated via (full derivation in Eq. 9):

$$P(X \mid \mathrm{do}(C)) = \sum_e P(X \mid C, E = e)\, P(E = e) \tag{2}$$

**Abduction–Action–Prediction (AAP) for CF-TSG** (Figure 2d). To estimate the counterfactual distribution $P(X' \mid X, C, C')$, which asks "what would the outcome $X'$ have been if $C$ had been $C'$ instead," we adopt the three-step procedure (Pearl et al., 2016):

- **Abduction:** Infer the posterior over latent environment $E$ from the observed sample $(X, C)$, i.e., estimate $P(E \mid X, C)$. This step captures the specific environment in which the observed data occurred.

- **Action:** Replace the observed $C$ with a counterfactual value $C'$ by simulating an intervention, i.e., modify the SCM by applying $\mathrm{do}(C = C')$.

- **Prediction:** Generate the counterfactual outcome $X'$ based on the modified SCM and the inferred $E$, yielding $P(X' \mid \mathrm{do}(C'), E)$.

For example, in the transportation scenario, suppose we observe low traffic flow $X$ on a hot day $C$. We first infer that the latent environment $E$ likely corresponds to a summer holiday based on our observation $X$ and $C$ (*abduction*); then replace $C$ with a counterfactual condition $C'$, such as 10°C (*action*); finally, we predict the counterfactual traffic $X'$ as if $C = C'$, while keeping $E$ (summer holiday) fixed (*prediction*). We provided more details on these causal concepts and derivations in Appendix B.1.

---

[1]Beyond the transportation example, we also include additional domains presented in Appendix H.2.

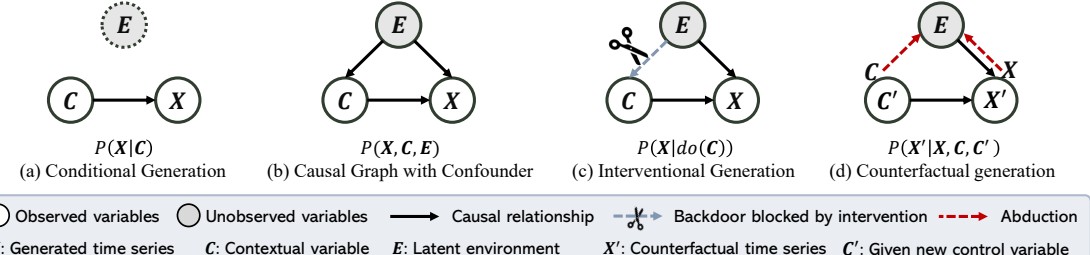

Figure 2: Structural Causal Models (SCMs) for different conditional time series generation paradigms.

## 4.2 Causal Score Functions

Building on these causal treatments, we next instantiate these formulations within a generative model. Diffusion models (Ho et al., 2020) have recently demonstrated strong performance in time series generation (Yang et al., 2024; Lin et al., 2024), where the denoising network learns to approximate the score function, i.e., the gradient of the log-density of noisy samples $s_t(x_t) = \nabla_{x_t} \log p(x_t)$. Extending this principle, we replace the standard conditional score $s_t(x_t, c) = \nabla_{x_t} \log p(x_t \mid c)$ in Eq. 1 with the following two causal counterparts:

$$s_t^{\text{int}}(x_t; \text{do}(c)) \triangleq \nabla_{x_t} \log p(x_t \mid \text{do}(c)) \quad \text{and} \quad s_t^{\text{cf}}(x_t'; x, c, c') \triangleq \nabla_{x_t'} \log p(x_t' \mid x, c, c'), \tag{3}$$

where $s_t^{\text{int}}$ denotes the interventional score function associated with the distribution $p(x_t \mid \text{do}(c))$, and $s_t^{\text{cf}}$ denotes the counterfactual score function associated with the distribution $p(x_t' \mid x, c, c')$. Under our SCM (Figure 2b), these score functions can be derived using backdoor adjustment (Eq. 2) and AAP:

$$s_t^{\text{int}} = \nabla_{x_t} \log p(x_t \mid \text{do}(c)) \propto (1+\omega) \, \mathbb{E}_{e \sim p(e|x,c)} \left[ \varepsilon_\theta(x_t, t, c, e) \right] - \omega \, \varepsilon_\theta(x_t, t) \tag{4}$$

$$s_t^{\text{cf}} = \nabla_{x_t} \log p(x_t' \mid x_t, c, c') \propto (1+\omega) \, \mathbb{E}_{e \sim p(e|x,c)} \left[ \varepsilon_\theta(x_t', t, c', e) \right] - \omega \, \varepsilon_\theta(x_t', t) \tag{5}$$

where $\varepsilon_\theta(\cdot)$ is the denoising network parameterized by $\theta$, and $\omega \geq 0$ is the guidance scale that balances conditional and unconditional terms. Detailed derivations are provided in Appendix C.2 due to the space limitation.

**Practical Discretized Implementation**. In practice, we discretize the continuous latent environment $e$ into a finite set of embeddings $\{\mathbf{e}_1, \ldots, \mathbf{e}_K\}$ with $\mathbf{e}_k \in \mathbb{R}^H$, and estimate their posterior weights as $\{w_1, \ldots, w_K\}$ with $w_k \in \mathbb{R}$. Denoting the environment-conditioned noise prediction $\varepsilon_\theta(\mathbf{x}_t, t, \mathbf{c}, \mathbf{e}_k)$ and unconditional noise predictions $\varepsilon_\theta(\mathbf{x}_t, t)$ from the denoising network $\varepsilon_\theta(\cdot)$ as $\boldsymbol{\epsilon}^{\text{env}}$ and $\boldsymbol{\epsilon}^{\text{base}}$, it leads to the following *unified approximation* of both interventional and counterfactual score functions:

$$s_t^{\text{int}} \text{ or } s_t^{\text{cf}} \approx (1+\omega) \sum_{k=1}^{K} w_k \, \boldsymbol{\epsilon}^{\text{env}} - \omega \, \boldsymbol{\epsilon}^{\text{base}}, \tag{6}$$

This causal function retains the same role as the standard diffusion score: during training, they act as targets for the denoising network, which learns them by regressing the injected Gaussian noise; during inference, they act as guidance signals that drive the reverse diffusion process.

## 4.3 A Unified Causality-Guided Diffusion Framework

A general conditional generative model $\mathcal{F}_\theta$ can be instantiated with different objectives depending on the target distribution of the generated samples (Figure 3a). Traditional Obs-TSG models $\mathcal{F}_\theta^{\mathcal{O}}$ generate samples from the observational distribution $P(X|C)$. To enable causal generation, one requires a causal generative model $\mathcal{F}_\theta^{\mathcal{C}}$. Building on the causal score functions (Eq. 6), we propose an instantiation of $\mathcal{F}_\theta^{\mathcal{C}}$, **CaTSG**, a unified diffusion-based generative framework for **Ca**usal **T**ime **S**eries **G**eneration, which supports both interventional and counterfactual settings. Importantly, when the environment component is omitted, **CaTSG** reduces to the standard Obs-TSG model. Thus, **CaTSG** serves as a general $\mathcal{F}_\theta$ that unifies all three TSG tasks within a single framework.

**Framework Overview** (Figure 3). **CaTSG** integrates three core components: (1) an environment inference module (**EnvInfer** $q_\phi$) that outputs latent representations $\mathbf{h}$, logits $\mathbf{s}$, and posterior environment weights $\mathbf{w}$ (the first two outputs used only for $\mathcal{L}_{\text{sw}}$), (2) a learnable environment bank $\mathbf{E}$ that provides representative embeddings of diverse environments, regularized by $\mathcal{L}_{\text{orth}}$, and (3) a denoising network (**Denoiser** $\varepsilon_\theta$) trained under a backdoor-adjusted formulation to regress Gaussian noise. Together, these modules instantiate the causal score function: EnvInfer assigns inputs $(\mathbf{x}, \mathbf{c})$ to anchor embeddings in $\mathbf{E}$, and Denoiser predicts noise conditioned on $\mathbf{c}$ and $\mathbf{E}$, as well as unconditional predictions to follow a Classifier-Free Guidance (CFG)-style scheme (Ho & Salimans, 2022). We next detail the training and generation procedure, while the architectural specifications of EnvInfer and the Denoiser are provided in Appendix C.4.

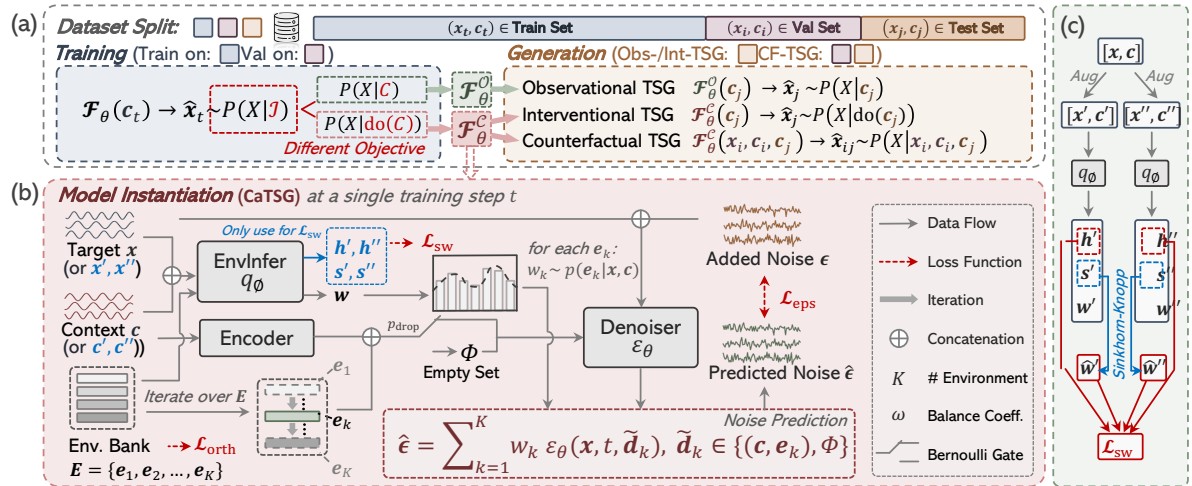

Figure 3: Overview of **CaTSG**. (a) Generative objectives for observational, interventional, and counterfactual TSG. (b) Model instantiation: **CaTSG** consists of EnvInfer, Environment Bank, and Denoiser. Blue variables are used only for $\mathcal{L}_{\text{sw}}$, Env. Bank is regularized by $\mathcal{L}_{\text{orth}}$, and Denoiser is regularized by $\mathcal{L}_{\text{eps}}$. (c) Swapped prediction loss.

**Training Objective.** We jointly optimize three components: Denoiser $\varepsilon_\theta$, EnvInfer $q_\phi$, and the learnable environment bank $E$. For Denoiser, to align with the causal score decomposition in Eq. 6, we use CFG-style training: with probability $p_{\text{drop}}$, we replace $(\mathbf{c}, \mathbf{e}_k)$ by $\varnothing$, otherwise we condition on $(\mathbf{c}, \mathbf{e}_k)$. Let $\boldsymbol{\epsilon} \sim \mathcal{N}(0, \mathbf{I})$ be the forward-process noise and $w_k$ the posterior environment weights from EnvInfer. The training objective is

$$\mathcal{L}_{\text{eps}} = \mathbb{E}_{\mathbf{x}_0, t, \boldsymbol{\epsilon}, \tilde{\mathbf{d}}}\left[\left\|\boldsymbol{\epsilon} - \sum_{k=1}^{K} w_k \varepsilon_\theta(\mathbf{x}_t, t, \tilde{\mathbf{d}}_k)\right\|_2^2\right], \quad \tilde{\mathbf{d}}_k \in \{(\mathbf{c}, \mathbf{e}_k), \varnothing\}. \tag{7}$$

For EnvInfer $q_\phi$, we expect it to autonomously discover latent environments from the input and produce a soft assignment over $K$ environment embeddings without labels. We thus adopt a SwAV-style swapped prediction loss (Caron et al., 2020), where each augmentation predicts the other's assignment, encouraging augmentations of the same sample to agree on the same environment assignment. Specifically, as illustrated in Figure 3c, for each sample $(\mathbf{x}, \mathbf{c})$, we create augmentations $\{(\mathbf{x}', \mathbf{c}'), (\mathbf{x}'', \mathbf{c}'')\}$. Via EnvInfer, we obtain latent representation $\mathbf{h}', \mathbf{h}'' \in \mathbb{R}^{N \times H}$, logits $\mathbf{s}, \mathbf{s}'' \in \mathbb{R}^K$ and the assignment weights $\mathbf{w}', \mathbf{w}'' \in \mathbb{R}^K$. The loss enforces $\mathbf{h}'$ to predict the target assignment $\hat{\mathbf{w}}''$. Here $\hat{\mathbf{w}}''$ are balanced soft assignments of $\mathbf{s}''$ via Sinkhorn-Knopp algorithm (Cuturi, 2013). Details of these computations are provided in the EnvInfer architecture (Figure 8; Appendix C.4). Therefore, the swapped loss encourages different augmentations of the same sample to yield consistent environment assignments:

$$\mathcal{L}_{\text{sw}} = \ell(\mathbf{h}', \hat{\mathbf{w}}'') + \ell(\mathbf{h}'', \hat{\mathbf{w}}'), \tag{8}$$

where $\ell(\cdot)$ denotes cross-entropy. In addition, an orthogonality loss $\mathcal{L}_{\text{orth}} = \|\mathbf{E}^\top \mathbf{E} - \mathbf{I}\|_F^2$ is applied to encourage diversity among the environment embeddings. As a result, the final training objective is: $\mathcal{L} = \mathcal{L}_{\text{eps}} + \alpha \mathcal{L}_{\text{sw}} + \beta \mathcal{L}_{\text{orth}}$, where $\alpha$ and $\beta$ are the balance coefficients. The full training procedure is summarized in Algorithm 1 and described in more detail in Appendix D.

**Generation Procedures.** **CaTSG** supports both interventional and counterfactual generation via backdoor-adjusted guided noise regression. At each denoising step, EnvInfer $q_\phi$ infers posterior weights over the environment bank $\mathbf{E}$, and Denoiser produces both unconditional prediction $\boldsymbol{\epsilon}^{\text{base}}$ and environment-aware predictions $\boldsymbol{\epsilon}_k^{\text{env}}$ for each environments $\mathbf{e}_k$. Under the standard diffusion formulation, regressing Gaussian noise is equivalent to estimating the score function (Ho et al., 2020), so $\hat{\boldsymbol{\epsilon}}$ provides a Monte Carlo approximation to the causal score defined in Eq. 6: $\hat{\boldsymbol{\epsilon}} = (1+\omega) \sum_{k=1}^{K} w_k \boldsymbol{\epsilon}_k^{\text{env}} - \omega \boldsymbol{\epsilon}^{\text{base}}$. Therefore, these predictions are aggregated by the backdoor-adjusted rule, yielding the practical noise estimate $\hat{\boldsymbol{\epsilon}}$, which replaces the vanilla noise in the DDPM update. This procedure is repeated for $t = T, \ldots, 1$ to produce the final sample. This generic process forms the backbone of generation, while the interventional and counterfactual settings differ only in their initialization and conditioning strategy. We briefly describe these two cases below, with the full procedures provided in Algorithms 2 and 3.

- **Intervention generation.** Given a context $\mathbf{c}$, **CaTSG** generates samples from the interventional distribution $P(X \mid \text{do}(C = \mathbf{c}))$. EnvInfer provides environment weights conditioned on $w_k = P(\mathbf{e}_k \mid \mathbf{x}_t, \mathbf{c})$, which are used in the backdoor-adjusted update so that the denoising process follows the interventional score function $s_t^{\text{int}}$.

- **Counterfactual generation.** Given an observed sample $(\mathbf{x}_0, \mathbf{c})$ and a counterfactual context $\mathbf{c}'$, **CaTSG** generates $P(X' \mid X = \mathbf{x}_0, C = \mathbf{c}, C' = \mathbf{c}')$ following the AAP procedure (Section 4.1). In the *abduction* step, EnvInfer

infers the environment posterior from $(\mathbf{x}_0, \mathbf{c})$: $w_k = P(\mathbf{e}_k \mid \mathbf{x}_t, \mathbf{c}), k = \{1, \dots, K\}$ ; in the *action* step, the original context $\mathbf{c}$ is replaced with $\mathbf{c}'$; and in the *prediction* step, the denoising process proceeds as above with backdoor-adjusted guidance via the score function $s_t^{\mathrm{cf}}$.

# 5 EXPERIMENTS

In this section, we empirically validate `CaTSG` by addressing the following Research Questions (**RQ**s): **RQ1.** Does incorporating environment-level interventions improve generation fidelity under environment shifts? **RQ2.** Can `CaTSG` generate counterfactual time series that are faithful? **RQ3.** Are the learned environment representations aligned with underlying generative factors? **RQ4.** How does each components of `CaTSG` contribute to overall performance?

## 5.1 EXPERIMENTAL SETTINGS

**Datasets & Baselines.** We conduct experiments on two synthetic datasets and two real-world datasets. The synthetic datasets, i.e., **Harmonic-VM** and **Harmonic-VP**, are generated from a simulated damped harmonic oscillator governed by second-order differential equations (see Appendix E.1), which provide ground-truth counterfactuals for direct evaluation. For real-world datasets, we use **Air Quality** (Chen, 2019) and **Traffic** (Hogue, 2019) datasets, which do not contain ground-truth counterfactuals (Appendix E.2). These variables are used only for splitting, and the environment labels themselves remain unobserved in the data. As baselines, we consider five representative conditional time series generative models: TimeGAN (Yoon et al., 2019), WaveGAN (Donahue et al., 2019), Pulse2Pulse (Thambawita et al., 2021), and TimeWeaver (Narasimhan et al., 2024) in its two forms, TimeWeaver-CSDI and TimeWeaver-SSSD. Further descriptions of these baselines are provided in Appendix F.1.

**Evaluation Metrics.** Following previous TSG works, we choose marginal distribution distance (MDD), Kullback–Leibler divergence (KL), maximum mean discrepancy (MMD) for evaluation. We also include a metric for conditional generation evaluation, Joint Frechet Time Series Distance (J-FTSD) proposed by Narasimhan et al. (2024), measuring the differences of $c$ and $x$'s joint distribution. Details on these metrics are presented in Appendix F.2.

**Implementation Details.** We implement `CaTSG` in PyTorch 1.13.0 with CUDA 11.7. For denoiser $\varepsilon_\theta$, we adopt a conventional U-Net with an encoder–bottleneck–decoder structure and skip connections. For swapped prediction, we set the softmax temperature to $\tau = 0.1$. For sampling, we use the second-order single-step DPM-Solver (Lu et al., 2022) with 20 steps. Final per-dataset settings are reported in Appendix F.3.

## 5.2 EFFECT OF ENVIRONMENT-AWARE GENERATION

To address **RQ1**, we create "environments"-based splits for all datasets using selected variables as proxies for latent contexts (Table 5). Baselines model observational TSG, i.e., generating samples from $P(X \mid C)$. In contrast, `CaTSG` incorporates latent environments and generates samples under interventional distributions $P(X \mid \mathrm{do}(C))$. Table 2 reports results across synthetic (Harmonic-VM/VP) and real-world (Air Quality, Traffic) datasets. Note that standard

Table 2: Results on synthetic and real-world datasets. Best results are in bold and second-best are underlined. TW-CSDI, TW-SSSD: TimeWeaver variants based on CSDI and SSSD.

| Dataset | Metric | TimeGAN | WaveGAN | Pulse2Pulse | TW-CSDI | TW-SSSD | CaTSG (ours) |
|---------|--------|---------|---------|-------------|---------|---------|--------------|
| Harmonic-VM | MDD | $0.525 \pm 0.087$ | $0.492 \pm 0.001$ | $0.339 \pm 0.092$ | $\underline{0.267} \pm 0.001$ | $0.297 \pm 0.001$ | $\mathbf{0.098} \pm 0.001$ |
| | KL | $11.926 \pm 0.256$ | $4.518 \pm 0.076$ | $1.717 \pm 0.146$ | $\underline{0.208} \pm 0.005$ | $0.230 \pm 0.003$ | $\mathbf{0.017} \pm 0.000$ |
| | MMD | $0.554 \pm 0.458$ | $0.234 \pm 0.000$ | $1.357 \pm 1.924$ | $\underline{0.080} \pm 0.000$ | $0.160 \pm 0.002$ | $\mathbf{0.012} \pm 0.000$ |
| | J-FTSD | $10.713 \pm 0.042$ | $7.079 \pm 0.973$ | $4.885 \pm 0.773$ | $\underline{2.782} \pm 0.246$ | $3.672 \pm 0.152$ | $\mathbf{0.548} \pm 0.085$ |
| Harmonic-VP | MDD | $0.332 \pm 0.011$ | $0.308 \pm 0.000$ | $0.288 \pm 0.029$ | $\underline{0.077} \pm 0.000$ | $0.083 \pm 0.000$ | $\mathbf{0.053} \pm 0.001$ |
| | KL | $17.337 \pm 0.675$ | $10.043 \pm 0.002$ | $12.018 \pm 4.605$ | $\underline{0.208} \pm 0.006$ | $0.275 \pm 0.002$ | $\mathbf{0.066} \pm 0.003$ |
| | MMD | $0.516 \pm 0.035$ | $0.711 \pm 0.000$ | $0.474 \pm 0.078$ | $0.106 \pm 0.001$ | $\underline{0.084} \pm 0.000$ | $\mathbf{0.034} \pm 0.001$ |
| | J-FTSD | $28.933 \pm 0.912$ | $38.522 \pm 6.362$ | $28.163 \pm 10.497$ | $\underline{13.114} \pm 2.464$ | $16.142 \pm 1.428$ | $\mathbf{12.677} \pm 2.431$ |
| Air Quality | MDD | $0.199 \pm 0.037$ | $0.254 \pm 0.000$ | $\underline{0.081} \pm 0.000$ | $0.100 \pm 0.000$ | $0.084 \pm 0.000$ | $\mathbf{0.069} \pm 0.000$ |
| | KL | $14.990 \pm 3.219$ | $16.513 \pm 0.000$ | $1.938 \pm 0.001$ | $0.502 \pm 0.004$ | $\underline{0.357} \pm 0.005$ | $\mathbf{0.301} \pm 0.001$ |
| | MMD | $0.224 \pm 0.113$ | $1.005 \pm 0.000$ | $0.116 \pm 0.000$ | $0.226 \pm 0.000$ | $\mathbf{0.059} \pm 0.001$ | $\underline{0.070} \pm 0.000$ |
| | J-FTSD | $12.749 \pm 1.132$ | $10.302 \pm 0.628$ | $16.555 \pm 4.784$ | $\underline{3.757} \pm 0.634$ | $17.191 \pm 1.251$ | $\mathbf{2.929} \pm 0.497$ |
| Traffic | MDD | $0.443 \pm 0.017$ | $0.375 \pm 0.000$ | $0.231 \pm 0.000$ | $0.094 \pm 0.000$ | $\underline{0.091} \pm 0.000$ | $\mathbf{0.071} \pm 0.000$ |
| | KL | $19.339 \pm 0.110$ | $12.101 \pm 0.175$ | $4.228 \pm 0.001$ | $\mathbf{0.155} \pm 0.001$ | $0.157 \pm 0.001$ | $\underline{0.172} \pm 0.045$ |
| | MMD | $0.404 \pm 0.002$ | $0.547 \pm 0.000$ | $0.153 \pm 0.000$ | $\underline{0.033} \pm 0.000$ | $0.038 \pm 0.000$ | $\mathbf{0.006} \pm 0.000$ |
| | J-FTSD | $10.516 \pm 0.326$ | $23.649 \pm 4.027$ | $21.076 \pm 0.776$ | $\underline{6.564} \pm 0.525$ | $7.880 \pm 0.726$ | $\mathbf{0.723} \pm 0.055$ |

deviations for diffusion-based methods reflect variability over five sampling runs, while GAN-based baselines are averaged over five independent trainings. Significance and confidence interval analysis are presented in Appendix G.3. According to the table, across four datasets and four metrics (16 comparisons), `CaTSG` gains the best score in 14/16 cases. At the family level, diffusion-based models (TW-CSDI, TW-SSSD) reduce error by an average of 60.4% compared with GAN-based models (TimeGAN, WaveGAN, Pulse2Pulse). This advantage is consistent across metrics, with only a single cell showing a slight disadvantage. Building on this, `CaTSG` improves over the best GAN baseline by an average of 73.1%. Relative to the best diffusion baseline, it yields an additional average reduction of 41.4%. Diffusion methods already outperform GANs for conditional TSG. By targeting the interventional distribution, `CaTSG` further improves generation fidelity under environment shifts through autonomous inference of latent environments. The gains are most pronounced on the synthetic datasets, where shift patterns and counterfactual ground truth are controlled. In these settings, `CaTSG` achieves the largest margins on distributional and geometric criteria such as MMD and J-FTSD, while still delivering consistent improvements on real datasets.

## 5.3 COUNTERFACTUAL GENERATION ABILITY

For **RQ2**, since ground-truth counterfactuals are available only for synthetic data, we report quantitative results for CF-TSG on two Harmonic datasets (Table 3). For real-world datasets, we present visualizations (Figure 5, left). Additional discussion of potential evaluation protocols for real-world counterfactual generation is provided in Appendix G.4. We first report the quantitative evaluation on synthetic data. Since existing base-

Table 3: Counterfactual generation results.

| Method | Harmonic-VM | | Harmonic-VP | |
|---|---|---|---|---|
| | MDD | KL | MDD | KL |
| `CaTSG` | **0.374** $\pm$ 0.002 | **0.210** $\pm$ 0.004 | **0.256** $\pm$ 0.001 | **0.660** $\pm$ 0.100 |
| **RandEnv** | 0.469 $\pm$ 0.002 | 0.691 $\pm$ 0.014 | 0.272 $\pm$ 0.001 | 2.243 $\pm$ 0.045 |
| **w/o SW** | 0.474 $\pm$ 0.004 | 0.816 $\pm$ 0.027 | 0.275 $\pm$ 0.001 | 2.144 $\pm$ 0.066 |
| **FrozenEnv** | 0.462 $\pm$ 0.001 | 0.760 $\pm$ 0.016 | 0.271 $\pm$ 0.002 | 2.490 $\pm$ 0.056 |

lines are not applicable in this setting, we therefore compare `CaTSG` against its ablated variants (see detail settings in Table 6). As shown in Table 3, `CaTSG` consistently achieves the best performance on both datasets in terms of MDD and KL, highlighting the importance of accurate environment inference for counterfactual generation.

For qualitative analysis on real-world data, we provide visualization results (Figure 4). On Traffic dataset, we select test samples whose factual temperature exceeds $22°C$ (example $\mathbf{x}$ in Figure 4a). Note that temperature is *not* included in the contextual vector $\mathbf{c}$ but is used solely to define the environment split (see Table 5 for details). We first *abduct* the environment posterior $\mathbf{w}$ from $(\mathbf{x}, \mathbf{c})$ and keep it fixed. We then *action* on $\mathbf{c}$ to obtain $\mathbf{c}'$ by setting the snow indicator to 1 to serve as a counterfactual assignment incompatible with the factual temperature (i.e., $> 22°C$). Next, we *predict* (i.e., generate) the resulting traffic volumes $\hat{\mathbf{x}}'$ (examples in Figure 4b). The generated sequences preserve the temporal structure of the factual traffic volumes while exhibiting a reduced amplitude. To corroborate this effect at scale, Figure 4c compares the global distributions of $\mathbf{x}$ and $\hat{\mathbf{x}}'$ using histograms and empirical cumulative distribution functions (ECDFs). Relative to $\mathbf{x}$, the ECDF of $\hat{\mathbf{x}}'$ lies to the left and is steeper around the median. Together with the narrower histogram, this pattern indicates a lower location and reduced dispersion (smaller amplitude) in the counterfactual "snow in summer" setting.

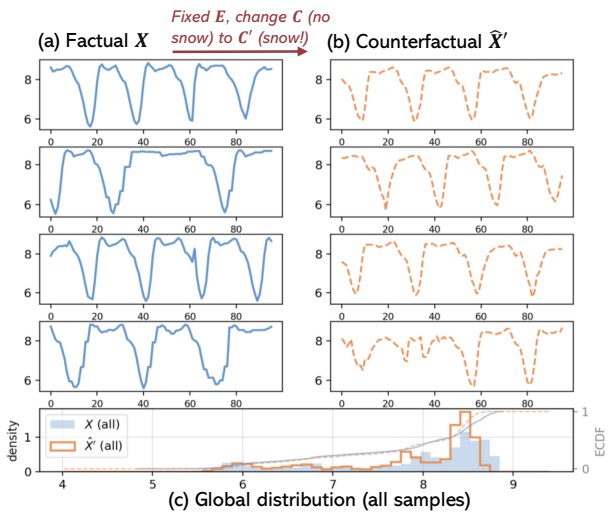

Figure 4: (a) Factual samples $X$ and (b) counterfactual generations $\hat{X}'$ on Traffic. (c) Global distribution over all samples: histogram and ECDF of $X$ (blue) and $\hat{X}'$ (orange).

## 5.4 INTERPRETATION ANALYSIS

To answer **RQ3**, we present environment interpretation analysis. Note that the environment bank $\mathbf{E}$ is latent, and its size $K$ is a dataset-specific hyperparameter (not tied to the number of splits or any variables), enabling `CaTSG` to capture richer context structure and more complex scenarios. We visualize the learned environments and posteriors for Harmonic-VM with $K = 8$. In Figure 5a, bars show the mean soft posterior per environment, and grey markers denote hard assignments frequency. Their close agreement indicates confident and consistent environment inference, and the moderate shifts across splits reflect the intended split-specific context. In Figure 5b, t-SNE reveals compact clusters around the environment embeddings (denoted as stars). The limited overlap and cross-split shift indicate that the environment bank is identifiable and transferable.

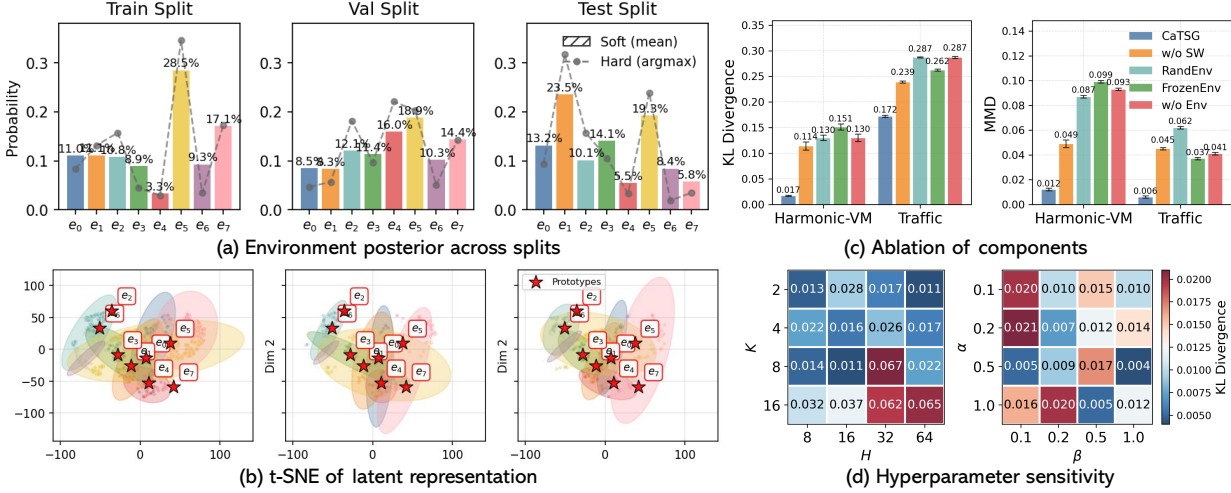

Figure 5: (a) Environment posterior $\mathbf{w}$ across splits. (b) t-SNE of representations on Harmonic-VM. Shaded ellipses are one standard-deviation regions. (c) Ablation of components. (d) Hyperparameter sensitivity on Harmonic-VM.

## 5.5 ABLATION STUDY

For **RQ4**, we conduct a series of ablation studies on core components in the proposed model, hyperparameter sensitivity and the speed–quality trade-off.

**Effects of Core Components.** To evaluate the contribution of each core component in `CaTSG`, we conducted an ablation study using the following variants: a) **RandEnv**: replace $p(e)$ with random assignments. b) **w/o** $L_{sw}$: Disable $L_{sw}$. c) **FrozenEnv**: Freeze the environment bank. d) **w/o Env**: Remove the environment bank. Detail settings on these variants are shown in Table 6.

Figure 6: #Steps v.s. performance on Harmonic-VM.

| Steps | Runtime (s) | MDD | KL |
|---|---|---|---|
| **5** | $5.49 \pm 0.45$ | $0.231 \pm 0.002$ | $0.244 \pm 0.006$ |
| **10** | $9.96 \pm 1.39$ | $0.122 \pm 0.002$ | $0.035 \pm 0.001$ |
| **20** | $19.30 \pm 2.85$ | $0.098 \pm 0.001$ | $0.017 \pm 0.000$ |
| **50** | $54.63 \pm 2.22$ | $0.088 \pm 0.001$ | $0.014 \pm 0.000$ |
| **100** | $107.01 \pm 4.09$ | $0.087 \pm 0.001$ | $0.013 \pm 0.000$ |

Figure 5c reports their results across Harmonic-VM and Traffic datasets, showing that all components contribute significantly to model's overall performance.

**Hyperparameter Sensitivity.** Figure 5d shows the impact of number of environment embeddings $K$, hidden dimension $H$, and loss coefficients $\alpha$ and $\beta$ on KL divergence on Harmonic-VM. Overall, the KL varies only modestly, indicating that `CaTSG` is broadly robust to hyperparameter choices. Two mild trends emerge: (1) small–to–moderate $K \in \{2, 4, 8\}$ with compact hidden size $H \in \{8, 16\}$ give the lowest KL and (2) for loss weights, a shallow "good" basin appears for moderate/large values ($\alpha, \beta \in [0.5, 1.0]$).

**Speed & Quality Trade-off.** We ablate the number of DPM-Solver steps on Harmonic-VM (Table 6). Increasing the step count consistently improves MDD and KL, while runtime scales approximately linearly. A 20-step sampler offers a clear Pareto point: it is $5.5\times$ faster than 100 steps yet remains within $0.011$ MDD and $0.004$ KL of the best scores. Variances are small across settings, suggesting stable sampling. We therefore adopt 20 steps as the default in main experiments. Full results are provided in Table 8 (Appendix G.2).

## 6 CONCLUSION AND FUTURE WORK

In this work, we presented a causal perspective for conditional TSG. By positioning conditional TSG tasks within Pearl's causal ladder, we extended the conventional observational setting to include interventional and counterfactual generation. To support these new tasks, we proposed an SCM and derived the corresponding objectives using backdoor adjustment and abduction–action–prediction. Grounded in these causal principles, we developed `CaTSG`, a diffusion-based generative framework equipped with a causal score function that unifies all three-level TSG. Extensive experiments on both synthetic and real-world datasets demonstrated that `CaTSG` achieves high-quality generation, faithfully recovers interventional and counterfactual distributions.

While we provide a proof-of-concept, it still highlights several open challenges, such as relaxing the reliance on predefined SCMs, multi-modal conditions, and developing systematic evaluation protocols for counterfactuals in real-world data. These challenges also mark rich opportunities for future research toward more flexible and generalizable causal generative models and evaluation framework.

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

## A  NOTATION

Table 4: **Notation summary**. Random variables are uppercase math italic (e.g., $\mathcal{X}$), vectors are lowercase bold (e.g., $\mathbf{x}$), matrices are uppercase bold (e.g., $\mathbf{X}$), and scalars are non-bold (e.g., $x$).

| Symbol | Description |
|---|---|
| $X, C, E$ | Variables: target time series, contextual variable, latent environment |
| $\mathbf{x}, \mathbf{c}, \mathbf{e}$ | Sample of time series, context, or environment |
| $\mathrm{do}(\cdot)$ | do-calculus, an intervention operation that sets a variable independent of its causes |
| $\boldsymbol{\epsilon}$ | Noise vector sampled from Gaussian $\mathcal{N}(0, I)$ |
| $\hat{\boldsymbol{\epsilon}}$ | Predicted noise (from denoising model) |
| $p_\theta(\cdot), \mathcal{F}_\theta(\cdot)$ | Learnable generative model (e.g., diffusion model) |
| $q_\phi(\cdot), \varepsilon_\theta(\cdot)$ | EnvInfer and Denoiser module |
| $\mathbf{E} = \{\mathbf{e}_1, \ldots, \mathbf{e}_K\}$ | A learnable latent environment bank of size $K$ |
| $\mathbf{w} = \{w_1, \ldots, w_K\}$ | Predicted environment assignment probability. |
| $s_t(\cdot)$ | Score function |
| $\mathcal{L}_{\mathrm{eps}}, \mathcal{L}_{\mathrm{sw}}, \mathcal{L}_{\mathrm{orth}}$ | Noise prediction loss, swapped prediction loss and orthogonal loss |
| $\mathbf{s}, \hat{\mathbf{w}}$ | Predicted logits and balanced soft assignments by applying Sinkhorn–Knopp to the batch of logits |
| $V$ | Number of augmentations for each samples for swapped prediction loss $\mathcal{L}_{\mathrm{sw}}$ |
| $\alpha, \beta$ | Balance coefficients for $\mathcal{L}_{\mathrm{sw}}$ and $\mathcal{L}_{\mathrm{orth}}$ |
| $\sigma_t^2$ | Noise variance at diffusion step $t$ |
| $H, N$ | Hidden dimension, the number of samples |
| $t \in \{1, \ldots, T\}$ | Diffusion timestep |
| $\omega$ | Weight for controlling conditional-vs-unconditional trade-off |

## B  MORE DETAILS ON PRELIMINARIES

Due to space constraints in the main text, we provide additional preliminaries in this appendix. Specifically, we elaborate on two key components: **(B.1) causal concepts**, which is related to the motivation and theoretical foundations of our proposed task, and **(B.2) diffusion models**, which is related to the technical background of the backbone architecture of our proposed model. These preliminaries are covered in this section, while the full derivation of the proposed CaTSG model is deferred to Appendix C.2.

### B.1  CAUSAL CONCEPTS

We now present the causal foundations relevant to our work. We begin by introducing Structural Causal Models (SCMs) (Pearl et al., 2000; Pearl, 2009), followed by a brief review of the rules of do-calculus (Pearl et al., 2016), which enable formal reasoning under interventions. We then derive the backdoor adjustment formula (Figure 2c) and introduce the three-step process of counterfactual generation (Figrue 2d).

**Structural Causal Model.**  A Structural Causal Model (SCM) (Pearl, 2009) is defined as a directed graph where nodes represent variables and directed edges denote direct causal dependencies between them. This graph-based representation allows us to reason about how interventions propagate through a system. Example SCMs are shown in Figure 7.

**Three rules of Do-calculus.**  The *do-calculus* (Pearl et al., 2016) is a formal system that enables the identification of interventional distributions, such as $P(Y \mid \mathrm{do}(X))$, from purely observational data, given a SCM. Its core goal is to transform expressions involving interventions (i.e., $\mathrm{do}(\cdot)$) into expressions involving only observed variables. Do-calculus consists of three transformation rules that operate on probabilistic expressions involving the *do*-operator, leveraging the conditional independencies encoded in the causal graph.

Figure 7: Illustration of (a) original DAG $\mathcal{G}$, (b) interventional graph $\mathcal{G}_{\mathrm{do}(X)}$, and (c) nullified graph $\mathcal{G}_{\mathrm{null}(X)}$

Before introducing the rules, we first define the relevant graph transformations used to represent interventions and their structural consequences as shown in Figure 7. Let $\mathcal{G}$ be a directed acyclic graph (DAG) with nodes $X$, $Y$, and $Z$ (confounder) (Figure 7a). The *interventional graph* $\mathcal{G}_{\mathrm{do}(X)}$ is defined by removing all incoming edges to $X$ (Figure 7b), while the *nullified graph* $\mathcal{G}_{\mathrm{null}(X)}$ removes all outgoing edges from $X$ (Figure 7c).

Based on these graph structures, the following three rules govern valid manipulation of probabilities under interventions Pearl et al. (2016):

- **Rule 1 (Insertion/deletion of observations):**

$$P(y \mid do(x), z) = P(y \mid do(x)) \quad \text{if } (y \perp\!\!\!\perp z \mid x)_{G_{do(x)}}$$

- **Rule 2 (Action/observation exchange):**

$$P(y \mid do(x), do(z)) = P(y \mid do(x), z) \quad \text{if } (y \perp\!\!\!\perp z \mid x)_{G_{do(x), null(z)}}$$

- **Rule 3 (Insertion/deletion of actions):**

$$P(y \mid do(x), do(z)) = P(y \mid do(x)) \quad \text{if } (y \perp\!\!\!\perp z \mid x)_{G_{do(x), do(z)}}$$

Here, $(y \perp\!\!\!\perp z \mid x)_{\mathcal{G}}$ denotes conditional independence of $y$ and $z$ given $x$ in the graph $\mathcal{G}$. For instance, Rule 1 states that if $x$ d-separates $y$ from $z$ in $\mathcal{G}_{\mathrm{do}(x)}$, then $z$ can be removed from the conditional set.

**Backdoor Adjustment.** We now apply these rules to derive the backdoor adjustment in Figure 2c, where we conduct the $do$-operation on variable $C$, which we treat as a confounder influencing both $C$ and $X$. Using $do$-calculus, we derive:

$$
\begin{aligned}
P(X \mid \mathrm{do}(C)) &= \sum_e P(X \mid \mathrm{do}(C), E = e)\, P(E = e \mid \mathrm{do}(X)) \quad \textit{(Bayes rule)} \\
&= \sum_e P(X \mid \mathrm{do}(C), E = e)\, P(E = e) \quad \textit{(Rule 3)} \\
&= \sum_e P(X \mid C, E = e)\, P(E = e) \quad \textit{(Rule 2)}
\end{aligned}
\tag{9}
$$

**Three-Step Counterfactuals Computing.** Counterfactual reasoning aims to answer questions of the form: *"What would have happened, had a certain condition been different?"* In the language of SCMs, a counterfactual query typically takes the form $P(Y_{x'} \mid X = x, Y = y)$, which quantifies the outcome $Y$ under a hypothetical intervention $do(X = x')$, given that we observed $X = x$ and $Y = y$ in the real world.

To compute such queries, taking our case as an example (Figure 2d), SCMs follow a principled three-step procedure (Pearl et al., 2000):

- **Abduction**: Infer the latent background variables $E$ from the observed variables, i.e., estimate $P(E \mid X, C)$.
- **Action**: Modify the model by applying an intervention $\mathrm{do}(C = C')$, which removes all incoming edges to $C$ and sets it to a new value $C'$.
- **Prediction**: Use the modified model and the inferred $E$ to compute the counterfactual outcome, i.e., evaluate $P(X' \mid E, C')$.

The resulting counterfactual distribution is given by:

$$P(X' \mid X, C, C') = \sum_e P(X' \mid \mathrm{do}(C'), E = e)\, P(E = e \mid X, C) \tag{10}$$

## B.2 Diffusion Models and Conditional Diffusion Models

Diffusion models (Sohl-Dickstein et al., 2015; Ho et al., 2020) are a class of probabilistic generative models that learn to generate data through a two-phase procedure: *a forward diffusion process* that progressively adds noise to the data, followed by *a reverse denoising process* that learns to recover the original data distribution. In recent years, diffusion models have become a powerful paradigm for time series modeling, achieving strong performance across a wide range of tasks (Tashiro et al., 2021; Wen et al., 2023; Yuan & Qiao, 2024; Yang et al., 2024; Lin et al., 2024). We begin by introducing the unconditional setting, detailing both the forward and reverse processes. We then introduce the extension of this framework to the conditional setting, where external factors are incorporated to guide the generation process.

### B.2.1 Diffusion Models

To simplify the presentation, we begin by considering the standard generation setting, without restricting ourselves to time series inputs.

**Forward Process.** Diffusion probabilistic models (Sohl-Dickstein et al., 2015) define a generative process by reversing a Markovian forward noising process. Let $x_0 \in \mathbb{R}^d$ denote a clean data sample drawn from an unknown data distribution $q(x_0)$. The forward process gradually adds Gaussian noise over $T$ discrete time steps to destroy the data:

$$q(x_{1:T}|x_0) = \prod_{t=1}^{T} q(x_t|x_{t-1}), \quad q(x_t|x_{t-1}) = \mathcal{N}(x_t; \sqrt{1-\beta_t}x_{t-1}, \beta_t \mathbf{I}), \tag{11}$$

where $\beta_t \in (0,1)$ is a pre-defined variance schedule. Thanks to the Gaussian property, we can marginalize the entire forward process in closed form:

$$q(x_t|x_0) = \mathcal{N}(x_t; \sqrt{\bar{\alpha}_t}x_0, (1-\bar{\alpha}_t)\mathbf{I}), \tag{12}$$

where $\alpha_t := 1 - \beta_t$ and $\bar{\alpha}_t := \prod_{s=1}^{t} \alpha_s$.

**Reverse Process.** The generative model learns to reverse this process by estimating the posterior $p_\theta(x_{t-1}|x_t)$ with a neural network. The model is defined as a chain of conditional distributions:

$$p_\theta(x_{0:T}) := p(x_T) \prod_{t=1}^{T} p_\theta(x_{t-1}|x_t), \tag{13}$$

with Gaussian transitions:

$$p_\theta(x_{t-1}|x_t) = \mathcal{N}(x_{t-1}; \mu_\theta(x_t, t), \Sigma_\theta(x_t, t)). \tag{14}$$

**Training Objective.** To enable efficient training, we first leverage the closed-form expression of the forward process in Eq 12, which allows us to directly sample a noisy input $x_t$ at any timestep $t$ from the clean data $x_0$ via:

$$x_t = \sqrt{\bar{\alpha}_t}x_0 + \sqrt{1-\bar{\alpha}t}\epsilon, \quad \epsilon \sim \mathcal{N}(0, \mathbf{I}). \tag{15}$$

Instead of directly maximizing the log-likelihood $\log p\theta(x_0)$, the model is trained by optimizing a variational lower bound (ELBO). Ho et al. (2020) show that this is equivalent to minimizing a simplified denoising objective that predicts the added noise $\epsilon$:

$$\mathcal{L} = \mathbb{E}x_0, \epsilon, t \left[ |\epsilon - \varepsilon_\theta(x_t, t)|^2 \right]. \tag{16}$$

This objective guides the denoising network $\varepsilon_\theta$ to recover the noise component from the perturbed observation $x_t$, thereby implicitly learning the score function of the data distribution.

**Sampling Procedure.** During inference, the model generates samples by iteratively denoising:

$$x_{t-1} = \frac{1}{\sqrt{\alpha_t}} \left( x_t - \frac{1-\alpha_t}{\sqrt{1-\bar{\alpha}_t}} \varepsilon_\theta(x_t, t) \right) + \sigma_t z, \quad z \sim \mathcal{N}(0, \mathbf{I}), \tag{17}$$

where $x_{t-1}$ refers to the noisy sample at time step $t$, $\varepsilon_\theta(x_t, t)$ denotes the predicted noise by the denoising model $\varepsilon_\theta$, $\alpha_t$ and $\bar{\alpha}_t$ denote the noise schedule parameters, $\sigma_t^2 = \beta_t$ or some learnable variance, $z$ denotes the standard Gaussian noise.

**Score Function Interpretation.** Diffusion models can also be interpreted as a special class of *score-based generative models* (Song & Ermon, 2019), which aim to learn the gradient of the log-density of the data at each timestep. This gradient, known as the *score function*, indicates the direction in which the noisy input $x_t$ should be adjusted to increase its likelihood under the marginal distribution $p_t(x_t)$. The score function is formally defined as:

$$s_t(x_t) := \nabla_{x_t} \log p(x_t). \tag{18}$$

Ho et al. (Ho et al., 2020) show that, under the Gaussian forward process (Eq. 15), learning to approximate this score is equivalent to learning to predict the noise $\epsilon$ added during forward diffusion. This leads to the following approximate relationship:

$$\nabla_{x_t} \log p(x_t) \approx -\frac{1}{\sqrt{1 - \bar{\alpha}_t}} \cdot \varepsilon_\theta(x_t, t) \propto \varepsilon_\theta(x_t, t). \tag{19}$$

This result provides a principled link between score-based modeling and the denoising objective in DDPMs: the trained network $\varepsilon_\theta$ can be interpreted as predicting the (scaled) score function at each timestep.

### B.2.2 CONDITIONAL DIFFUSION MODELS

**Conditional Setup.** To enable guided generation under external control or domain-specific information (e.g., class labels, time-of-day, etc.), diffusion models can be extended to the conditional setting. Here we show the case where the generative process is conditioned on an auxiliary variable $c$, and both the reverse transition and the denoising network are modified accordingly. Specifically, the reverse process in Eq. 14 becomes:

$$p_\theta(x_{t-1} \mid x_t, c) = \mathcal{N}(\mu_\theta(x_t, t, c), \Sigma_\theta(x_t, t, c)), \tag{20}$$

where the network $\varepsilon_\theta(x_t, t, c)$ now takes $c$ as an additional input to predict the noise added during the forward process. This conditional formulation allows the model to generate samples consistent with the specified condition $c$, providing a foundation for controllable or guided generation tasks. Accordingly, instead of learning the unconditional score function $s_t(x_t)$ in Eq.18, we aim to estimate the conditional score $s_t(x_t, c)$:

$$
\begin{aligned}
s_t(x_t, c) &:= \nabla_{x_t} \log p(x_t \mid c) && \textit{(Difinition)} \\
&\approx -\frac{1}{\sqrt{1 - \bar{\alpha}_t}} \cdot \varepsilon_\theta(x_t, t, c) && \textit{(Similar to Eq.19)} \\
&\propto \varepsilon_\theta(x_t, t, c)
\end{aligned}
\tag{21}
$$

**Classifier-Free Guidance.** One classical perspective decomposes the conditional score $\nabla_{x_t} \log p(x_t \mid c)$ in Eq. 21 via Bayes' rule:

$$\nabla_{x_t} \log p(x_t \mid c) = \nabla_{x_t} \log \left( \frac{p(x_t)p(c \mid x_t)}{p(c)} \right) = \nabla_{x_t} \log p(x_t) + \nabla_{x_t} \log p(c \mid x_t) \tag{22}$$

where the first term $\nabla_{x_t} \log p(x_t)$ corresponds to the gradient of the unconditional model (Eq 19), and the second term $\nabla_{x_t} \log p(c \mid x_t)$ introduces an additional signal from the classifier $p(c \mid x_t)$. This decomposition reveals the principle behind classifier guidance: to improve conditional alignment during generation, one may simply augment the unconditional score with a gradient term derived from a classifier.

Ho & Salimans (2022) propose *Classifier-Free Guidance* (CFG), which can be interpreted as a reparameterization of classifier guidance by manipulating the gradient expression of the conditional log-probability. Starting from Bayes' rule, we can write the second term in Eq 22 as:

$$\nabla_{x_t} \log p(c \mid x_t) = \nabla_{x_t} \log p(x_t \mid c) - \nabla_{x_t} \log p(x_t) \tag{23}$$

Plugging this into back into the classifier guidance formulation Eq. 22 with a guidance scale $\omega$ to control the strength of guidance, we get:

$$
\begin{aligned}
\nabla_{x_t} \log p(x_t \mid c) &= \nabla_{x_t} \log p(x_t) + \omega \nabla_{x_t} \log p(c \mid x_t) && \textit{(Introduce guidance scale $\omega$)} \\
&= \nabla_{x_t} \log p(x_t) + \omega \left( \nabla_{x_t} \log p(x_t \mid c) - \nabla_{x_t} \log p(x_t) \right) && \textit{(Plugging Eq. 23)} \\
&= \omega \nabla_{x_t} \log p(x_t \mid c) - (\omega - 1) \nabla_{x_t} \log p(x_t) && \textit{(Algebraic rearrangement)} \\
&= \omega \cdot s_t(x_t, c) - (\omega - 1) \cdot s_t(x_t) && \textit{(Score-based rewriting)} \\
&\propto \omega \cdot \varepsilon_\theta(x_t, t, c) - (\omega - 1) \cdot \varepsilon_\theta(x_t, t) && \textit{(According to Eq. 19 and 21)} \\
&= (1 + \omega) \cdot \varepsilon_\theta(x_t, t, c) - \omega \cdot \varepsilon_\theta(x_t, t) && \textit{(Rewriting)}
\end{aligned}
$$

where $\varepsilon_\theta(x_t, t, c)$ denotes the conditional noise prediction networks, $\varepsilon_\theta(x_t, t)$ the unconditional one, and $\omega$ is a scalar that controls the guidance scale or strength. Therefore, the conditional score can be written as follow:

$$s_t(x_t, c) = \nabla_{x_t} \log p(x_t \mid c) \propto (1 + \omega) \cdot \varepsilon_\theta(x_t, t, c) - \omega \cdot \varepsilon_\theta(x_t, t) \tag{24}$$

In practice, CFG reformulates conditional generation as a single denoising network training task. This is achieved by randomly dropping the condition $c$ to an empty set $c \leftarrow \varnothing$ during training. This technique allows the network to learn to predict both conditional and unconditional noise simultaneously. Consequently, at inference time, guidance is performed by interpolating between these two predictions to steer the generation process.

## C  MORE DETAILS ON MODEL DESIGN

### C.1  MOTIVATION FOR A DIFFUSION BACKBONE

Our choice of a diffusion backbone is driven by the goals of Int-TSG and CF-TSG. First, under the score-based view, diffusion models learn and approximate scores $s(x, c) = \nabla_x \log p(x \mid c)$ (Eq. 1). Given identifiability of the interventional target $p(x \mid \mathrm{do}(c))$ instead of $p(x \mid c)$, our objective naturally extends to the interventional score $s(x, \mathrm{do}(c)) = \nabla_x \log p(x \mid \mathrm{do}(c))$, which can be derived from the backdoor-adjusted formulation (Eq. 2). The derivation will be introduced in following sections. Second, diffusion models have become a powerful paradigm for time series modeling, achieving strong performance across imputation, generation, and forecasting tasks (Tashiro et al., 2021; Wen et al., 2023; Yuan & Qiao, 2024; Yang et al., 2024; Lin et al., 2024), offering stable training and good coverage compared to likelihood-free adversarial alternatives. These properties make diffusion a pragmatic and principled backbone for our causal TSG agenda.

### C.2  DERIVATION OF BACKDOOR-ADJUSTED SCORE FUNCTION

**Goal.**  For the interventional and counterfactual generation tasks, rather than sampling from the observational distribution $p(x \mid c)$, we aim to perform guided sampling from the interventional distribution $p(x \mid \mathrm{do}(c))$ and the counterfactual distribution $p(x' \mid x, c, c')$. Accordingly, instead of the conditional score $s_t(x_t, c) = \nabla_{x_t} \log p(x_t \mid c)$ in Eq. 21, we target

$$s_t^{\mathrm{int}}(x_t; \mathrm{do}(c)) \triangleq \nabla_{x_t} \log p(x_t \mid \mathrm{do}(c)) \quad \text{and} \quad s_t^{\mathrm{cf}}(x_t'; x, c, c') \triangleq \nabla_{x_t'} \log p(x_t' \mid x, c, c').$$

For notational clarity, we refer to these causal score functions as $s_t^{\mathrm{int}}$ and $s_t^{\mathrm{cf}}$, where $s_t^{\mathrm{int}}$ denotes the interventional score function associated with the distribution $p(x_t \mid \mathrm{do}(c))$, and $s_t^{\mathrm{cf}}$ denotes the counterfactual score function associated with the distribution $p(x_t' \mid x, c, c')$. In the following, we first introduce the underlying causal assumptions, and then detail the derivation of both score functions.

**Causal assumption.**  We posit an SCM with a latent confounder $e$ that influences both $x$ and $c$ (Figure 2b). Under this SCM, we apply the backdoor adjustment (Eq. 2) to block the confounding path $c \leftarrow e \rightarrow x$ (Figure 2c) and using Abduction-Action-Prediction (Eq. 10) to do the counterfactual generation (Figure 2d). Based on this SCM and the backdoor adjustment, we derive backdoor-adjusted forms of the score functions for $s_t^{\mathrm{int}}$ and $s_t^{\mathrm{cf}}$ below.

**Interventional Score Function.**  We first derive the interventional score function $s_t^{\mathrm{int}} = \nabla_{x_t} \log p(x_t \mid \mathrm{do}(c))$. Following the standard backdoor formulation in Eq. 2, the interventional score function $s_t^{\mathrm{int}}$ can be expressed by marginalizing over $e$:

$$\nabla_x \log p(x \mid \mathrm{do}(c)) = \nabla_x \log \left( \sum_e p(x \mid c, e) \, p(e) \right). \tag{25}$$

The summation over latent confounder $e$ is inside the logarithm, preventing direct application of the gradient operator. This is a well-known difficulty in score-based inference under marginalization. To address this, we adopt a classical result often referred to as the *log-derivative of a marginal* or *softmax trick* (Mnih & Gregor, 2014), given by:

$$\nabla_x \log \sum_i a_i(x) = \sum_i \frac{a_i(x)}{\sum_j a_j(x)} \nabla_x \log a_i(x). \tag{26}$$

Applying this identity to our target objective with $a_i(x) = p(x \mid c, e_i) \, p(e_i)$, Eq. 25 can be written as:

$$\nabla_x \log \left( \sum_e p(x \mid c, e) \, p(e) \right) = \sum_e \underbrace{\frac{p(x \mid c, e) \, p(e)}{\sum_{e'} p(x \mid c, e') \, p(e')}}_{\text{posterior over } e} \cdot \nabla_x \log p(x \mid c, e). \tag{27}$$

That is, the score $s_t^{\text{int}}$ can be expressed as the expectation of conditional scores under the posterior over confounders:

$$\nabla_x \log p(x \mid \text{do}(c)) = \mathbb{E}_{e \sim p(e \mid x, c)} \left[ \nabla_{x_t} \log p(x \mid c, e) \right], \tag{28}$$

where the posterior over $e$ is given by:

$$p(e \mid x, c) = \frac{p(x \mid c, e)\, p(e)}{\sum_{e'} p(x \mid c, e')\, p(e')}. \tag{29}$$

The full derivation of $s_t^{\text{int}}$ can be summarized as follows:

$$
\begin{aligned}
\nabla_x \log p(x \mid \text{do}(c)) &= \nabla_x \log \left( \sum_e p(x \mid c, e)\, p(e) \right) && \textit{(Using the backdoor adjustment in Eq. 2)} \\
&= \sum_e \frac{p(x \mid c, e)\, p(e)}{\sum_{e'} p(x \mid c, e')\, p(e')} \cdot \nabla_x \log p(x \mid c, e) && \textit{(Softmax trick in Eq. 26)} \\
&= \mathbb{E}_{e \sim p(e \mid x, c)} \left[ \nabla_{x_t} \log p(x \mid c, e) \right] && \textit{(Where posterior over $e$ is given in Eq. 29)} \\
&\propto \mathbb{E}_{e \sim p(e \mid x, c)} \left[ (1 + \omega)\, \varepsilon_\theta(x_t, t, c, e) - \omega\, \varepsilon_\theta(x_t, t) \right] && \textit{(CFG-style parameterization in Eq. 24)} \\
&= (1 + \omega)\, \mathbb{E}_{e \sim p(e \mid x, c)} \left[ \varepsilon_\theta(x_t, t, c, e) \right] - \omega\, \varepsilon_\theta(x_t, t) && \textit{(Rewriting)}
\end{aligned}
$$

where $\omega$ denotes the guidance scale.

Therefore, we have interventional score function:

$$s_t^{\text{int}} = \nabla_{x_t} \log p(x_t \mid \text{do}(c)) \propto (1 + \omega)\, \mathbb{E}_{e \sim p(e \mid x, c)} \left[ \varepsilon_\theta(x_t, t, c, e) \right] - \omega\, \varepsilon_\theta(x_t, t) \tag{30}$$

**Counterfactual Score Function.** We now detail how to derive the counterfactual score function $s_t^{\text{cf}} = \nabla_{x_t} \log p(x_t' \mid x_t, c, c')$. Following the abduction–action–prediction procedure (Eq. 10), the full derivation of $s_t^{\text{cf}}$ can be summarized as follows:

$$
\begin{aligned}
\nabla_{x'} \log p(x' \mid x, c, c') &= \nabla_{x'} \log \left( \sum_e p(x' \mid c', e)\, p(e \mid x, c) \right) && \textit{(AAP factorization in Eq. 10)} \\
&= \sum_e \frac{p(x' \mid c', e)\, p(e \mid x, c)}{\sum_{e'} p(x' \mid c', e')\, p(e' \mid x, c)} \cdot \nabla_{x'} \log p(x' \mid c', e) && \textit{(Softmax trick in Eq. 26)} \\
&= \mathbb{E}_{e \sim p(e \mid x', x, c, c')} [\nabla_{x'} \log p(x' \mid c', e)] && \textit{(Posterior over $e$)} \\
&\approx \mathbb{E}_{e \sim p(e \mid x, c)} [\nabla_{x'} \log p(x' \mid c', e)] && \textit{(Fix $e$ from the factual pair $(x, c)$)} \\
&\propto \mathbb{E}_{e \sim p(e \mid x, c)} [(1 + \omega)\, \varepsilon_\theta(x_t', t, c', e) - \omega\, \varepsilon_\theta(x_t', t)] && \textit{(CFG-style parameterization in Eq. 24)} \\
&= (1 + \omega)\, \mathbb{E}_{e \sim p(e \mid x, c)} [\varepsilon_\theta(x_t', t, c', e)] - \omega\, \varepsilon_\theta(x_t', t) && \textit{(Rewriting).}
\end{aligned}
$$

Analogous to the derivation of the interventional score (Eq. 30), this can be expressed in terms of the denoising network outputs as

$$s_t^{\text{cf}} = \nabla_{x_t} \log p(x_t' \mid x_t, c, c') \propto (1 + \omega)\, \mathbb{E}_{e \sim p(e \mid x, c)} \left[ \varepsilon_\theta(x_t', t, c', e) \right] - \omega\, \varepsilon_\theta(x_t', t) \tag{31}$$

### C.3 Practical Discretization Implementation

Having derived the noise-prediction-based score functions in Eq. 30 and Eq. 31, we now describe how these are implemented in practice. Both score functions rely on the conditional noise prediction network $\varepsilon_\theta(x_t, t, c, e)$, its unconditional counterpart $\varepsilon_\theta(x_t, t)$, and a guidance scale $\omega$ that balances the strength of the guidance. The key computational challenge lies in evaluating the expectation

$$\mathbb{E}_{e \sim p(e \mid x_t, c)} \left[ \varepsilon_\theta(x_t, t, c, e) \right],$$

which requires marginalizing over a continuous latent confounder space. This is computationally expensive and difficult to parameterize directly.

To address this, inspired by prior works (Xia et al., 2023; Huang et al., 2025), we discretize the latent environment space into a finite set of embeddings. Specifically, we introduce a learnable environment bank $\boldsymbol{E} = \{\mathbf{e}_1, \ldots, \mathbf{e}_K\} \in$

$\mathbb{R}^{K \times d}$, where each embedding $\mathbf{e}_k \in \mathbb{R}^d$ represents a distinct latent environment and is encouraged to be mutually orthogonal.

Under this discretization, the interventional and counterfactual score functions become:

$$s_t^{\text{int}} \propto (1 + \omega) \sum_{k=1}^{K} p(e_k \mid x_t, c)\, \varepsilon_\theta(x_t, t, c, e_k) - \omega\, \varepsilon_\theta(x_t, t), \tag{32}$$

$$s_t^{\text{cf}} \propto (1 + \omega) \sum_{k=1}^{K} p(e_k \mid x_t, c)\, \varepsilon_\theta(x_t', t, c', e_k) - \omega\, \varepsilon_\theta(x_t', t), \tag{33}$$

where $p(e_k \mid x_t, c)$ is a soft posterior distribution over environments.

To estimate this posterior, we design an environment inference module **EnvInfer** $q_\phi$, which takes as input the current sample $\mathbf{x}_t$ and condition $\mathbf{c}$, and outputs a normalized weight vector:

$$(\mathbf{x}_t, \mathbf{c}) \xrightarrow{q_\phi} \mathbf{w} = \{w_1, \dots, w_K\} \in \mathbb{R}^K, \quad \text{where } \sum_{k=1}^{K} w_k = 1. \tag{34}$$

The weight vector $\mathbf{w}$ represents the posterior belief over the $K$ environment $\{\mathbf{e}_k\}$, and is used to select or mix environment embeddings for subsequent generation. Note that in implementation we use stepwise responsibilities: at each reverse step $t$ we compute $w_k(t) = q_\phi(\mathbf{e}_k \mid \mathbf{x}_t, \mathbf{c})$ from $(\mathbf{x}_t, \mathbf{c})$ and use it to guide sampling. As $t \to 0$ we have $\mathbf{x}_t \to \mathbf{x}_0$, hence $w_k(t)$ reduces to the responsibility based on $(\mathbf{x}_0, \mathbf{c})$.

Building on this inference step, we employ a **Denoiser** $\varepsilon_\theta(\cdot)$ to realize the actual generative process. This network is parameterized by $\theta$ and takes as input the noisy sequence, time step, and (optionally) conditioning information, and outputs noise predictions that guide the reverse diffusion trajectory. Specifically, it produces two forms of predictions:

$$\begin{aligned} (\mathbf{x}_t, t, \mathbf{c}, \mathbf{e}_k) &\xrightarrow{\varepsilon_\theta(\cdot)} \boldsymbol{\epsilon}^{\text{env}} \in \mathbb{R}^{T \times D} \\ (\mathbf{x}_t, t) &\xrightarrow{\varepsilon_\theta(\cdot)} \boldsymbol{\epsilon}^{\text{base}} \in \mathbb{R}^{T \times D} \end{aligned} \tag{35}$$

Here, $\boldsymbol{\epsilon}^{\text{env}}$ denotes the environment-conditioned noise prediction that incorporates both $\mathbf{c}$ and $\mathbf{e}_k$, while $\boldsymbol{\epsilon}^{\text{base}}$ denotes the unconditional baseline prediction without any conditioning.

Together, EnvInfer $q_\phi$ and the denoising network $\varepsilon_\theta$ form the core pipeline (Figure 3): the former identifies the relevant latent environment, and the latter performs denoising under this environment to realize causal generation. The details of the architecture of these network will be introduced in the following section (Appendix C.4).

With the environment inference module and the denoising network, we can now instantiate the interventional and counterfactual score functions, i.e., $s_t^{\text{int}}$ and $s_t^{\text{cf}}$ in practice. Although Eq. 32 and Eq. 33 are derived for different causal queries, their discrete forms share the same computational structure. Specifically, at each timestep $t$, both can be approximated by:

$$s_t^{\text{int}} \text{ or } s_t^{\text{cf}} \approx (1 + \omega) \sum_{k=1}^{K} w_k\, \boldsymbol{\epsilon}^{\text{env}} - \omega\, \boldsymbol{\epsilon}^{\text{base}}, \tag{36}$$

where the input pair $(\mathbf{x}_t, \mathbf{c})$ corresponds to interventional guidance and $(\mathbf{x}_t', \mathbf{c}')$ corresponds to counterfactual guidance.

## C.4 MODULE ARCHITECTURE

Based on above discussion, we need one denosing network $\varepsilon_\theta$ and one EnvInfer module $q_\phi$, will detailed below.

**EnvInfer $q_\phi$.** (Figure 8) Take $\mathbf{x} \in \mathbb{R}^{N \times T \times D}$, $\mathbf{c} \in \mathbb{R}^{N \times T \times D_c}$ and $\mathbf{E} = \{\mathbf{e}_1, \dots, \mathbf{e}_K\} \in \mathbb{R}^{K \times H}$ (a set of learnable embedding) as input, EnvInfer $q_\phi$ produces (1) a latent environment embedding $\mathbf{h} \in \mathbb{R}^{N \times D_h}$ and (2) a probability $\mathbf{w} = \{w_1, \dots, w_K\} \in \mathbb{R}^K$ over the environment bank $\mathbf{E}$. First, a Temporal Convolutional Network (TCN) (Bai et al., 2018) as encoder maps the concatenation $[\mathbf{x}, \mathbf{c}]$ to $\mathbf{h}' \in \mathbb{R}^{N \times T \times H}$. Then, we extract three types of features from $\mathbf{h}'$:

- **Temporal statistics:** $\boldsymbol{\mu} = \text{mean}_t(\mathbf{h}')$, $\boldsymbol{\sigma} = \text{std}_t(\mathbf{h}')$, $\boldsymbol{m} = \text{max}_t(\mathbf{h}')$, concatenate them we have $\mathbf{h}_{\text{stat}} = [\boldsymbol{\mu}, \boldsymbol{\sigma}, \boldsymbol{m}] \in \mathbb{R}^{N \times 3H}$.

- **Attention pooling:** a linear score on channel vectors yields weights $\mathbf{a} = \mathrm{softmax}_t(\mathrm{Linear}(\mathbf{h}'^{\top})) \in \mathbb{R}^{N \times T \times 1}$; the attention feature is $\mathbf{h}_{\mathrm{att}} = (\mathbf{h}' \odot \mathbf{a}^{\top}) \mathrm{sum}_t \in \mathbb{R}^{N \times H}$.

- **Spectral analysis:** power spectrum $\mathrm{psd} = |\mathrm{rFFT}_t(\mathbf{h}')|^2 \in \mathbb{R}^{N \times H \times T_f}$ with $T_f = \lfloor T/2 \rfloor + 1$; spectral centroid per channel $\mathbf{c}_{\mathrm{spec}} = \frac{(\mathrm{psd} \cdot f) \, \mathrm{sum}_t}{\mathrm{psd} \, \mathrm{sum}_t} \in \mathbb{R}^{N \times H}$, and top-$K_p$ global peaks from psd averaged over channels, $\mathbf{p}_{\mathrm{top}} \in \mathbb{R}^{N \times K_p}$. Concatenate and we have $\mathbf{h}_{\mathrm{spec}} = [\mathbf{c}_{\mathrm{spec}}, \mathbf{p}_{\mathrm{top}}] \in \mathbb{R}^{N \times (H + K_p)}$.

Concatenating all representations, we have $\mathbf{h}'' = [\mathbf{h}_{\mathrm{stat}}, \mathbf{h}_{\mathrm{att}}, \mathbf{h}_{\mathrm{spec}}] \in \mathbb{R}^{N \times (5H + K_p)}$. We then project this representation of features to latent and normalization. A LayerNorm and MLP projects features $\mathbf{h} = \tanh(\mathrm{MLP}(\mathbf{h}'')) \in \mathbb{R}^{N \times H}$. We apply row-wise $\ell_2$-normalization to $\mathbf{h}$ and $\mathbf{E}$, yielding unit-norm vectors for bank matching. Next, temperature-scaled dot-product scores and softmax produce the environment probabilities:

$$\mathbf{s} = \frac{\mathbf{h}\mathbf{E}^{\top}}{\tau} \in \mathbb{R}^{N \times K}, \qquad \mathbf{w} = \mathrm{softmax}(\mathbf{s}) \in \mathbb{R}^{N \times K},$$

where $\mathbf{w} = \{w_1, \ldots, w_K\}$ and $\sum_{k=1}^{K} w_k = 1$. In training, these scores $\mathbf{s}$ can be combined with a balanced assignment (e.g., Sinkhorn) to form targets, while $\mathbf{w}$ serves as posterior weights in our guidance rules at inference, which will be detailed in the following section. Therefore, EnvInfer outputs a tuple consisting of latent representations, scores, and posterior weights $(\mathbf{h}, \mathbf{s}, \mathbf{w})$. The first two components are only used to compute the

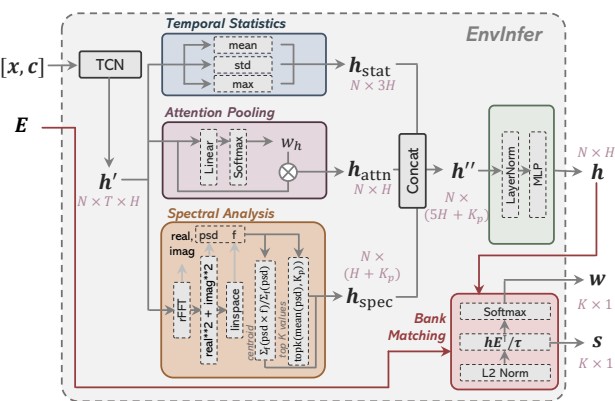

Figure 8: Architecture of **EnvInfer** $q_\phi$. The module takes input sequences $(\mathbf{x}, \mathbf{c})$ (or their augmented versions $(\mathbf{x}_v, \mathbf{c}_v)$) together with the environment bank $\mathcal{E}$, and produces latent representations $\mathbf{h}$, sample weights $\mathbf{w}$, and logits $\mathbf{s}$.

swapped prediction loss $\mathcal{L}_{\mathrm{sw}}$ (Eq. 8) and do not participate in the generation process. In contrast, the estimated environment probability $\mathbf{w}$ is employed both for computing the noise prediction loss $\mathcal{L}_{\mathrm{eps}}$ (Eq. 7) during training and for backdoor-adjusted guidance in generation (Eq. 28).

**Denoiser $\varepsilon_\theta$.** The denoiser $\varepsilon_\theta$ in `CaTSG` is implemented as a conventional U-Net architecture for 1D time series. It consists of a downsampling encoder path, a bottleneck module, and an upsampling decoder path with skip connections. This standard design has proven effective in diffusion-based generative models, and we adopt it directly as the backbone of our framework.

# D  MORE DETAILS ON TRAINING AND INFERENCE

We now describe how `CaTSG` is trained and how it is used during inference. Specifically, we first detail the training objective and optimization strategy for learning EnvInfer module $q_\phi$, the environment bank $E$, and the denoising network $\varepsilon_\theta$. We then explain the inference procedures for interventional and counterfactual generation with the equation we have derived previously. The pseudo-code of the training, interventional sampling, and counterfactual sampling procedures is summarized in Algorithm 1, 2 and 3, respectively.

## D.1  TRAINING OBJECTIVE & OPTIMIZATION

We optimize three components jointly: (1) the denoising network $\varepsilon_\theta$, (2) the environment inference network (i.e., EnvInfer $q_\phi$), (3) the learnable environment bank $E$.

**Training Denoiser.** For the backbone of the proposed framework. i.e., denoising network $\varepsilon_\theta$, we train it via $\mathcal{L}_{\mathrm{eps}} = \mathbb{E}_{\mathbf{x}_0, t, \boldsymbol{\epsilon}, \tilde{\mathbf{d}}} \left[ \left\| \boldsymbol{\epsilon} - \sum_{k=1}^{K} w_k \varepsilon_\theta(\mathbf{x}_t, t, \tilde{\mathbf{d}}_k) \right\|_2^2 \right]$ with $\tilde{\mathbf{d}}_k \in \{(\mathbf{c}, \mathbf{e}_k), \varnothing\}$ as described in Eq. 7. In practice, we use its unbiased Monte-Carlo estimate by sampling $\mathbf{x}, t, \boldsymbol{\epsilon}, \tilde{\mathbf{d}}_k$ at each step (mini-batch averaging), which yields the per-step loss $\|\boldsymbol{\epsilon} - \hat{\boldsymbol{\epsilon}}\|_2^2$, where $\hat{\boldsymbol{\epsilon}} = \sum_{k=1}^{K} w_k \varepsilon_\theta(\mathbf{x}_t, t, \tilde{\mathbf{d}}_k)$.

**Training EnvInfer.** For training EnvInfer $q_\phi$, inspired by SwAV (Caron et al., 2020) we adopted a swapped prediction loss. Intuitively, the SwAV-style loss encourages agreement within the same environment and diversity across environments: two augmentations of the same sample are encouraged to produce consistent environment assignments

---

**Algorithm 1 `CaTSG` Training procedure**

**Input:** Training data $\mathcal{D} = \{(\mathbf{x}_0, \mathbf{c})\}$; #env $K$; temperature $\tau$; Sinkhorn reg. $\eta$; weights $\alpha, \beta$, drop probability $p_{\text{drop}}$
**Output:** Trained Denoiser $\varepsilon_\theta$, EnvInfer $q_\phi$, Env. Bank $\mathbf{E} = \{\mathbf{e}_1, \ldots, \mathbf{e}_K\}$
**repeat**
    Sample data $(\mathbf{x}_0, \mathbf{c}) \in \mathcal{D}$
    // For optimizing EnvInfer
    Construct two augmentations $\{(\mathbf{x}'_0, \mathbf{c}'), (\mathbf{x}''_0, \mathbf{c}'')\}$      $\triangleright$ Two-view synchronized augmentations
    $\{(\mathbf{h}^{(v)}, \mathbf{s}^{(v)}, \mathbf{w}^{(v)})\}_{v=1}^2 \leftarrow q_\phi(\{\mathbf{x}_0^{(v)}, \mathbf{c}^{(v)}\}_{v=1}^2, \mathbf{E})$      $\triangleright$ Obtain representations, logits, probabilities
    $\{\hat{\mathbf{w}}^{(v)}\}_{v=1}^2 \leftarrow$ Sinkhorn-Knopp($\{\mathbf{s}^{(v)}\}_{v=1}^2$)      $\triangleright$ Obtain balanced soft assignments
    $\mathcal{L}_{\text{sw}} \leftarrow \ell(\mathbf{h}', \hat{\mathbf{w}}'') + \ell(\mathbf{h}'', \hat{\mathbf{w}}')$ with $\ell(\cdot)$ denotes cross-entropy      $\triangleright$ Swapped prediction loss (Eq. 8)
    // For optimizing Denoiser
    Sample $t \sim \mathcal{U}(\{1, \ldots, T\})$, noise $\boldsymbol{\epsilon} \sim \mathcal{N}(0, \mathbf{I})$
    Corrupt data: $\mathbf{x}_t \leftarrow \sqrt{\bar{\alpha}_t} \mathbf{x}_0 + \sqrt{1 - \bar{\alpha}_t} \boldsymbol{\epsilon}$
    $(\mathbf{h}, \mathbf{s}, \mathbf{w}) \leftarrow q_\phi(\mathbf{x}_0, \mathbf{c}, \mathbf{E})$, where $\mathbf{w} = \{w_1, \ldots, w_K\}$      $\triangleright$ Infer environment posterior
    **for** $k = 1$ **to** $K$ **do**
        $(\mathbf{c}, \mathbf{e}_k) \leftarrow \varnothing$ with probability $p_{\text{drop}}$      $\triangleright$ Randomly drop conditions
        $\hat{\boldsymbol{\epsilon}}_k \leftarrow \varepsilon_\theta(\mathbf{x}_t, t, \tilde{\mathbf{d}}_k), \quad \tilde{\mathbf{d}}_k \in \{(\mathbf{c}, \mathbf{e}_k), \varnothing\}$      $\triangleright$ Predicting environment-aware noises
    $\mathcal{L}_{\text{eps}} \leftarrow \|\boldsymbol{\epsilon} - \sum_{k=1}^K w_k \hat{\boldsymbol{\epsilon}}_k\|_2^2$      $\triangleright$ Noise prediction loss (Eq. 7)
    // For optimizing Env. Bank
    $\mathcal{L}_{\text{orth}} \leftarrow \|\mathbf{E}^\top \mathbf{E} - \mathbf{I}\|_F^2$      $\triangleright$ Orthogonality loss
    $\mathcal{L} \leftarrow \mathcal{L}_{\text{eps}} + \alpha \mathcal{L}_{\text{sw}} + \beta \mathcal{L}_{\text{orth}}$
    Update $\theta, \phi, \mathbf{E}$ by one optimizer step on $\mathcal{L}$
**until** *converged*;

---

(capturing shared causal context), while the orthogonality regularization keeps different environment embeddings distinct. This "consistency-under-augmentation" principle aligns closely with the causal goal of discovering latent environmental factors that remain stable across perturbations. In a world, we take two augmented versions of the same input and force the embedding of one to predict the environment assignment of the other, so both agree on an environment embedding despite different temporal perturbations. Specifically, for each sample $(\mathbf{x}, \mathbf{c})$, we form $V$ synchronized augmentations $\{(\mathbf{x}_v, \mathbf{c}_v)\}_{v=1}^V$. For augmentation $v$, via EnvInfer, we obtain latent representation $\mathbf{h}_v \in \mathbb{R}^{N \times H}$, logits $\mathbf{s}_v = \frac{1}{\tau} \mathbf{E}^\top \mathbf{h}_v \in \mathbb{R}^K$ and the predicted probability $\mathbf{w}_v = \text{softmax}(\mathbf{s}_v) \in \mathbb{R}^K$, where $\mathbf{E} = \{e_1, \ldots, e_K\} \in \mathbb{R}^{H \times K}$ is the learnable environment bank and $\tau > 0$ is a temperature. Given two augmentations $u \neq v$, the loss enforces $\mathbf{h}_u$ to predict the target assignment $\hat{\mathbf{w}}_v$. So different augmentations agree on the same environment:

$$\mathcal{L}_{\text{sw}} = \frac{1}{V(V-1)} \sum_{u \neq v} \ell(\mathbf{h}_u, \hat{\mathbf{w}}_v),$$

where $\ell(\mathbf{h}_u, \hat{\mathbf{w}}_v) = -\sum_{k=1}^K \hat{w}_{v,k} \log w_{u,k}$ is a cross-entropy loss between the target assignment $\hat{\mathbf{w}}_v$ and the predicted distribution $w_k$. Following (Caron et al., 2020), the target assignments $\hat{\mathbf{w}}_v$ are computed online using the Sinkhorn-Knopp algorithm (Cuturi, 2013) to enforce balanced partitioning of samples across environment embeddings. This encourages different augmentations of the same input to agree on the same environment embeddings, while avoiding degenerate solutions where all samples collapse to a single embedding. In our practice, we set the number of augmentations with $V = 2$.

**Training Environment Bank.** The environment bank $E$ is initialized with orthogonal initialization, followed by L2 normalization along the feature dimension. In addition, to encourage diversity among environment embeddings, we regularize the learnable environment bank $\mathbf{E}$ with an orthogonality loss:

$$\mathcal{L}_{\text{orth}} = \|\mathbf{E}^\top \mathbf{E} - \mathbf{I}\|_F^2,$$

where $\mathbf{I}$ is the identity matrix. This loss penalizes correlations between different environment representative embeddings. This ensures that embedding span distinct directions in the embedding space rather than collapsing to similar representations.

The total training objective combines the denoising and environment regularization terms: $\mathcal{L} = \mathcal{L}_{\text{eps}} + \alpha \mathcal{L}_{\text{sw}} + \beta \mathcal{L}_{\text{orth}}$, where $\alpha$ and $\beta$ are balance coefficients.

**Computational considerations.** The mixture-style denoising loss in Eq. 7 requires $K$ forward passes per sample and step. When $K$ is moderate, this overhead is acceptable. In our tasks $K$ is small (see datasets), and the end-to-end cost remains modest (Appendix G.2). If needed, the cost can be reduced without changing the target: (1) unbiased Monte Carlo mixing by sampling $k \sim \mathrm{Cat}(\mathbf{w})$ and replacing the sum with a single forward pass (variance can be lowered by stratified or multi-sample averaging); (2) top-$r$ pruning by keeping the largest weights $\{w_k\}$ above a threshold and renormalizing; and (3) sparse MoE-style gating to activate only a few environment experts per step.

### D.2 Intervention Generation

We show how backdoor-adjusted guidance is used for intervention generation. Given a context $\mathbf{c}$, our goal is to generate samples from the interventional distribution $P(X \mid \mathrm{do}(C = \mathbf{c}))$. To achieve this, we combine the environment inference module $q_\phi$ with the environment bank $\mathcal{E}$ to approximate the environment posterior, and the denoiser $\varepsilon_\theta$ to predict noise. Here, $q_\phi$ produces a latent embedding from $(\mathbf{x}_t, \mathbf{c})$, which is matched against $\mathbf{E}$ to obtain posterior probabilities $\mathbf{w} = \{w_1, \ldots, w_K\}$ with $\sum_{k=1}^K w_k = 1$.

At each denoising step, the denoiser produces both an unconditional baseline prediction $\boldsymbol{\epsilon}^{\mathrm{base}} = \varepsilon_\theta(x_t, t)$ and environment-aware predictions $\boldsymbol{\epsilon}_k^{\mathrm{env}} = \varepsilon_\theta(x_t, t, c, e_k)$ for each embedding $e_k \in \mathcal{E}$. These predictions are then aggregated according to the backdoor-adjusted rule

$$\hat{\boldsymbol{\epsilon}} = (1 + \omega) \sum_{k=1}^K w_k \, \boldsymbol{\epsilon}_k^{\mathrm{env}} - \omega \, \boldsymbol{\epsilon}^{\mathrm{base}}, \tag{37}$$

which corresponds to Eq. 36. Intuitively, this adjustment interpolates between environment-specific and unconditional guidance, preventing trivial dominance of either branch. The resulting $\hat{\epsilon}$ replaces the vanilla noise prediction in the DDPM update, $\boldsymbol{\mu}_t = \frac{1}{\sqrt{\bar{\alpha}_t}}\left(\mathbf{x}_t - \frac{1 - \bar{\alpha}_t}{\sqrt{1 - \bar{\alpha}_t}} \cdot \hat{\epsilon}\right)$. The full procedure is summarized in Algorithm 2.

---

**Algorithm 2** Interventional Generation via Backdoor-adjusted Guidance

**Input:** Guidance scale $\omega$, context $\mathbf{c}$, Env. Bank $\mathbf{E} = \{\mathbf{e}_1, \ldots, \mathbf{e}_K\}$, Denoiser $\varepsilon_\theta$, EnvInfer $q_\phi$
**Output:** Intervened sample $\hat{\mathbf{x}} \sim P(\mathcal{X} \mid \mathrm{do}(\mathcal{C} = \mathbf{c}))$
Sample $\mathbf{x}_T \sim \mathcal{N}(\mathbf{0}, \boldsymbol{I})$
**for** $t = T$ **to** $1$ **do**
  $\quad \hat{\boldsymbol{\epsilon}}^{\mathrm{base}} \leftarrow \varepsilon_\theta(\mathbf{x}_t, t)$ $\qquad\qquad\qquad\qquad\qquad\qquad\qquad$ ▷ Predict unconditional noise
  $\quad (\mathbf{h}, \mathbf{w}) \leftarrow q_\phi(\mathbf{x}_t, \mathbf{c}, \mathbf{E})$, where $\mathbf{w} = \{w_1, \ldots, w_K\}$ $\qquad$ ▷ Infer environment posterior
  $\quad$ **for** $k = 1$ **to** $K$ **do**
  $\qquad \hat{\boldsymbol{\epsilon}}_k^{\mathrm{env}} \leftarrow \varepsilon_\theta(\mathbf{x}_t, t, \mathbf{c}, \mathbf{e}_k)$ $\qquad\qquad\qquad\qquad\qquad$ ▷ Predict env-aware noise
  $\quad \hat{\boldsymbol{\epsilon}} \leftarrow (1 + \omega) \sum_{k=1}^K w_k \cdot \hat{\boldsymbol{\epsilon}}_k^{\mathrm{env}} - \omega \cdot \hat{\boldsymbol{\epsilon}}^{\mathrm{base}}$ $\qquad\qquad$ ▷ Backdoor-adjusted guidance in Eq.36
  $\quad \boldsymbol{\mu}_t \leftarrow \frac{1}{\sqrt{\bar{\alpha}_t}}\left(\mathbf{x}_t - \frac{1 - \bar{\alpha}_t}{\sqrt{1 - \bar{\alpha}_t}} \cdot \hat{\boldsymbol{\epsilon}}\right)$ $\qquad\qquad\qquad$ ▷ Compute predicted mean
  $\quad$ Sample $\mathbf{x}_{t-1} \sim \mathcal{N}(\boldsymbol{\mu}_t, \sigma_t^2 \boldsymbol{I})$ $\qquad\qquad\qquad\qquad\qquad$ ▷ Sampling step
**return** $\hat{\mathbf{x}} \leftarrow \mathbf{x}_0$

---

### D.3 Counterfactual Generation

We show how AAP (Eq. 10) works in counterfactual generation. Given an observed target $\mathbf{x}_0$ with context $\mathbf{c}$, our goal is to generate a counterfactual sample corresponding to an alternative context $\mathbf{c}'$, i.e., $P(X' \mid X = \mathbf{x}_0, C = \mathbf{c}, C' = \mathbf{c}')$. To this end, we follow the AAP procedure. In the *abduction* step, the environment inference module EnvInfer $q_\phi$ infers the environment posterior $q(\mathbf{e} \mid \mathbf{x}_0, \mathbf{c})$ by matching the latent embedding of $(\mathbf{x}_0, \mathbf{c})$ against the environment bank $\mathbf{E}$. In the *action* step, we replace the original context $\mathbf{c}$ with the counterfactual context $\mathbf{c}'$. Finally, in the *prediction* step, we run the denoising diffusion process with backdoor-adjusted guidance. At each denoising step, the denoiser produces both an unconditional baseline prediction $\boldsymbol{\epsilon}^{\mathrm{base}} = \varepsilon_\theta(\mathbf{x}'_t, t)$ and environment-aware predictions $\boldsymbol{\epsilon}_k^{\mathrm{env}} = \varepsilon_\theta(\mathbf{x}'_t, t, \mathbf{c}', \mathbf{e}_k)$ for each environment embedding $\mathbf{e}_k \in \mathbf{E}$. These predictions are aggregated using the same rule as in intervention generation, which instantiates the counterfactual score in Eq. 10. The resulting $\hat{\epsilon}$ is plugged into the DDPM update to compute the mean $\boldsymbol{\mu}_t$ and sample $\mathbf{x}'_{t-1}$, leading to a counterfactual trajectory that remains faithful to the abducted environment while reflecting the counterfactual context. The full procedure is summarized in Algorithm 3.

---

**Algorithm 3** Counterfactual Generation via Abduction-Action-Prediction

---

**Input:** Guidance scale $\omega$, observed target $\mathbf{x}_0$, observed context $\mathbf{c}$, counterfactual context $\mathbf{c}'$, Env. Bank $\mathbf{E} = \{\mathbf{e}_1, \ldots, \mathbf{e}_K\}$, Denoiser $\varepsilon_\theta$, EnvInfer $q_\phi$
**Output:** Counterfactual sample $\hat{\mathbf{x}}' \sim P(\mathcal{X}' \mid \mathcal{X} = \mathbf{x}_0, \mathcal{C} = \mathbf{c}, \mathcal{C}' = \mathbf{c}')$
**Step 1 - Abduction:** $(\mathbf{h}, \mathbf{w}) \leftarrow q_\phi(\mathbf{x}_0, \mathbf{c}, \mathbf{E})$, where $\mathbf{w} = \{w_1, \ldots, w_K\}$      $\triangleright$ Infer environment posterior
**Step 2 - Action:** Replacing $\mathbf{c} \leftarrow \mathbf{c}'$      $\triangleright$ Construct new world
**Step 3 - Prediction:**
Sample $\mathbf{x}'_T \sim \mathcal{N}(\mathbf{0}, \boldsymbol{I})$
**for** $t = T$ **to** 1 **do**
    $\hat{\boldsymbol{\epsilon}}_{\text{base}} \leftarrow \varepsilon_\theta(\mathbf{x}'_t, t)$      $\triangleright$ Predict unconditional noise
    **for** $k = 1$ **to** $K$ **do**
        $\hat{\boldsymbol{\epsilon}}_k^{\text{env}} \leftarrow \varepsilon_\theta(\mathbf{x}'_t, t, \mathbf{c}', \mathbf{e}_k)$      $\triangleright$ Predict env-aware noise
    $\hat{\boldsymbol{\epsilon}} \leftarrow (1 + \omega) \sum_{k=1}^{K} w_k \cdot \hat{\boldsymbol{\epsilon}}_k^{\text{env}} - \omega \cdot \hat{\boldsymbol{\epsilon}}^{\text{base}}$      $\triangleright$ Backdoor-adjusted guidance in Eq. 36
    $\boldsymbol{\mu}_t \leftarrow \frac{1}{\sqrt{\bar{\alpha}_t}} \left( \mathbf{x}'_t - \frac{1 - \bar{\alpha}_t}{\sqrt{1 - \bar{\alpha}_t}} \cdot \hat{\boldsymbol{\epsilon}} \right)$      $\triangleright$ Compute predicted mean
    $\mathbf{x}'_{t-1} \sim \mathcal{N}(\boldsymbol{\mu}_t, \sigma_t^2 \boldsymbol{I})$      $\triangleright$ Sampling step
**return** $\hat{\mathbf{x}}' \leftarrow \mathbf{x}'_0$

---

# E MORE DETAILS ON DATASETS

## E.1 SYNTHETIC DATASET DESCRIPTION

To systematically evaluate causal time series generation under known causal mechanisms, we construct two synthetic benchmark datasets: **Harmonic-VM** and the more complex **Harmonic-VP**. These datasets simulate a class of damped mechanical oscillators (Figure 9a) governed by second-order differential equations (Eq. 38). We first describe the governing equation and then explain how the datasets are constructed.

**Governing Equation.** We consider a damped harmonic oscillator illustrated in Figure 9a, governed by the following second-order differential equation:

$$m \cdot \ddot{x}(t) + \gamma \cdot \dot{x}(t) + k \cdot x(t) = 0, \tag{38}$$

where:

- $x(t)$ denotes the position at time $t$,
- $\dot{x}(t) \equiv v(t)$ denotes the velocity at time $t$,
- $\ddot{x}(t) \equiv a(t)$ denotes the acceleration at time $t$,
- $m$ is the mass of the object,
- $\gamma$ is the damping coefficient,
- $k$ is the stiffness coefficient.

**Variable Roles.** We then introduce how we define the variable to construct the dataset under the SCM in Figure 2b based on Eq. 38.

- **Choice of $X$, $C$, and $E$.** As shown in Figure 9b (*left* part), we define the target series $X$ as the acceleration sequence $a(t)$ (or $\ddot{x}(t)$) the contextual variables $C$ as the instantaneous position $x(t)$ and velocity $v(t)$ (or $\dot{x}(t)$), and the environment-level parameters $E$ as the physical coefficients governing the dynamical system, which will be introduced below. Note that we choose $a(t)$ as $X$ because, according to the governing second-order differential equation of motion, acceleration is uniquely determined by the current position, velocity, and system parameters (Lehrman, 1998). This ensures a clear causal direction $E \to X$ and $C \to X$ without instantaneous feedback from $X$ to $C$.

- **Design of $E$.** As defined above, $E$ corresponds to the physical coefficients $(m, \gamma, k)$ in Eq. 38, which govern the system's mass, damping, and stiffness, respectively. These coefficients can either remain constant or vary over time, allowing us to represent a wide range of dynamical behaviors. For example, an increasing $m$ simulates mass growth (e.g., a spring absorbing water), a periodically varying $\gamma$ models oscillatory damping, and a decaying $k$ captures progressive stiffness loss.

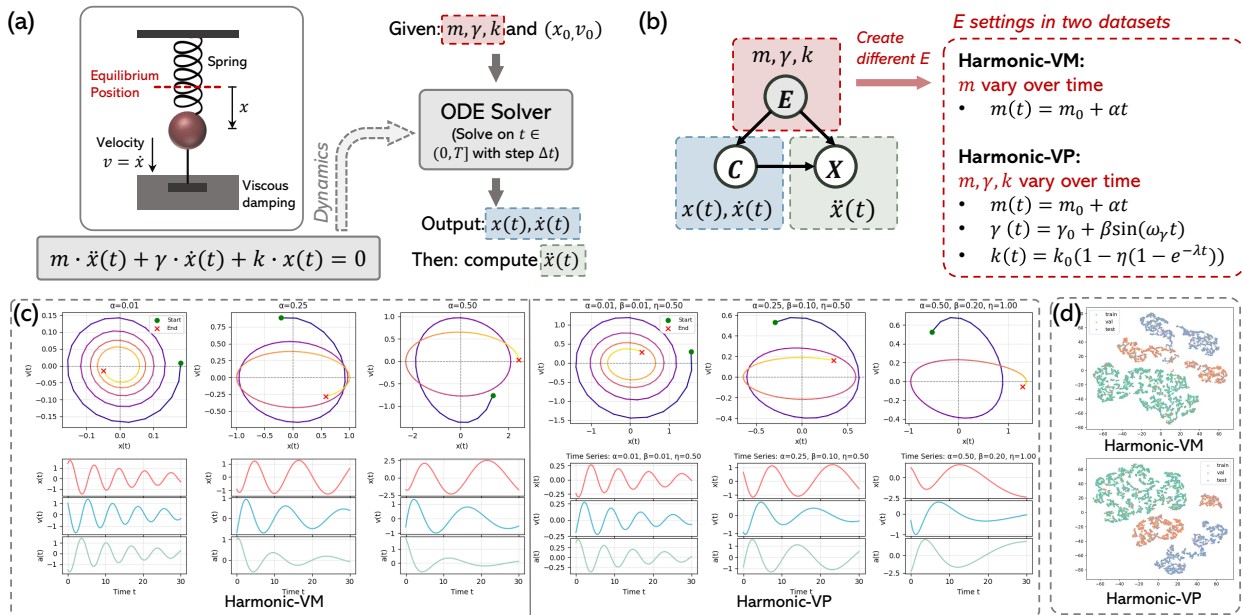

Figure 9: (a) A damped harmonic oscillator governed by $m\ddot{x}(t) + \gamma\dot{x}(t) + kx(t) = 0$, where an ODE solver, given $(m, \gamma, k)$ and initial state $(x_0, v_0)$, produces trajectories of position $x(t)$, velocity $\dot{x}(t)$, and acceleration $\ddot{x}(t)$. (b) Mapping the variables in this physical system to our SCM (Figure 2b), where the latent environment $E$ specifies physical parameters $(m, \gamma, k)$. By varying $E$, we create two datasets: **Harmonic-VM**, with mass $m$ varying over time, and **Harmonic-VP**, with mass, damping, and stiffness all varying over time. (c) Example trajectories under different parameter settings, shown in phase space (*top*) and as time series of position, velocity, and acceleration (*bottom*). (d) t-SNE visualization showing that different environments yield distinct clusters in the data distribution.

**Dataset Design.** We consider two different environment diversity level to create two dataset, shown in Figure 9b (*right* part). For both scenarios, we simulate environment diversity by sampling coefficients from non-overlapping intervals for training, validation, and testing, detailed as follows.

- **Harmonic-VM** (short for *Harmonic Oscillator with Variable Mass*): the mass evolves over time according to $m(t) = m_0 + \alpha\, t$, where $m_0 = 1.0$ is the initial mass. The damping coefficient $\gamma$ and the stiffness $k$ remain constant. This scenario is analogous to a sponge suspended from a spring gradually absorbing water, resulting in an increasing mass over time.

  We assign *dominant, non-overlapping* intervals of $\alpha$ to each split, detailed in Table 5. For each sample, $\alpha$ is drawn from an *80–20 mixture*: with probability $0.8$ from the split's dominant interval and with probability $0.2$ from the union of the other splits' intervals.

  - **Train range** (slow mass growth): represents relatively slow mass growth, where the system remains close to its original oscillation characteristics.
  - **Validation range** (moderate mass growth): moderate mass increase that noticeably reduces oscillation frequency over time.
  - **Test range** (rapid mass growth): rapid mass growth leading to substantial reduction in oscillation frequency and stronger damping effects.

- **Harmonic-VP** (short for *Harmonic Oscillator with Variable Parameters*): all three physical parameters $(m(t), \gamma(t)$, and $k(t))$ are allowed to evolve over time according to smooth nonlinear functions:

$$m(t) = m_0 + \alpha\, t, \quad \gamma(t) = \gamma_0 + \beta\, \sin(\omega_\gamma t), \quad k(t) = k_0\,(1 - \eta\,(1 - e^{-\lambda t})). \tag{39}$$

  where $m_0 = 1.0$ is the initial mass, $\gamma_0 = 0.1$ is the damping coefficient, $\omega_\gamma = 0.2$ controls the damping oscillation frequency, $k_0 = 1.0$ is the stiffness, and $\lambda = 0.05$ governs the stiffness decay rate.

  This setting reflects more complex environments in which the mass grows at a constant rate (controlled by $\alpha$), the damping exhibits periodic modulation (amplitude $\beta$ and frequency $\omega_\gamma$), and the stiffness undergoes an exponential decay (rate $\lambda$).

  We assign *dominant, non-overlapping* parameter intervals for $(\alpha, \beta, \eta)$ to train/validation/test to create clearly distinguishable oscillatory regimes (see Table 5). For each split and each parameter, samples are drawn from an 80–20

mixture: with probability $0.8$ from the split's dominant interval and with probability $0.2$ from the union of the other splits' intervals (tails). This mild cross-split mixing preserves regime separability while providing limited overlap to test robustness under small distribution shift.

- **Train range** (*light damping regime*): low $(\alpha, \beta, \eta)$ values correspond to slow mass saturation, gentle periodic damping modulation, and gradual stiffness decay. This regime resembles a lightly damped system where the mass increases slowly, damping changes are minimal, and stiffness remains nearly constant over long durations, resulting in sustained oscillations with only mild amplitude reduction.
- **Validation range** (*moderate damping regime*): moderate $(\alpha, \beta, \eta)$ values lead to faster mass growth, more pronounced damping oscillations, and noticeable stiffness weakening over time. The system exhibits a balance between energy retention and dissipation, with observable but not abrupt changes in both oscillation amplitude and frequency.
- **Test range** (*strong damping regime*): high $(\alpha, \beta, \eta)$ values produce rapid mass saturation, strong periodic damping fluctuations, and accelerated stiffness decay. This regime mimics a harsh environment where the system quickly loses stiffness and experiences significant damping variation, causing oscillations to decay rapidly and frequencies to shift noticeably from those in the training regime.

By explicitly parameterizing environment-level factors (e.g., $\alpha$, $\beta$, $\eta$), our synthetic datasets support counterfactual queries. Given an observed trajectory under one environment configuration, we can alter selected parameters while keeping others fixed, enabling controlled simulation of alternative outcomes. This property is crucial for evaluating counterfactual time series generation models in a setting with known ground truth.

**Simulation and Construction Process.** We construct the dataset in two stages:

**Step 1: Factual trajectory simulation.**

- *Parameter sampling.* Sample the environment-level factors from predefined ranges: in Harmonic-VM each environment is associated with a unique $\alpha$, and in Harmonic-VP with a triplet $(\alpha, \beta, \eta)$.
- *Initial-state sampling.* For each trajectory, sample the initial position $x_0$ and velocity $v_0$ from designated factual ranges $(x_{\min}, x_{\max})$ and $(v_{\min}, v_{\max})$.
- *Numerical integration.* With the initial state $(x_0, v_0)$, solve the ODE in Eq. 38 to obtain:

$$x = [x_0, \ldots, x_T] \in \mathbb{R}^T, \quad v = [v_0, \ldots, v_T] \in \mathbb{R}^T, \quad a = [a_0, \ldots, a_T] \in \mathbb{R}^T.$$

The position sequence $x$ is taken as the target time series $\mathbf{x}$, and the contextual sequence is defined as $\mathbf{c} = [\mathbf{c}_1, \ldots, \mathbf{c}_T] \in \mathbb{R}^{T \times 2}$, where $\mathbf{c}_t = (v_t, a_t)$. Example trajectories under different parameter settings are shown in Figure 9c.

**Step 2: Counterfactual trajectory construction.**

- *Intervention on initial state.* Given an observed trajectory $(\mathbf{x}, \mathbf{c})$ simulated under environment parameter $\alpha$ or $(\alpha, \beta, \eta)$, keep these parameters fixed and replace the initial state $(x_0, v_0)$ with an alternative $(x_0', v_0')$. Here, $x_0'$ and $v_0'$) are sampled from a separate, non-overlapping range $(x_{\min}', x_{\max}')$ and $(v_{\min}', v_{\max}')$.
- *Re-simulation.* Re-solve Eq. 38 under the same environment parameters but with the modified initial state to obtain the counterfactual trajectory $(\mathbf{x}', \mathbf{c}')$.
- *Pairing.* Store the factual–counterfactual pair: $\big[(\mathbf{x}, \mathbf{c}), (\mathbf{x}', \mathbf{c}')\big]$.

In our implementation, the factual initial states are sampled from $x_0 \in [-2.0, 2.0]$ and $v_0 \in [-1.5, 1.5]$ (expanded for diversity), while counterfactual initial states are drawn from non-overlapping ranges $x_0' \in [2.2, 4.0]$ and $v_0' \in [-2.5, -1.0]$.

**Discussion.** In our dataset construction, the contextual variables $C$ are represented by the instantaneous position $x(t)$ and velocity $v(t)$ derived from the simulated trajectories. Although in many real-world conditional time series generation tasks $C$ corresponds to exogenous variables (e.g., weather, holiday), here $C$ is an intermediate state variable determined jointly by the environment-level parameters $E$ and the system's dynamics. The target series $X$ is defined as the acceleration sequence $a(t)$, which, according to the governing second-order differential equation, is uniquely determined by the current $C$ and $E$. This design ensures a clear causal direction $E, C \to X$ without instantaneous feedback from $X$ to $C$, thereby creating a controlled setting where $E$ acts as a latent confounder influencing both $C$ and $X$. To further illustrate this effect, Figure 9d shows t-SNE visualizations of two datasets, where data from

Table 5: Summary of datasets and splitting strategies. Synthetic datasets are constructed from physical simulations with controlled parameter variations, while real-world datasets are split based on selected environmental factors. #Samples: number of samples in Train/Val/Test splits.

| Type | Dataset | Variables | Split Strategy | #Samples |
|---|---|---|---|---|
| *Synthetic* | **Harmonic-VM** | Target ($x$): Acceleration
Context ($c$): Velocity, Position | $\alpha$-based:
Train [0.0, 0.2]; Val [0.3, 0.5]; Test [0.6, 1.0] | 3,000/ 1,000/ 1,000 |
| | **Harmonic-VP** | Target ($x$): Acceleration
Context ($c$): Velocity, Position | Combination-based:
Train:$\alpha \in [0.0, 0.2]$, $\beta \in [0.0, 0.01]$, $\eta \in [0.002, 0.08]$
Val: $\alpha \in [0.3, 0.5]$, $\beta \in [0.018, 0.022]$, $\eta \in [0.18, 0.22]$
Test: $\alpha \in [0.6, 1.0]$, $\beta \in [0.035, 0.04]$, $\eta \in [0.42, 0.5]$ | 3,000/ 1,000/ 1,000 |
| *Real-world* | **Air Quality** | Target ($x$): $PM_{2.5}$
Context ($c$): TEMP, PRES, DEWP, WSPM, RAIN, wd | Station-based:
Train (Dongsi, Guanyuan, Tiantan, Wanshouxigong, Aotizhongxin, Nongzhanguan, Wanliu, Gucheng);
Val (Changping, Dingling); Test (Shunyi, Huairou) | 11,664/ 2,916/ 2,916 |
| | **Traffic** | Target ($x$): traffic_volume
Context ($c$): rain_1h, snow_1h, clouds_all, weather_main, holiday | Temperature-based:
Train (<12°C); Val ([12,22]°C); Test (>22°C) | 26,477/ 16,054/ 5,578 |

different environments form clearly separated clusters, confirming that the constructed environments induce distinct joint distributions of $C$ and $X$. Such a setup faithfully instantiates the causal graph in Figure 2b and offers the following advantages:

- **Clean causal structure.** By defining $X$ as acceleration, the instantaneous feedback loop from $X$ to $C$ is eliminated, simplifying the SCM and aligning it with physical laws.

- **Controlled confounding.** The environment-level parameters $E$ influence both $C$ and $X$, enabling systematic evaluation of methods under known and controllable confounding.

- **Counterfactual tractability.** Because $X$ is fully determined by $C$ and $E$, counterfactual trajectories can be generated by intervening on either $C$ or $E$ with clear physical interpretation.

- **Physical interpretability.** The mapping from $(E, C)$ to $X$ directly follows from the second-order differential equation, ensuring that generated data adhere to physically consistent dynamics.

E.2  REAL WORLD DATASET DESCRIPTION

We further evaluate our framework on two real-word datasets, i.e., **Air Quality** and **Traffic**. Across datasets, we construct *environment splits* by using selected observable variables as proxies to delineate latent contexts (i.e., treated as unobserved environments). These variables are used *only* for splitting Train/Val/Test and and the environment bank remains latent and learnable during training and inference.

**Air Quality**  dataset (Chen, 2019) is a comprehensive environmental monitoring dataset containing four years of hourly air quality and meteorological measurements from 12 monitoring stations across Beijing, China. The time period is from March 1st, 2013 to February 28th, 2017. We set the target variable as $PM_{2.5}$ concentration, while contextual variables include temperature (TEMP), pressure (PRES), dew point (DEWP), wind speed (WSPM), rainfall (RAIN), and wind direction (wd). We use station clusters (central vs. peripheral, derived from station coordinates) as observable proxies for latent environments to define the Train/Val/Test splits. Details are provided in Table 5, and the geographic distribution of selected stations is shown in Figure 10.

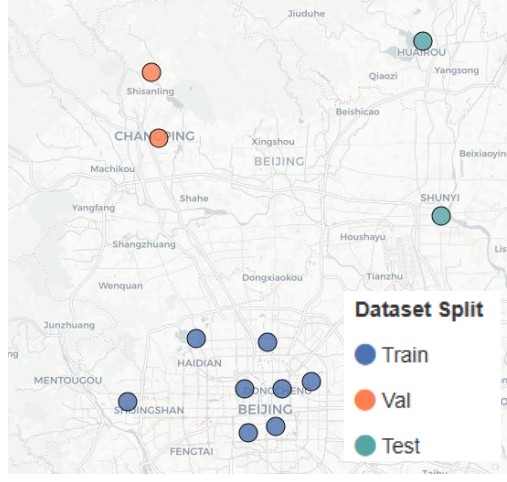

Figure 10: Geographic distribution of stations by split in the Air Quality dataset.

**Traffic**  dataset (Hogue, 2019) is an hourly traffic dataset collected on the westbound I-94 Interstate Highway between Minneapolis and St. Paul, MN, USA. It contains 48,204 instances with traffic volume as the prediction target, and contextual variables including temperature, rainfall (rain_1h), snowfall

1404 (snow_1h), cloud coverage (clouds_all), main weather type (weather_main), and holiday indicators. we use tempera-
1405 ture regime as proxies of latent environments to construct Train/Val/Test splits (details in Table 5).

### E.3 EMPIRICAL EVIDENCE OF SPURIOUS CORRELATIONS

To diagnose spurious correlations, we examine the stability of the conditional law $P(X \mid C)$ across environment regimes. We treat the dataset splits $\{\text{train}, \text{val}, \text{test}\}$ as proxies for environments; the splitting method is reported above and summarized in Table 5. We proceed in two steps:

- **Local conditional expectation (LOWESS).** For each contextual variable $C_j$ and each split $s \in \{\text{train}, \text{val}, \text{test}\}$, we fit a univariate locally weighted regression of $X$ on $C_j$ using only data from $s$. This provides an estimate of the conditional mean $\mu_s^{(j)}(c) \equiv \mathbb{E}_s[X \mid C_j = c]$, where all other components of $C$ are implicitly averaged out under their empirical distribution within $s$. We then plot the three fitted curves $\{\mu_s^{(j)}(c)\}$ to assess whether the $X$–$C_j$ relationship is stable across environments.

- **Binned conditional distribution.** For each $C_j$, we first define bin edges on the training split. Within each bin and each split we compute the conditional mean (and variance) of $X$, and compare these per bin estimates across splits. For each bin $b$ and split $s$, we compute the sample mean $\mu_s^{(j)}(b) \equiv \mathbb{E}_s[X \mid C_j \in b]$ (bar height) and its standard error (error bar). Thus, each bar represents the average target value for samples whose $C_j$ falls in that interval. Large differences across splits for the same bin indicate potentially spurious correlation.

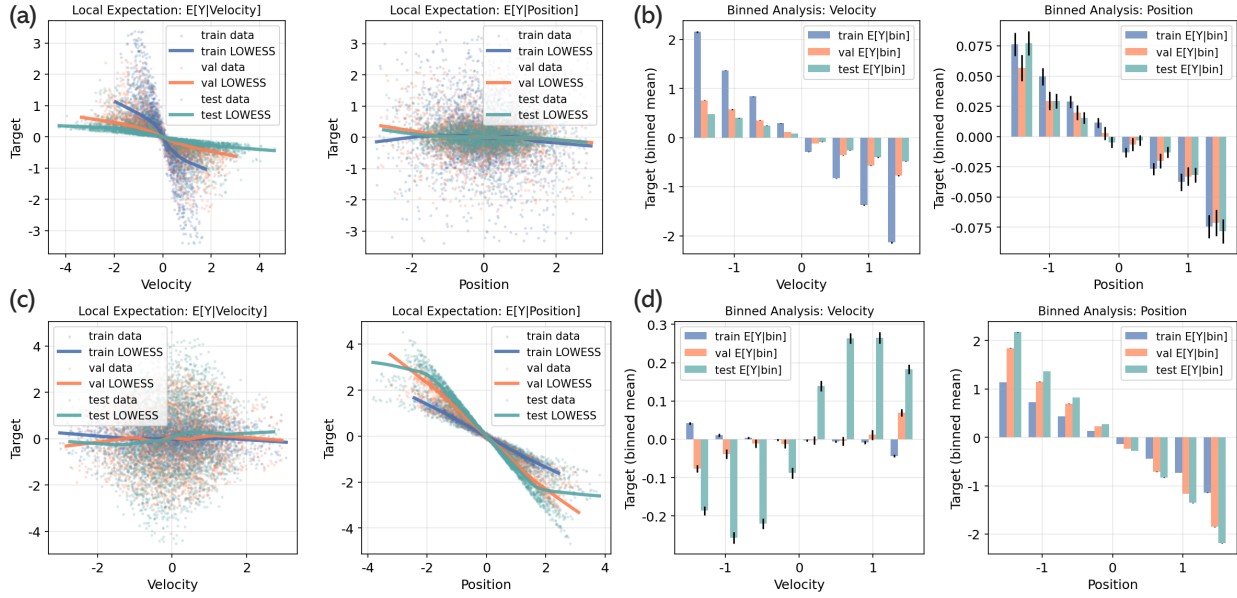

Figure 11: Empirical diagnostics on synthetic datasets. (a,b) Harmonic-VM and (c,d) Harmonic-VP with $C \in \{\text{position, velocity}\}$. (a,c) Local conditional expectation $E[X \mid C]$ with LOWESS for train, validation, and test. (b,d) Binned conditional means with standard-error bars for the same splits.

**Synthetic datasets.** Figure 11 contrasts Harmonic-VM *(a,b)* and Harmonic-VP *(c,d)*. Under our SCM, $X = a(t) = -(\gamma(t)/m(t))\,v(t) - \big(k(t)/m(t)\big)\,x(t)$, hence the conditional relation with velocity is governed by $\gamma/m$ while the relation with position is dominated by $k/m$. In Harmonic-VM, the LOWESS curves for $E[X \mid v]$ *(a, left)* separate by split, whereas $E[X \mid x]$ *(a, right)* remains close to a shared linear trend. The binned means *(b)* confirm this: the velocity–target association changes sign and magnitude across bins and splits, while the position–target association is comparatively stable. In Harmonic-VP the divergence is amplified *(c,d)* because both $\gamma(t)$ and $k(t)$ vary, leading to stronger environment dependence in $E[X \mid v]$ and mild level or slope shifts in $E[X \mid x]$.

**Real-world datasets.** For the Air Quality dataset (Figure 12a and 12b), the conditional relation with temperature shows moderate between–split shifts, and wind direction exhibits pronounced nonmonotonic drift, both evident in the LOWESS curves and confirmed by the binned means. For the Traffic dataset (Figure 12c and 12d), the association with cloud coverage shows mild shifts, whereas the relation with hourly rainfall is more strongly nonstationary across splits. Note that the missing bins in Figure 12d arise because rain_1h is zero–inflated with a heavy–tailed positive

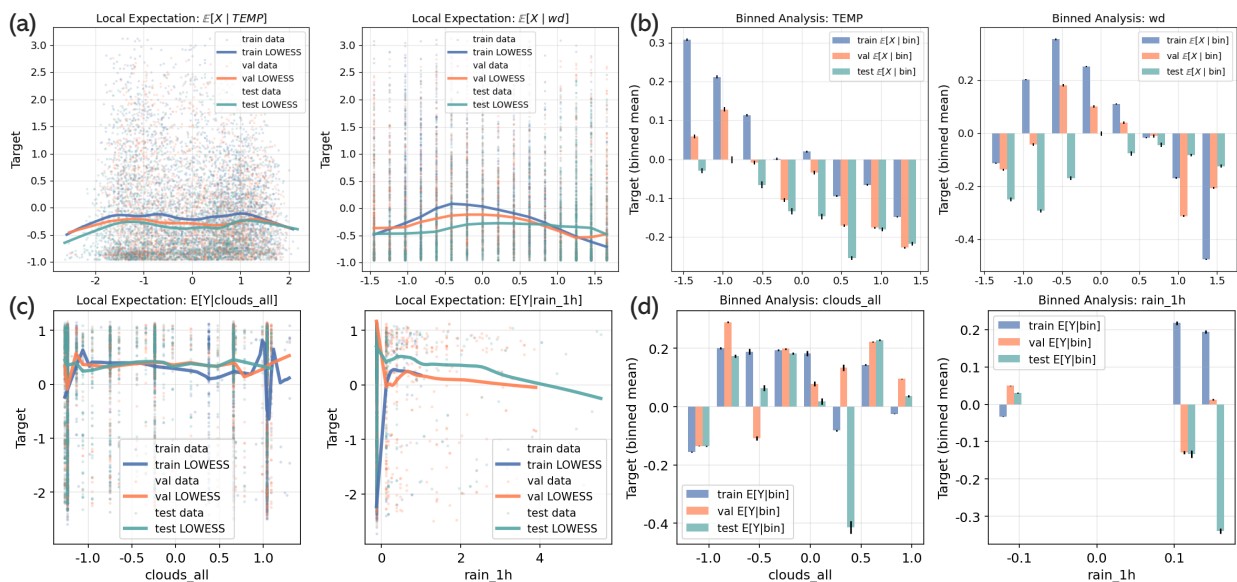

Figure 12: Empirical diagnostics on real-world datasets. (a) and (b) Air Quality with $C \in \{\text{TEMP}, \text{wd}\}$, (c) and (d) Traffic with $C \in \{\text{clouds\_all}, \text{rain\_1h}\}$. (a,c) Local conditional expectation $E[X \mid C]$ with LOWESS for train, validation, and test. (b,d) Binned conditional means with standard-error bars for the same splits.

part; under equal–width binning many mid–range intervals contain no observations and are omitted, producing the visible gap.

## F  MORE DETAILS ON EXPERIMENTS

### F.1  BASELINES

We compare our method against five representative baselines for conditional time series generation:

- **TimeGAN** (Yoon et al., 2019): A generative adversarial framework tailored for time series with embedding and recovery modules. Its architecture consists of an *embedder*, *recovery*, *generator*, and *discriminator*, enabling both autoencoding and adversarial learning. The training follows three stages: autoencoder pretraining, supervised alignment of latent trajectories, and joint adversarial optimization. We use hidden dimension 64, noise dimension 32, 3 layers, and train with learning rate $10^{-4}$.

- **WaveGAN** (Donahue et al., 2019): A convolutional GAN originally designed for audio, adapted for time series via 1D convolutions. The generator is composed of stacked 1D transposed convolutional layers (512, 256, 128, output channels), while the discriminator mirrors this with strided convolutions and global pooling. Conditional information is injected through a shared ConditionalEncoder. We set hidden dimension 64, 4 convolutional layers, and train with learning rate $10^{-4}$.

- **Pulse2Pulse** (Thambawita et al., 2021): A conditional UNet-based GAN for time series. The generator is a 1D UNet with four downsampling blocks, a bottleneck, and four upsampling blocks with skip connections, while the discriminator is a CNN with global pooling. Training combines adversarial loss with an $L_1$ reconstruction loss to enforce fidelity. We set hidden dimensions 64, $\lambda_{L1} = 10$, and learning rate $10^{-4}$.

- **TimeWeaver-CSDI** (Narasimhan et al., 2024): A score-based diffusion model adapted for conditional time series generation. It employs a 1D UNet backbone with residual blocks (64–512 hidden channels) for noise prediction, conditioned on processed context embeddings. Training follows DDPM with $T = 1000$ diffusion steps, and sampling adopts DDIM for acceleration. We use hidden dimensions 64 and learning rate $10^{-4}$.

- **TimeWeaver-SSSD** (Narasimhan et al., 2024): A diffusion-based model that replaces the UNet with a structured state space (S4) backbone to capture long-range dependencies. The architecture stacks four S4 layers with state dimension 64 and dropout 0.1, conditioned via the ConditionalEncoder. Training follows score matching under continuous-time diffusion. We set $d_{\text{model}} = 64$, $d_{\text{state}} = 64$, 4 S4 layers, and learning rate $10^{-4}$.

## F.2 EVALUATION METRICS

**Marginal Distribution Distance (MDD).** This metric measures the discrepancy between the marginal distributions of generated and real sequences. Concretely, we compute the distance between empirical marginals $p_{\text{gen}}(x)$ and $p_{\text{real}}(x)$ over each variable, then average across variables to assess distributional fidelity.

**Kullback–Leibler divergence (KL).** The KL divergence quantifies how much the generated distribution $p_{\text{gen}}$ diverges from the real distribution $p_{\text{real}}$, defined as

$$\text{KL}(p_{\text{real}} \,\|\, p_{\text{gen}}) = \sum_x p_{\text{real}}(x) \log \frac{p_{\text{real}}(x)}{p_{\text{gen}}(x)}.$$

Lower KL values indicate closer alignment between generated and real samples.

**Maximum Mean Discrepancy (MMD).** MMD is a kernel-based distance that compares two distributions by embedding them into a reproducing kernel Hilbert space (RKHS). It is defined as the squared distance between mean embeddings:

$$\text{MMD}^2(p_{\text{real}}, p_{\text{gen}}) = \left\| \mathbb{E}_{x \sim p_{\text{real}}}[\,\phi(x)\,] - \mathbb{E}_{x \sim p_{\text{gen}}}[\,\phi(x)\,] \right\|^2,$$

where $\phi(\cdot)$ is the kernel feature map. Smaller MMD indicates greater similarity between the two distributions.

**Joint Frechet Time Series Distance (J-FTSD).** To evaluate the quality of generated time series and metadata jointly, we adopt the Joint Frechet Time Series Distance (J-FTSD) proposed by Narasimhan et al. (2024). J-FTSD compares the joint distributions of real and generated samples in a learned embedding space. Specifically, the time series $x_i$ and the conditional data $c_i$ are mapped into embeddings via $\phi_{\text{time}}(\cdot)$ and $\phi_{\text{meta}}(\cdot)$, which are then concatenated to form a joint representation $z_i$. Let $\mu_d$ and $\Sigma_d$ ($d \in \{r, g\}$) denote the mean and covariance of the embeddings for real and generated data respectively. J-FTSD is defined as the Fréchet distance between the two Gaussian approximations:

$$\text{J-FTSD}(D_g, D_r) = \|\mu_r - \mu_g\|^2 + \text{Tr}\left( \Sigma_r + \Sigma_g - 2(\Sigma_r \Sigma_g)^{\frac{1}{2}} \right). \tag{40}$$

This metric thus quantifies the similarity between real and generated joint distributions of time series and metadata in the embedding space.

*Note:* In practice, we observed instability during training when learning the joint embedding space. To address this, we applied $\ell_2$ normalization to the embedding vectors during training, which stabilizes the contrastive learning objective. At inference time, however, we compute the final J-FTSD using the raw (unnormalized) embeddings. This design choice is motivated by two considerations: (1) normalization during training improves optimization stability, while (2) avoiding normalization at inference preserves the original scale of embeddings, which is crucial for reliable Fréchet-based distance estimation.

## F.3 IMPLEMENTATION DETAIL

**Selection of $K$.** In the proposed method, there is a hyperparameter $K$, which is the number of latent environments. For both synthetic and real-world datasets, we define "environments" splits by using selected variables to delineate latent contexts that differ across splits (see Table 5 for details). These variables are used only to define the splits; the environments themselves remain unobserved (latent). Because the environment bank is latent, we do not tie the number of environments to the number of split rules (e.g., fix $K = 3$). Instead, we treat $K$ as a dataset-specific hyperparameter. This flexibility allows the model to represent more complex scenarios (e.g., submodes within a split) and to capture richer latent environment information. We also conducted the experiments under different $K$ in Section 5.5. Since both $K$ and $H$ control the capacity of the representation of $E$, if multiple unobserved confounders are expected, we recommend increasing both $K$ and $H$ to allow the environment bank to capture combinatorial variation.

**Preparation of c.** To ensure a consistent **c** representation and handle different types contextual variables especially for real world dataset, we preprocess **c** as follows. Categorical attributes (e.g., weather type) are embedded into dense vectors using learnable embeddings. We then augment the sequence with a time-of-day phase encoding by adding sinusoidal features $\sin(2\pi t/T)$ and $\cos(2\pi t/T)$ for each time step $t$, which capture periodic temporal patterns. Finally, the resulting tensor concatenated with $\mathbf{e}_k$ for input to the denoising network.

**Dataset-specific settings of `CaTSG`.** We evaluate `CaTSG` on four datasets (Harmonic-VM, Harmonic-VP, Air Quality and Traffic). For all these datasets, we set $p_{\text{drop}} = 20\%$ for training, and we use the DPM-Solver sampler (Lu et al., 2022) for generation, enabling the single-step variant with second order. For evaluation, we repeat sampling five times and fix the random seeds to {42, 123, 456, 789, 999}. Harmonic-VM and Harmonic-VP are synthetic datasets and we use the same setting. Below we list the dataset–specific settings.

- **Harmonic-VM & Harmonic-VP.** For both synthetic datasets, we set the batch size to 512, the hidden dimension to 64, and the base learning rate to $1 \times 10^{-3}$. We set the number of environments $K$ to 4. We warm up the model for 200 optimization steps, during which we optimize the swapped-prediction and orthogonality terms, i.e., $\mathcal{L}_{\text{sw}}$ and $\mathcal{L}_{\text{orth}}$; in the normal phase we optimize the swapped, MSE, and orthogonality losses, i.e., $\mathcal{L}_{\text{sw}}$, $\mathcal{L}_{\text{orth}}$ and $\mathcal{L}_{\text{eps}}$ .

- **Air Quality.** For the air quality dataset, we set the batch size to 256, the hidden dimension to 32, and the base learning rate to $1 \times 10^{-3}$. We set the number of environments $K$ to 8. We use a short warm-up of 50 steps and, in the normal training phase, we optimize the MSE and orthogonality losses, i.e., $\mathcal{L}_{\text{orth}}$ and $\mathcal{L}_{\text{eps}}$ .

- **Traffic.** For the traffic dataset, we set the batch size to 256, the hidden dimension to 32, and the base learning rate to $1 \times 10^{-3}$. We set the number of environments $K$ to 4. We use a 50-step warm-up and optimize the MSE and orthogonality losses during the normal phase, i.e., $\mathcal{L}_{\text{orth}}$ and $\mathcal{L}_{\text{eps}}$ .

**Ablation study experimental setup.** We evaluate a suite of `CaTSG` variants to isolate the contribution of each component. The configuration of all variants is summarized in Table 6.

Table 6: Ablation settings details.

| Mode | Env. Bank $E$ | Env. Prob $p(e)$ | $\mathcal{L}_{\text{sw}}$ | $\mathcal{L}_{\text{orth}}$ | Detail |
|---|---|---|---|---|---|
| `CaTSG` | Learnable | Predicted | ✓ | ✓ | Full model. |
| **w/o SW** | Learnable | Predicted | ✗ | ✓ | Randomize $p(e)$. |
| **RandEnv** | Learnable | Random | ✓ | ✓ | Remove the swapped prediction loss. |
| **FrozenEnv** | Frozen | Predicted | ✓ | ✗ | Freeze the environment bank (no learnng). |
| **w/o Env** | None | None | ✗ | ✗ | Remove the environment bank. |

## G MORE EXPERIMENTAL RESULTS

### G.1 ENVIRONMENTAL INTERPRETABILITY

In Section 5.4, we analyzed the distribution and embedding structure of the learned environments on the Harmonic-VM dataset with $K = 8$. Here, we provide additional visualizations on the Harmonic-VM and Air Quality datasets with $K = 4$, as shown in Figure 13. These results further illustrate the consistency of the environment posteriors across splits and the clustering patterns around the learned embeddings.

### G.2 EFFICIENCY & SCALABILITY

**Efficiency Comparison.** Table 7 compares `CaTSG` with classical baselines under matched loaders and precision. For the TimeWeaver variants, since the original paper does not specify the sampling solver, we adopt DDPM with 20 steps to ensure a fair and consistent comparison. As expected, GAN-based models perform a single forward pass at inference and therefore achieve very high sampling throughput. Focusing on diffusion models with the same 20-step budget, `CaTSG` reaches 144 seq/s, exceeding TW-CSDI (90 seq/s) and TW-SSSD 14 seq/s, which indicates a clear wall-clock advantage from using an ODE-based DPM-Solver (Lu et al., 2022) at fixed steps. `CaTSG`'s parameter count and model size are comparable to peers. During training, CaTSG processes 16 seq/s, lower than TW-CSDI and TW-SSSD, reflecting the additional cost of environment-aware components.

**Trade-off between speed and quality.** We vary the number of DPM-Solver steps {5, 10, 20, 50, 100} on Harmonic-VM dataset and report 5-run's runtime and accuracy in Table 8. Increasing steps markedly improves quality at roughly linear cost in runtime. From 5 to 10 steps, MDD/KL/MMD/J-FTSD drop by 47.2%/85.7%/81.7%/91.1% while runtime rises $\sim 1.8\times$ (5.49 s→9.96 s). Going to 20 steps yields a further 19.7%/51.4%/45.5%/50.7% reduction at another $\sim 1.9\times$ cost (9.96 s→19.30 s), forming a clear knee. Beyond this point the gains are marginal: 20 to 50 steps improves

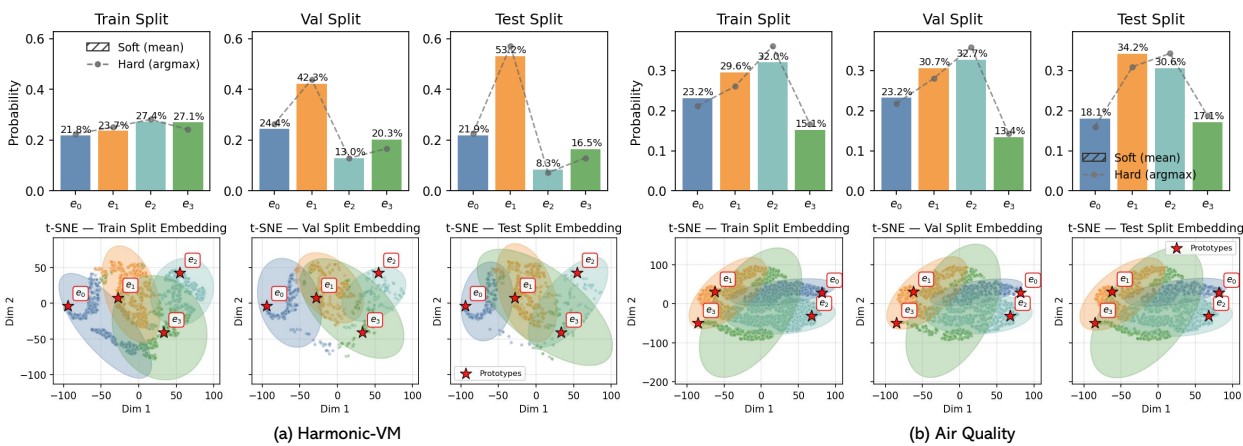

Figure 13: Environment posterior and t-SNE of latent representations on (a) Harmonic-VM and (b) Air Quality dataset When $K = 4$.

Table 7: Efficiency summary. Params: parameter counts. Memory: a model-size proxy in megabytes. Thpt: throughput during training or sampling, sequences per second.

| Method | Params (M) | Memory (MB) | Training | Sampling | | |
|---|---|---|---|---|---|---|
| | | | Thpt (seq/s) | Solver | Steps | Thpt (seq/s) |
| **TimeGAN** | 10.8 | 41.1 | 45 | - | - | 2,250 |
| **WaveGAN** | 9.2 | 35.2 | 180 | - | - | 11,349 |
| **Pulse2Pulse** | 20.3 | 77.4 | 120 | - | - | 4,638 |
| **TW-CSDI** | 18.9 | 72.2 | 25 | DDPM | 20 | 90 |
| **TW-SSSD** | 11.4 | 43.3 | 35 | DDPM | 20 | 14 |
| **CaTSG** | 21.0 | 80.2 | 16 | DPM-Solver | 20 | 144 |

by only 10.2–25.0% at $\sim$2.8$\times$ time (19.30 s$\rightarrow$54.63 s), and 50 to 100 steps yields $\leq$7.1% additional improvement with $\sim$2.0$\times$ time (54.63 s$\rightarrow$107.01 s). Overall, 20 steps offers the best speed–quality trade-off.

Table 8: Ablation on the number of DPM-Solver steps on Harmonic-VM: runtime and accuracy (mean$\pm$std over 5 runs).

| Steps | Runtime (s) | MDD | KL | MMD | J-FTSD |
|---|---|---|---|---|---|
| **5** | $5.49 \pm 0.45$ | $0.231 \pm 0.002$ | $0.244 \pm 0.006$ | $0.120 \pm 0.003$ | $12.489 \pm 2.031$ |
| **10** | $9.96 \pm 1.39$ | $0.122 \pm 0.002$ | $0.035 \pm 0.001$ | $0.022 \pm 0.001$ | $1.111 \pm 0.168$ |
| **20** | $19.30 \pm 2.85$ | $0.098 \pm 0.001$ | $0.017 \pm 0.000$ | $0.012 \pm 0.000$ | $0.548 \pm 0.085$ |
| **50** | $54.63 \pm 2.22$ | $0.088 \pm 0.001$ | $0.014 \pm 0.000$ | $0.009 \pm 0.000$ | $0.447 \pm 0.068$ |
| **100** | $107.01 \pm 4.09$ | $0.087 \pm 0.001$ | $0.013 \pm 0.000$ | $0.009 \pm 0.000$ | $0.424 \pm 0.062$ |

We also visualize the results in Figure 14, which shows that increasing steps improves accuracy with near-linear runtime growth, with a clear knee around 20 steps.

### G.3 STATISTICAL SIGNIFICANCE

.

**Combined Test (Stouffer's Method).** To assess whether the performance gains of **CaTSG** in Table 2 are statistically reliable, we aggregate per–task (dataset$\times$metric) evidence using Stouffer's method (Van Zwet & Oosterhoff, 1967). For each task $i = 1, \ldots, m$, we compute a two–sided Welch $t$–test on the reported mean$\pm$std with $n=5$

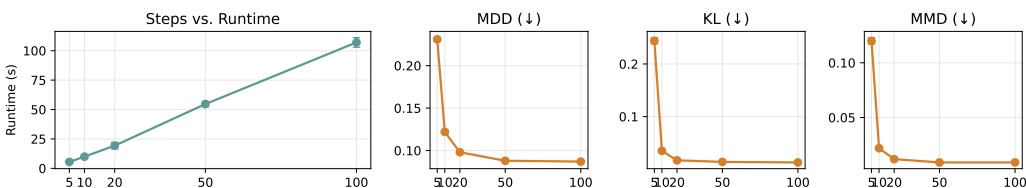

Figure 14: Speed-quality trade-off on Harmonic-VM: runtime vs. steps (*left*) and MDD/KL/MMD vs. steps (*right*).

seeds (unequal–variance correction), obtain its $p$–value $p_i$, convert it to a signed normal score $Z_i$ and combine them as $Z_{\text{comb}} = \frac{\sum_{i=1}^m Z_i}{\sqrt{m}}$. We then read the overall two–sided $p$–value from the standard normal distribution as $p_{\text{overall}} = 2\big(1 - \Phi(|Z_{\text{comb}}|)\big) = \text{erfc}\big(\frac{|Z_{\text{comb}}|}{\sqrt{2}}\big)$, where $\Phi$ is the standard normal CDF and $\text{erfc}$ is the complementary error function. As summarized in Table 9, all baselines exhibit very large positive $Z_{\text{comb}}$ (10–18) with correspondingly tiny $p$–values ($10^{-25}$–$10^{-74}$), indicating consistent and substantial superiority of **CaTSG** across tasks.

Table 9: Overall significance against **CaTSG** using Stouffer's combined $Z$.

| Baseline | $Z_{\text{comb}}$ | $p$-value | Stars |
|---|---|---|---|
| **TimeGAN** | 18.186 | $6.60 \times 10^{-74}$ | *** |
| **WaveGAN** | 10.329 | $5.17 \times 10^{-25}$ | *** |
| **Pulse2Pulse** | 9.6947 | $3.18 \times 10^{-22}$ | *** |
| **TW-CSDI** | 10.007 | $1.42 \times 10^{-23}$ | *** |
| **TW-SSSD** | 13.721 | $7.54 \times 10^{-43}$ | *** |

**Confidence Interval Analysis.** To rigorously evaluate the performance of our proposed model, we report $95\%$ Confidence Intervals (CIs) to assess the robustness of per-task comparisons against the second-best baseline (Table 10). CI separation is summarized in the last column, where ✓ denotes non-overlapping and **CaTSG** significantly better, ≈ denotes overlapping and × denotes non-overlapping and **CaTSG** significantly worse. On the synthetic datasets, Harmonic-VM shows consistent and large improvements across all metrics with non-overlapping CIs, while Harmonic-VP is mostly significant except for J-FTSD. On real-world benchmarks, Air Quality exhibits significant gains on MDD and KL, and Traffic has significant gains on MDD, MMD, and J-FTSD. Overall, **CaTSG** delivers reliable improvements where CIs are non-overlapping, with the largest margins in synthetic settings.

G.4 COUNTERFACTUAL EVALUATION FOR REAL-WORLD DATA

In real deployments, counterfactual ground truth is unobservable. In addition, unlike computer vision tasks with readily inspectable targets, counterfactual ground truth in time series is inherently unobservable, making evaluation nontrivial. Beyond observational fit, potential evaluations for CF-TSG include: (1) J-FTSD–style joint alignment of $(X, C, E)$ using task-aware embeddings; (2) environment-distribution preservation, certifying that counterfactual samples do not spuriously alter environmental attributes (e.g., season, location) relative to the inferred $E$; and (3) content-level similarity under interventions, which measures distances along content axes after disentangling environment from content (e.g., with PCA/LDA or disentangled autoencoders).

**A preliminary and practical evaluations.** The metrics presented below are interpreted as a preliminary set of evaluation tools that we found feasible in practice to characterize different aspects of counterfactual quality. We view these evaluations as an initial step toward more rigorous validation protocols for CF-TSG.

- **E-FTSD** (Environment-aware Fréchet Time Series Distance). To jointly evaluate counterfactual fidelity while accounting for latent environmental structure, we introduce a J-FTSD-style (Narasimhan et al., 2024) metric, i.e., E-FTSD. Given real samples $D = \{(x_i, c_i)\}_{i=1}^n$ and counterfactual samples $\hat{D} = \{(\hat{x}_i, c_i)\}_{i=1}^n$, and the environment bank $E = \{e_k\}_{k=1}^K$ with inferred environment distribution $p(e_k)$, we first obtain time-series and condition embeddings via two encoders $\phi_x(\cdot)$ and $\phi_c(\cdot)$. For each real sample we construct the joint embedding $z_i^r = \phi_x(x_i) \oplus \phi_c(c_i)$, and for each generated sample under environment $e_k$ we compute $z_{i,k}^g = \phi_x(\hat{x}_{i,k}) \oplus \phi_c(c_i) \oplus e_k$. To remove the variance introduced by sampling different environments, we marginalize environment variability by

Table 10: Per-task comparison against the second-best baseline. Entries show the $95\%$ CI, relative improvement of **CaTSG**, and whether CIs are non-overlapping. CI sep.: ✓ = non-overlapping (**CaTSG** significantly better), ≈ = overlapping (inconclusive), × = non-overlapping (**CaTSG** significantly worse).

| Dataset | Metric | Second-best | Second-best (95% CI) | **CaTSG** (95% CI) | Rel. $\Delta$ | CI sep. |
|---|---|---|---|---|---|---|
| **Harmonic-VM** | MDD | TW-CSDI | [0.266, 0.268] | [0.097, 0.099] | 63.3% | ✓ |
| | KL | TW-CSDI | [0.202, 0.214] | [0.017, 0.017] | 91.8% | ✓ |
| | MMD | TW-CSDI | [0.080, 0.080] | [0.012, 0.012] | 85.0% | ✓ |
| | J-FTSD | TW-CSDI | [2.476, 3.088] | [0.442, 0.654] | 80.3% | ✓ |
| **Harmonic-VP** | MDD | TW-CSDI | [0.077, 0.077] | [0.052, 0.054] | 31.2% | ✓ |
| | KL | TW-CSDI | [0.201, 0.215] | [0.062, 0.070] | 68.3% | ✓ |
| | MMD | TW-SSSD | [0.084, 0.084] | [0.034, 0.034] | 59.5% | ✓ |
| | J-FTSD | TW-CSDI | [12.583, 13.645] | [9.660, 15.694] | 3.3% | ≈ |
| **Air Quality** | MDD | Pulse2Pulse | [0.081, 0.081] | [0.069, 0.069] | 14.8% | ✓ |
| | KL | TW-SSSD | [0.357, 0.357] | [0.300, 0.302] | 15.7% | ✓ |
| | MMD | TW-SSSD | [0.058, 0.060] | [0.070, 0.070] | -18.6% | × |
| | J-FTSD | TW-CSDI | [2.970, 4.544] | [2.312, 3.546] | 22.0% | ≈ |
| **Traffic** | MDD | TW-SSSD | [0.091, 0.091] | [0.071, 0.071] | 22.0% | ✓ |
| | KL | TW-CSDI | [0.154, 0.156] | [0.116, 0.228] | -11.0% | ≈ |
| | MMD | TW-SSSD | [0.038, 0.038] | [0.006, 0.006] | 84.2% | ✓ |
| | J-FTSD | TW-CSDI | [5.912, 7.216] | [0.655, 0.791] | 89.0% | ✓ |

forming a weighted average: $z_i^g = \sum_{k=1}^K p(e_k) z_{i,k}^g$. Let $\mu_d$ and $\Sigma_d$ ($d \in \{r, g\}$) denote the empirical mean and co-variance of $\{z_i^d\}_{i=1}^n$. E-FTSD is then defined as the Fréchet distance between the real and environment-marginalized generated distributions:

$$\text{E-FTSD}(D, \hat{D}) = \|\mu_r - \mu_g\|_2^2 + \text{Tr}\left(\Sigma_r + \Sigma_g - 2(\Sigma_r \Sigma_g)^{\frac{1}{2}}\right).$$

This metric evaluates distributional alignment between real and counterfactual samples while explicitly marginalizing latent environmental variability, thereby isolating differences attributable to the intervention rather than to changes in the environment.

- **Env-KL** (Environment Preservation Score). To ensure that counterfactual samples do not inadvertently alter latent environmental attributes (e.g., season or location), we measure the divergence between the original environment distribution and the environment distribution predicted from counterfactual samples. Let $P_{\text{orig}}(e)$ denote the environment distribution of real samples, obtained from the soft environment assignments $p(e \mid x, c)$ during training, and let $P_{\text{cf}}(e)$ denote the environment distribution predicted by a trained environment classifier applied to the counterfactual samples $\hat{x}$. The environment-preservation metric is defined as the KL divergence:

$$\text{Env-KL} = D_{\text{KL}}\left(P_{\text{orig}} \,\|\, P_{\text{cf}}\right) = \sum_{k=1}^K P_{\text{orig}}(e_k) \log \frac{P_{\text{orig}}(e_k)}{P_{\text{cf}}(e_k)}.$$

A lower Env-KL value indicates that the counterfactual sample remains closer to its original latent environment and therefore preserves environment-related characteristics under intervention.

- **C-Consis** (Counterfactual Content Similarity) To assess whether counterfactual samples preserve the content associated with the original contextual variable **c**, we estimate a content subspace by removing dimensions that are predictive of the inferred environment **e**. Given real and counterfactual samples **x** and $\hat{x}$, we first flatten each sequence into vectors in $\mathbb{R}^{LD}$ and fit a Linear Discriminant Analysis (LDA) classifier using the environment indices as labels. LDA yields a set of environment-discriminative directions $\mathbf{W}_e \in \mathbb{R}^{LD \times (K-1)}$ that capture variation explained by **e**, from which we obtain the residual content subspace by projection:

$$\mathbf{x}_{\text{content}} = \mathbf{x} - \mathbf{W}_e(\mathbf{W}_e^\top \mathbf{x}), \qquad \hat{\mathbf{x}}_{\text{content}} = \hat{\mathbf{x}} - \mathbf{W}_e(\mathbf{W}_e^\top \hat{\mathbf{x}}).$$

The C-Consis score is defined as the Euclidean distance between real and counterfactual content representations:

$$\text{C-Consis} = \frac{1}{N} \sum_{i=1}^N \left\| \hat{\mathbf{x}}_{\text{content}}^{(i)} - \mathbf{x}_{\text{content}}^{(i)} \right\|_2^2,$$

Table 11: Preliminary evaluation results on the real-world Air Quality dataset.

|  | CaTSG | RandEnv | w/o SW | FrozenEnv |
|---|---|---|---|---|
| **MDD** (val) | $0.064 \pm 0.000$ | $0.086 \pm 0.000$ | $0.085 \pm 0.000$ | $0.089 \pm 0.000$ |
| **KL** (val) | $0.290 \pm 0.001$ | $2.549 \pm 0.002$ | $2.542 \pm 0.001$ | $2.631 \pm 0.002$ |
| **MMD** (val) | $0.058 \pm 0.000$ | $0.091 \pm 0.001$ | $0.086 \pm 0.001$ | $0.061 \pm 0.000$ |
| **J-FTSD** (val) | $0.806 \pm 0.608$ | $1.622 \pm 0.552$ | $2.868 \pm 2.780$ | $0.953 \pm 0.785$ |
| **E-FTSD** (val) | $1.532 \pm 0.608$ | $2.545 \pm 0.552$ | $3.237 \pm 2.778$ | $1.877 \pm 0.786$ |
| **MDD** (test) | $0.064 \pm 0.000$ | $0.086 \pm 0.000$ | $0.084 \pm 0.000$ | $0.089 \pm 0.000$ |
| **KL** (test) | $0.264 \pm 0.002$ | $1.907 \pm 0.002$ | $1.893 \pm 0.001$ | $1.957 \pm 0.001$ |
| **MMD** (test) | $0.070 \pm 0.000$ | $0.095 \pm 0.001$ | $0.090 \pm 0.001$ | $0.069 \pm 0.000$ |
| **J-FTSD** (test) | $2.905 \pm 0.457$ | $4.900 \pm 0.313$ | $4.757 \pm 0.606$ | $3.586 \pm 0.799$ |
| **E-FTSD** (test) | $3.631 \pm 0.456$ | $5.823 \pm 0.314$ | $5.128 \pm 0.606$ | $4.510 \pm 0.799$ |
| **Env-KL** | $3.435 \pm 0.103$ | $4.175 \pm 0.218$ | $3.746 \pm 0.118$ | $4.594 \pm 0.233$ |
| **C-Consis** | $1.754 \pm 0.005$ | $11.921 \pm 0.154$ | $30.188 \pm 0.355$ | $27.520 \pm 0.213$ |

where lower values indicate better preservation of content under the intervention.

In Table 11 we summarize our evaluation of **CaTSG** and its ablation variants (RandEnv, w/o SW, FrozenEnv) under a comprehensive set of metrics. First, we report the standard distributional similarity metrics (MDD, KL, MMD, J-FTSD) that are also used in the main text. Second, to support our preliminary exploration of counterfactual quality on real-world data—where ground-truth counterfactuals are inherently unobservable—we further include the additional validity metrics introduced above (E-FTSD, Env-KL, and C-Consis).

Because some metrics measure distributional alignment between generated samples and the underlying data distribution, while others rely on comparing generated samples with held-out observations, we report results on both the validation and test sets. The results consistently show that **CaTSG** achieves the best or near-best scores across both sets and both groups of metrics, while ablation variants tend to introduce noticeable degradation in either distributional alignment or environment/content preservation.

### G.5 Variance Analysis

As causal estimation typically involves a bias–variance trade-off, we investigate whether incorporating a latent environment variable $E$ introduces unstable variance. To this end, Figure 15 visualizes the standard deviation of evaluation metrics across multiple random seeds (as reported in Table 2) for each method, focusing on **KL** and **MMD**, which reflect distributional alignment.

We plot results separately for each metric across three datasets (Harmonic-VM, Harmonic-VP, Air Quality). Despite involving latent variables, **CaTSG** exhibits the lowest or near-lowest variance in all settings, indicating that our model achieves effective bias reduction without introducing instability. We attribute this to our design choices, such as soft posterior inference and embedding regularization, which mitigate stochasticity while preserving representation flexibility.

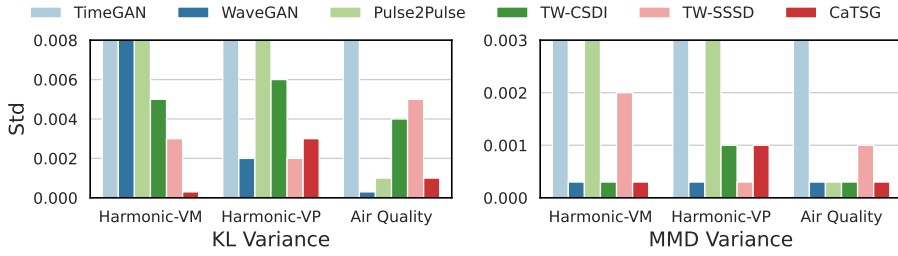

Figure 15: Standard deviation across seeds on KL and MMD metrics. Lower is better.

## G.6 GUIDANCE COEFFICIENT SENSITIVITY

To assess the robustness of classifier-free guidance in our model, we conduct a sensitivity analysis over the guidance coefficient $\omega$, which controls the trade-off between fidelity and diversity during sampling. We evaluate $\omega \in \{0.1, 0.2, 0.5, 1.0\}$ on two datasets using standard generation metrics (MDD and KL divergence). As shown in Table 12, small values of $\omega$ (e.g., 0.2) yield the best or near-best scores across datasets, while larger $\omega$ amplifies conditional signal at the cost of diversity, leading to over-regularized generations. This trend justifies our default choice of $\omega = 0.2$.

Table 12: Sensitivity analysis of guidance coefficient $\omega$.

| $\omega$ | Harmonic-VM | | Air Quality | |
|---|---|---|---|---|
| | MDD $\downarrow$ | KL $\downarrow$ | MDD $\downarrow$ | KL $\downarrow$ |
| 0.1 | $0.077 \pm 0.001$ | $0.011 \pm 0.001$ | $0.081 \pm 0.000$ | $0.316 \pm 0.003$ |
| 0.2 | $0.098 \pm 0.001$ | $0.017 \pm 0.000$ | $0.069 \pm 0.000$ | $0.301 \pm 0.001$ |
| 0.5 | $0.169 \pm 0.001$ | $0.069 \pm 0.002$ | $0.092 \pm 0.000$ | $0.409 \pm 0.003$ |
| 1.0 | $0.243 \pm 0.002$ | $0.171 \pm 0.001$ | $0.101 \pm 0.000$ | $0.512 \pm 0.002$ |

## H MORE DISCUSSION

### H.1 CAUSAL TIME SERIES GENERATION: WHAT, WHY, AND HOW

We introduce a family of causal time series generation tasks. Below we provide a concise takeaway via four aspects: *What*, *Why*, and *How*.

**What are they?** We extend conditional TSG from observational setting to causal targets at two levels on Pearl's causal ladder (Section 4.1):

- **Observational TSG (Obs-TSG):** given observed context $C$, generate sequences $X$ from the observational distribution $P(X \mid C)$.

- **Interventional TSG (Int-TSG):** simulate how $X$ responds to $C$ after neutralizing all other latent factors (i.e., altering the *data-generating mechanism* rather than just conditioning), so that the result reflects a *population-average* $X$ across typical environments/individuals.

- **Counterfactual TSG (CF-TSG ):** for a specific individual/trajectory with realized $(X, C)$, ask "what would have happened if $C$ had been $C'$ at that time?" The target is $P(X' \mid X, C, C')$, which is the *individual-level* counterfactual generation.

Intuitively, Obs-TSG *replays* the world as it is; Int-TSG reports how the *population* responds to $C$ on an *average environment*; CF-TSG reports how *this individual* would respond under an alternative $C'$ while preserving its own latent condition.

**Why are they needed?**

- **Avoiding confounding bias.** In many domains (traffic, air quality, energy, healthcare), $c$ and $x$ are jointly influenced by unobserved factors and thus may have spurious correlations (Appendix E.3). Reporting $P(X \mid C)$ alone risks telling the story of a particular environment/individual. Int-TSG targets $p(x \mid \mathrm{do}(c))$ and, by design, removes confounding bias and give a the environment-neutral response, which averages over typical environments rather than any single setting.

- **Individual accountability and explanation.** Many decisions require an individual-level "what if". CF-TSG preserves the individual's unobserved condition (the case-specific circumstances, including the patient's health state) as inferred from the observed heart-rate series $X$ and the administered treatment $C$, and changes only the treatment from $C$ to $C'$ to generate a counterfactual sequence $X'$ under the same circumstances. For example, given a patient's vital-sign trajectory under dose $C$, CF-TSG imagines how the *same* patient's trajectory would have evolved under dose $C'$, while keeping the patient's condition on that day unchanged.

- **Robust generalization.** By averaging over $E$ (Int-TSG) and preserving individual $E$ (CF-TSG), the learned generator becomes less sensitive to shifts in season, region, or load regime, improving portability and stability.

- **Unifying generation–decision–explanation.** Obs-TSG can replay data but cannot support intervention planning or counterfactual reasoning. Int/CF-TSG turn a generator into a tool that can *act* (simulate environment-neutral responses to $C$) and *explain* (instance-level alternatives under $C'$), closing the loop from modeling to decision support.

**How to realize them?** We adopt diffusion as the backbone, given its strong performance on time–series generation (Yang et al., 2024). Our unified model, `CaTSG`, enables Obs-/Int-/CF-TSG by (1) introducing a learnable environment bank that captures latent background conditions and (2) replacing standard conditional guidance with backdoor adjusted guidance during sampling (Eq. 6). Sepcifically, training follows Algorithm 1. For Int-TSG, `CaTSG` aggregates per-environment predicted noise with environment weights (Algorithm 2). For CF-TSG, it follows an Abduction–Action–Prediction routine: infer the instance's environment posterior from $(X, C)$, set the alternative control $C'$, and predict the noise (Algorithm 3). When the environment module is disabled, `CaTSG` reduces to a standard observational generator for Obs-TSG.

### H.2 MORE EXAMPLES ACROSS DOMAINS

| | Variable | Seeing | Doing | Imaging |
|---|---|---|---|---|
| *General* | $X$: Target time series
$C$: Contextual variable
$E$: Latent confounder influencing both $X$ and $C$ | Observe that $X$ is correlated with $C$, but this may be confounded by latent factor $E$. | Estimate how $X$ would change under an intervention on $C$, adjusting for $E$. | Given observed $(X, C)$, simulate what $X'$ would have been under alternative $C'$, keeping $E$ fixed. |
| *Healthcare* | $X$: Patient physiological indicators (e.g., temperature, heart rate)
$C$: Treatment (e.g., type and dosage of medication)
$E$: Health condition (e.g., disease severity, comorbidities) | Observe that patients who received Treatment A recover faster, but this may reflect differences in health condition. | Estimate how recovery would look under Treatment A for any patient, regardless of their health condition. | For a patient who received Treatment B and recovered slowly, generate how they would have recovered under Treatment A. |
| *Energy* | $X$: Electricity demand (hourly/daily)
$C$: Energy-saving policy level
$E$: City profile (e.g., industrial vs. residential) | Observe that stricter policies are associated with lower demand, but this may be biased by city profile. | Estimate demand if all cities are given strict policy, regardless of city-level differences. | For a city with observed demand under lenient policy, ask what the demand would have been under stricter policy. |
| *Transportation* | $X$: Traffic flow
$C$: Traffic Contextual strategy (e.g., signal timing, lane closures)
$E$: Road network structure / commuter behavior | Observe that Strategy A reduces traffic jams, but that strategy may be applied only in low-density areas. | Estimate traffic flow under Strategy A, adjusting for road and commuter profiles. | Ask what traffic would have looked like on this road if a different Contextual strategy had been applied. |
| *Education* | $X$: Test score
$C$: Learning aid (e.g., reminders, videos)
$E$: Student type (self-discipline, background knowledge) | Observe that students who got reminders learned faster (higher test scores), but self-disciplined students may opt into them more. | Estimate learning outcomes if all students received the same intervention, regardless of student type. | For a student without reminders, simulate their learning outcome if reminders had been provided. |
| *Finance* | $X$: Stock price sequence
$C$: Public policy signals (e.g., interest rate change)
$E$: Market context (e.g., economic cycle, volatility regime) | Observe that rate hikes tend to reduce stock prices, but this may be due to broader economic downturns. | Estimate how stock price sequences would evolve under a given policy, adjusting for latent market regimes. | Given observed stock prices under a no-hike policy, simulate the price trajectory that would have occurred under a rising-rate policy, assuming the same market condition. |

Table 13: Representative TSG examples under observational (Seeing), interventional (Doing), and counterfactual (Imagining) settings (Pearl, 2009) across different application domains.

To further illustrate the applicability of the proposed causal time series generation tasks, i.e., Int-TSG and CF-TSG in real-world scenarios, we provide additional examples across various domains in Table 13. These complement the transportation case discussed in the Causal Ladder (Table 1). Together, these examples highlight the significance and versatility of causal time series generation across different application settings.

### H.3 CAUSAL ASSUMPTION AND CONFOUNDING TREATMENTS

**Fixed SCM assumption.** In our `CaTSG` solution for casual time series generation, we adopts a predefined SCM (Section 4.1) to enable identification of interventional and counterfactual objectives. While this SCM affords a principled bridge from observational to interventional and counterfactual generation, it may also introduce several limitations: First, if true mechanisms violate the assumed edges or independences, e.g., unmodeled mediators, feedback loops ($C \leftrightarrow X$), time-varying confounding, or nonstationarity, the backdoor set may fail to block spurious paths and Counterfactual inferences may be biased. Second, we approximate confounding $E$ by a finite environment bank with

$K$ embeddings, implicitly assuming that conditioning on $E$ renders $C$ ignorable for $X$. When $K$ is misspecified or $E$ cannot summarize all relevant confounding, residual bias can remain; identifiability is only up to a permutation of embeddings and may be sensitive to representation entanglement. Third, if certain $(\mathbf{c}, \mathbf{e})$ combinations are rarely or never observed, estimating $P(X \mid \mathrm{do}(C{=}\mathbf{c}))$ may be much more challenging. Fourth, applying a fixed SCM across domains (e.g., finance or healthcare) presumes mechanism invariance and similar support; violations (shifts in causal structure, intervention ranges, or measurement processes) can degrade robustness and external validity.

Overall, the predefined SCM assumption should be regarded as a simplifying choice that enables tractable formulation and analysis of causal time series generation. Our framework thus represents only an initial step toward this new direction, i.e., causal time series generation. We anticipate that future work will develop more expressive and adaptive models, e.g., with learned or dynamic causal structures, that can support richer forms of interventional and counterfactual generation in complex real-world settings.

**Genuine and spurious relationship.**    In many observational time series systems, an observed dependence combines both a genuine causal effect and a spurious component induced by external contextual factors. The edge $C \to X$ in our running example (i.e., Figure 1) reflects this mixed structure: while $C$ (e.g., environmental conditions) may exert a true influence on $X$, the dominant variation in school-neighborhood traffic is also strongly shaped by the school calendar, producing a confounded association between $C$ and $X$. The resulting dependence therefore has a mixture of structural and spurious effects. Therefore, our model addresses this issue by introducing the environment variable $E$, which models the spurious calendar-driven variation, thereby isolating the stable causal component of the $C \to X$ relationship.

**Finite environment bank.**    Using a finite environment bank $E$ may limit the expressiveness of the learned latent confounder representation, potentially leaving residual confounding if $K$ is too small or the embedding space lacks sufficient capacity. In our work, we adopt this design as a tractable and interpretable approximation to the latent confounding space, with the goal of enabling causally grounded generation via a structured yet trainable environment representation. To mitigate the risk of under-specification, our framework incorporates several mechanisms: (1) We allow soft environment inference with probabilistic weighting $w_k$, reducing the impact of hard mis-assignment. (2) We apply diversity-promoting regularization (e.g., orthogonality loss) across the environment embeddings to encourage factor disentanglement. (3) The adopted swapped loss promotes environment consistency under stochastic augmentations, discouraging over-fragmentation or collapse. These strategies aim to ensure that even with moderate $K$, the environment bank captures a diverse and functionally sufficient set of latent confounders for effective adjustment. In future work, one could consider hierarchical or multi-factor latent environment models to further improve confounding representation granularity.

### H.4    POSITIONING & SCOPE

**CaTSG** is designed to target interventional and counterfactual objectives, i.e., $P(X \mid \mathrm{do}(C))$ and $P(X' \mid X, C, C')$. This is particularly useful in regimes with environment shifts, where a latent environment $E$ varies and acts as a confounder for the observational relation between $C$ and $X$ to mirrors real-world scenarios (i.e., under SCM in Figure 2b instead of Figure 2a). Our curated datasets and split strategy are constructed to highlight this setting by inducing changes in $E$. The goal is to evaluate generative targets aligned with interventions and counterfactuals, rather than to optimize observational fit under near-iid conditions. In such shifted regimes, modeling $P(X \mid \mathrm{do}(C))$ with latent $E$ yields observable gains in fidelity metrics. When $E$ are stable and fully observed, standard conditional Obs-TSG models that targets $P(X \mid C)$ will still be competitive. In scenarios where $E$ is effectively constant, **CaTSG** reduces to the conditional case (a single or degenerate environment), and performance improvement are expected to be small or negligible. Typical applications of the targeted regime include policy simulation and what-if planning under seasonal or regime changes (e.g., weather patterns in traffic, meteorology in air quality), and cross-region transfer. Examples across different domains are presented in Table 13.

### H.5    COMPARISON WITH RELATED WORKS

Prior causal generative models are mostly developed for static modalities, e.g., images via label-graph–consistent GANs or modular recombinations (Kocaoglu et al., 2018; Sauer & Geiger, 2021), text via interventions on latent narrative factors (Hu & Li, 2021), and knowledge graphs via hypothetical triple generation (Liu et al., 2021). our work, to the best of our knowledge, is the *first* attempt to propose and instantiate *causal time series generation*, where causal queries target entire sequential data under interventions and regime shifts. Methodologically, within an SCM-based formulation, we realize causality through a back-door–adjusted score guidance that injects confounder adjustment directly into the diffusion sampling field, and we couple it with a learnable environment bank to model

latent environments that drive nonstationarity. This is different from designs that either intervene on observed, single-shot factors or require acyclicity-constrained latent DAGs as in CausalVAE (Yang et al., 2021).

## H.6 LIMITATIONS AND FUTURE DIRECTIONS

This work represents an initial attempt to formulate and address the new task of *causal time series generation*. While our framework provides a proof-of-concept by combining predefined SCMs with backdoor-adjusted diffusion, it should be viewed as a starting point rather than a complete solution. As such, there are several limitations, but these also open up numerous opportunities for future research.

**Limitations.** (1) we currently restrict conditions $\mathbf{c}$ to time series inputs, whereas in practice they could be multi-modal (e.g., textual guidelines or visual cues), which would enable richer interventional and counterfactual generation. (2) the SCM structure is predefined, which may reduce robustness when the assumed causal graph does not hold or when transferring to domains with different mechanisms, such as finance or healthcare (as discussed in Appendix H.3). Third, evaluating counterfactual generation remains challenging, since ground-truth counterfactuals are only available in synthetic settings, limiting systematic validation in real-world data (as discussed in Appendix G.4). (3) `CaTSG` is design for Int-/CF-TSG and thus tailored for settings with environment shifts and latent confounding E, where aligning the generator with interventional and counterfactual targets is most beneficial. When environment is stable, CaTSG effectively reduces to the conditional case ($\omega=0$) and any performance gains are expected to be small or negligible (as discussed in Appendix H.4).

**Future directions**. Addressing these limitations opens multiple avenues for future work. One direction is to explore multi-modal conditioning (e.g., textual guidelines or visual cues) like TimeWeaver (Narasimhan et al., 2024), which would enable richer interventional and counterfactual generation scenarios. Another is to learn or adapt SCM structures directly from data rather than relying on predefined graphs, thereby improving robustness when transferring across domains. Finally, evaluation should be extended beyond synthetic and benchmark datasets to diverse real-world domains, where causal assumptions can be more rigorously examined and the framework's robustness more thoroughly validated.

## H.7 USE OF LARGE LANGUAGE MODELS (LLMS)

In preparing this manuscript, we used LLMs (ChatGPT) for language polishing purposes. Specifically, it was employed to improve clarity, grammar, and fluency of the text. All conceptual contributions, methodological designs, experiments, and analyses were conceived and executed entirely by the authors.

