# OpenReview forum: "Causal Time Series Generation via Diffusion Models"
_ICLR.cc/2026/Conference — ICLR 2026 Conference Desk Rejected Submission_

### Official Review · Reviewer_zSS3 · 2025-10-20

**Soundness:** 3
**Presentation:** 4
**Contribution:** 3
**Rating:** 8
**Confidence:** 3

**Summary:**

The paper proposes a diffusion-based framework that extends observational time series generation (TSG) to interventional and counterfactual TSG under Pearl’s causal hierarchy, addressing unobserved confounding. Specifically, the authors generalize the standard definition of score functions to their causal counterparts under the structural causal model (SCM) framework. Building on this formulation, they propose CaTSG, a model comprising an environment inference module (EnvInfer), a learnable environment bank with an encoder, and a denoiser that integrates both conditional and environmental variables to produce the final samples. The effectiveness of CaTSG is validated on two synthetic and two real-world datasets, demonstrating its capability to learn unobserved confounders and generate coherent counterfactuals.

**Strengths:**

•	The authors explain the effect of unobserved confounding in the introduction by using an intuitional example, which enhances readers’ understanding of the concept, though I think summer holidays do not “cause” high temperature, but only “coincide” with high temperature instead.

•	The concepts of “intervention” and “abduction” are well illustrated in Figure 2.

•	The idea of extending CFG and score functions to causal learning is interesting, though there exist some studies that follow similar ideas, such as [1].

•	The proposed model consistently outperforms other benchmarks on various types of metrics. The counterfactual analysis on the results for real-world data mostly makes sense.

•	The authors explore different combinations of hyperparameters and conduct a brief analysis on the efficiency (e.g., number of inference steps) of the proposed method.

References:

•	[1]. Xia, Tian, et al. "Decoupled Classifier-Free Guidance for Counterfactual Diffusion Models." arXiv preprint arXiv:2506.14399 (2025).

**Weaknesses:**

•	It would be better to further discuss recent studies that utilize diffusion model or Classifier-free Guidance for counterfactual generation, such [1] and [2].

•	For other aspects of potential weaknesses, please refer to the “Questions” section.

References:

[1]. Komanduri, Aneesh, et al. "Causal diffusion autoencoders: Toward counterfactual generation via diffusion probabilistic models." arXiv preprint arXiv:2404.17735 (2024).

[2]. Wu, Shenghao, et al. "Counterfactual generative models for time-varying treatments." Proceedings of the 30th ACM SIGKDD Conference on Knowledge Discovery and Data Mining. 2024.

**Questions:**

•	What is the rationale behind the adoption of the swapped loss $\mathcal{L}_{sw}$? For example, can we use a SimSiam-style loss $\mathcal{L} = \frac{1}{2} \mathcal{D}(h’, w’’) + \frac{1}{2} \mathcal{D}(h’’, w’)$ where $\mathcal{D}(\cdot)$ is a similarity measure?

•	Since the environment bank $\textbf{E}$ is learnable, how shall we initialize it?

•	Although the authors do an ablation study on the hyperparameters of CaTSG including number of environment embeddings $K$ and their dimension $H$, the search space appears to be quite limited. To be specific, how shall we choose $K$ and $H$ when there might exist more than one unobserved confounder?

•	Have you tried the Flow Matching solver for the denoiser?

---

> ### Author Response · Authors · 2025-11-20
> **Author Response**
>
> We sincerely thank the the reviewer for the constructive and insightful comments. We appreciate the reviewer’s positive assessment of our work’s presentation quality and the clarity of our causal formulation. To address the feedback, we have submitted an updated version of the manuscript, with all revisions highlighted in red. Below, we address each point in detail.
>
> **[Weaknesses]**
>
> **W1.** It would be better to further discuss recent studies [1] and [2].
>
> **A1.** We thank the reviewer for suggesting these related works, and we agree they are relevant. CausalDiffAE [1] learns causal representation learning for static image data, aiming to disentangle latent factors and generate counterfactual images via DDIM sampling. In contrast, our work focuses on time series generative modeling. The second work [2] estimates counterfactual outcomes under time-varying treatment, using inverse-probability weighting and plug-in conditional generators. In contrast, our work instead provides a unified causal diffusion framework directly for interventional and counterfactual time series generation. We appreciate the suggestion and have included them in the revised version (Line 129-132).
>
> **[Questions]**
>
> **Q1.** The rationale behind SwAV-style loss? Can we use SimSiam-style loss?
>
> **A1.** Thank you for the question. Our goal in EnvInfer is to discover latent environments and output a soft assignment over $K$ environment embeddings that later control diffusion sampling. A SwAV-style loss is better aligned with this objective: it learns by predicting cross-view embedding assignments, which (i) induces discrete, controllable environment clusters (the env bank), (ii) encourages separability and balanced prototype usage (via assignment competition/Sinkhorn), and (iii) yields assignments that can be directly used for interventional/counterfactual generation. While a SimSiam-style loss aligns continuous features without online clustering and it does not explicitly form prototype-level structure, making environment identifiability weaker (post-hoc clustering would be needed). Therefore, SwAV-style loss is more suitable in our case.
>
> **Q2.** Since the environment bank $E$ is learnable, how shall we initialize it?
>
> **A2.** Thank you for pointing this out. The environment bank $E$ is initialized with orthogonal initialization to encourage diversity among environment embeddings, followed by L2 normalization along the feature dimension. We have clarified this initialization process to the revision (Line 1124-1125). Thank you!
>
> **Q3**. How shall we choose $K$ and $H$ when there might exist more than one unobserved confounder?
>
> **A3.** Thank you for the insightful comment. The latent environment variable $E$ is intended to represent an effective abstraction over **all** unobserved confounders. Accordingly, both $K$ and $H$ control the capacity of this representation. Therefore, if multiple unobserved confounders are expected, we recommend increasing both $K$ and $H$ to allow the environment bank to capture combinatorial variation. In our current experiments, we chose $K \in$ {$2, 4, 8, 16$} and $H \in$ {$8, 16, 32, 64$}, which provided sufficient variation without exceeding memory limits. Exploring beyond this range may offer further gains but would require scaled-up resources. In addition, CaTSG supports soft assignment $w_k$ and uses regularization (e.g., orthogonality loss) to encourage disentanglement, which mitigates the risk of under-representing multi-causal structures even when $K$ is moderate. We have clarified this rationale in the revision (Line 1558-1560). Thank you!
>
> **Q4.** Have you tried the Flow Matching solver for the denoiser?
>
> **A4.** We appreciate the question. In this work, we prioritized DDPM because it is one of the dominant paradigms in current time series generation (alongside GAN-based approaches). In addition, the score-based formulation of DDPM naturally aligns with our causal setting: by leveraging the backdoor adjustment, we can derive a modified score function that enables principled causal sampling. Nevertheless, we agree that Flow Matching presents a possible alternative for modeling continuous generative trajectories. Adapting our method to Flow Matching would likely require defining a causally adjusted target flow and reformulating the denoising objective to align with vector field matching under interventions. We appreciate this idea as a promising future direction. Thanks!
>
> We thank the reviewer for the helpful comments. We hope that our responses address all the concerns, and we remain happy to clarify any remaining points.

---

### Official Review · Reviewer_wJr5 · 2025-10-25

**Soundness:** 3
**Presentation:** 3
**Contribution:** 2
**Rating:** 6
**Confidence:** 4

**Summary:**

In this work, the authors propose a causal perspective on conditional time-series generation (TSG) and introduce causal TSG as a new task family. The paper makes three key contributions:
1.	Causal expansion of conditional TSG. The authors formalize causal TSG as an extension of conditional TSG along Pearl’s ladder, adding two tasks beyond association—interventional (Int-TSG) and counterfactual (CF-TSG)—to better align generative modeling with real-world decision making.
2.	Unified causality-guided diffusion framework. They derive causal score functions via backdoor adjustmentand abduction–action–prediction (AAP), and instantiate CaTSG, which embeds these principles into diffusion sampling. Using backdoor-adjusted guidance and a learnable latent environment bank, CaTSG supports observational, interventional, and counterfactual generation within a single framework.
3.	Comprehensive empirical evaluation. Across four datasets, CaTSG consistently improves observational fidelity and achieves competitive interventional performance, outperforming the second-best baseline by 2.4%–98.7%across four metrics. For counterfactuals, CaTSG produces accurate CF generations on synthetic datasets (with ground-truth counterfactuals) supported by environment analyses, and yields reasonable CF results on real-world datasets via case-study visualizations.
Empirically, CaTSG is evaluated on two synthetic oscillator datasets and two real datasets (Air Quality, Traffic), showing improvements over TimeGAN/WaveGAN/Pulse2Pulse and TimeWeaver variants on MDD, KL, MMD, and J-FTSD; ablations indicate that each component (environment bank, SwAV-style loss, and guidance) contributes to the gains.

**Strengths:**

The method demonstrates good originality. The idea of using the environment as additional conditioning information is an effective way to help the model learn more stable relationships between X and C. The use of the SwAV-style training paradigm is also an interesting and sensible approach to learn useful "clustering" of the (X,C) pairs, enabling the model to absorb the underlying factors that generate the variation in these pairs.

The paper presents thorough experiments across four datasets and consistently demonstrates strong performance.

**Weaknesses:**

The explanation of how the environment is learned and generated could be clearer. More intuition should be provided regarding why the SwAV-style training task is well-suited for learning the environment.

Adopting the backdoor-adjusted approach is not without its challenges, as it requires learning additional components. Specifically, for each environment, the model needs to learn a conditional distribution P(X∣C,E=e). The paper does not adequately discuss the potential cost of using backdoor adjustment. A possible experiment could involve using untrained or completely noisy environment variables to see how the model performs compared to the unadjusted approach. It would be insightful to observe the negative impact of using backdoor adjustment and how the dimension and complexity of the (uninformative) environment variable might affect performance.

Following up on the previous point, causal methods typically target bias reduction. In this case, adding a learned environment variable is intended to decrease the bias in estimating the conditional distribution P(X∣do(C)). Usually, the trade-off is an increase in variance. However, the paper does not address this. Most of the uncertainty quantification analyses show that CaTSG outperforms other methods. This could be because the uncertainty metrics include both bias and variance, so the increase in variance is not separately observable. I recommend adding metrics, plots, or visualizations to demonstrate the increased variance and engineering challenging cases where the variance dominates the bias, causing CaTSG to underperform compared to conventional methods. I believe this experiment is essential to demonstrate the reliability of causal-based methods.

While I appreciate the visualization of the entire pipeline for the proposed method, I personally find Figure 3 a bit too dense for easy reading. Perhaps the key components could be emphasized, and less important parts of the method could be removed or downplayed.

**Questions:**

Questions:
1.	Have you considered using a continuous or high-dimensional environment variable E? In many cases, splitting the confounding information into K discrete clusters or categories of environments can lead to poor performance, especially when there are many weak confounders that cannot be easily separated into K discrete environments.
2.	Have you considered using additional data or information to aid in learning the confounders? For example, instead of using SwAV, could you use supervised learning with additional data/information to predict (X,C), which could help in learning better embeddings for the confounders?
3.	Have you considered an inverse probability weighting (IPW) formulation for the problem?
If you have considered the above questions, I would be interested to hear why these alternatives were not explored.

---

> ### Author Response · Authors · 2025-11-20
> **Author Response (Part 1)**
>
> We sincerely thank the reviewer for the thoughtful feedback. We are pleased that the originality of our approach, the use of environment-based conditioning and the overall empirical strength across four datasets were positively received. In response to the reviewer’s comments, we have uploaded a revised manuscript in which all modifications are marked in red for easy reference. Below, we address the reviewer’s comments and questions point by point.
>
> **[Weaknesses]**
>
> **W1.** More intuition should be provided regarding why the SwAV-style training task is well-suited for learning the environment.
>
> **A1.** We thank the reviewer for this valuable comment. Intuitively, the SwAV-style loss encourages **agreement within the same environment** and **diversity across environments**: two augmentations of the same sample are encouraged to produce consistent environment assignments (capturing shared causal context), while the orthogonality regularization keeps different environment embeddings distinct. This consistency-under-augmentation principle aligns closely with the causal goal of discovering latent environmental factors that remain stable across perturbations. We have added this clarification to the revision (Line 1079-1107).
>
> **W2.** Adopting the backdoor-adjusted approach requires learning additional components.  Possible experiments could involve using untrained or completely noisy environment variables to see how the model performs.
>
> **A2.** We thank the reviewer for the thoughtful comment. We agree that the backdoor-adjusted method will introduce additional cost, mainly arising from learning environment-aware components, but this overhead is small since the number of environments $K$ is limited (typically 2–8). As detailed in Appendix G.2, we already report comprehensive efficiency results (e.g., parameter size, GPU memory, and throughput) showing that CaTSG achieves near-linear scalability with moderate additional cost.
>
> Regarding the suggested experiment with uninformative or random environments, this is provided in our ablation study (Figure 5a, Section 5.5), where variants such as RandEnv, FrozenEnv, and w/o Env simulate noisy, untrained or no-environment settings. The results clearly demonstrate performance degradation in these cases, validating both the necessity and robustness of the environment-based backdoor adjustment. Furthermore, Figure 5d examines the effect of environment dimension and complexity ($K$ and $H$), showing that performance remains stable over a broad range with only mild variation, indicating robustness to environment size and representation capacity.
>
> **W3.** Causal methods typically target bias reduction. Usually, the trade-off is an increase in variance.
>
> **A3.** We thank the reviewer for highlighting this important point. For the variance analysis, we performed the standard deviation of uncertainty metrics across multiple random seeds (reported in Table 2). In response to the reviewer’s suggestion, we added a visualization of the results by dataset and metric in Figure 15 of our revision. Notably, CaTSG consistently exhibits the lowest or near-lowest variance across all settings, suggesting that our design (e.g., soft posterior inference, embedding regularization) effectively controls variance while achieving bias reduction.
>
> **W4.** Figure 3 a bit too dense for easy reading.
>
> **A4.** We thank the reviewer for the suggestion. We agree that improving Figure 3’s readability is valuable. Due to space and time constraints,, we were unable to fully redesign the figure, but we have strengthened key highlights (e.g., the $\epsilon$-combination equation) and darkened previously light text to improve clarity in our revision. We hope the updated figure is now easier to read.
>
> **[Questions]**
>
> **Q1.** Have you considered using a continuous or high-dimensional environment variable E?
>
> **A1.** Thank you for the valuable suggestion. While our environment bank consists of $K$ discrete embeddings, the model does not rely on hard assignments. Instead, each sample is assigned a soft probability distribution $w_k = p(e_k | x, c)$, and the final noise prediction aggregates all environments as a weighted mixture $\hat{\epsilon}= \sum_{k} w_k \varepsilon_{\theta}(x, t, \tilde{d}_{t})$ (Figure 3b, Section 4.3). This design effectively forms a continuous latent representation over the environment space, allowing the model to capture multiple weak or overlapping confounders. However, we acknowledge this remains a finite approximation and may not fully capture highly entangled or high-dimensional confounding. We have clarified this in the revised version (Appendix H.3). Thanks!

---

> ### Author Response · Authors · 2025-11-20
> **Author Response (Part 2)**
>
> **Q2.** Have you considered using additional data or information to aid in learning the confounders, instead of using SwAV?
>
> **A2.** Thank you for the suggestion. Our setting assumes the confounders $E$ are **unobserved** and complex, composed of many latent factors. Thus, supervised prediction of $E$ from $(X, C)$ is infeasible in this setting. The SwAV-style loss design enables the model to discover latent environments without labels, which better matches realistic scenarios where such annotations do not exist. Nevertheless, we agree with the reviewer that if certain confounders are observed, they can be used to supervise the environment bank learning. In this case, no major changes to the training or sampling pipeline are required, only the loss for the environment bank needs to incorporate the observed confounder features as supervision signals.
>
> **Q3.** Have you considered an IPW formulation for the problem?
>
> **A3.** Thank you for the insightful question. Indeed, backdoor adjustment and IPW are two ways to adjust for confounding and recover interventional distributions $p(x \mid do(c))$. Our method adopts the backdoor view: we introduce a latent environment variable ($E$) to capture unobserved confounding and incorporate the adjustment directly into the score function, allowing the generative process itself to perform the causal sampling without estimating explicit propensity scores. Nevertheless, we agree that an IPW formulation is conceptually possible: estimating $p(e \mid c)$, computing weights $1/p(e \mid c)$, and reweighting the objectives. Propensity estimation may become unstable and lead to identifiability issues, which are especially problematic in diffusion-based generation. For these reasons, we chose a backdoor adjustment that naturally fits our generative framework. Exploring explicit IPW-based weighting could be an interesting complementary direction. Thanks for raising this point.
>
> We sincerely appreciate the reviewer’s detailed comments and valuable suggestions. We hope that our responses and revisions address all raised questions, and we would be very happy to provide further clarification if needed.

---

### Official Review · Reviewer_AVVu · 2025-10-30

**Soundness:** 4
**Presentation:** 3
**Contribution:** 3
**Rating:** 8
**Confidence:** 3

**Summary:**

The paper formalizes causal time-series generation by extending conditional TSG beyond observation $P(X\mid C)$ to interventional $P(X\mid do(C))$ and counterfactual $P(X'\mid X,C,C')$ tasks. It introduces CaTSG, a diffusion framework that derives causal score functions via backdoor adjustment and the abduction–action–prediction procedure, enabling interventional and counterfactual sampling without ground-truth interventions.

CaTSG uses a learnable environment bank and an EnvInfer module to approximate latent confounders, trained with a swapped-prediction objective and an orthogonality regularizer; generation combines environment-aware denoiser outputs with posterior weights.  Empirically, across two synthetic and two real datasets, the method reports improved observational fidelity and mostly competitive interventional results against GAN and diffusion baselines, with counterfactual quality demonstrated on synthetic data.  The paper notes limitations from the assumed SCM and finite environment approximation.

**Strengths:**

* Thorough derivations and empirical study.
* Clear, well-structured presentation, helped by a consistent running example.
* Novel yet practical approach that avoids manual confounder hunting.
* Strong, natural integration with recent diffusion methods.
* Unified treatment of observational, interventional, and counterfactual generation.

**Weaknesses:**

* Most points below are already acknowledged by the authors; they’re noted here to contextualize impact rather than as surprises.
* Dependence on a fixed SCM (E->C, E->X, C->X) can limit validity under feedback, mediators, or time-varying confounding; conclusions may be sensitive to misspecification.
* Finite “environment bank” approximates latent causes; the choice of $K$ and representation quality could leave residual confounding or introduce bias.
* Real-data counterfactuals lack ground truth; evidence necessarily leans on synthetic settings or qualitative case studies, which constrains external validity.
* Robustness to guidance/inference hyperparameters (e.g., the guidance scale $\omega$) could be probed more systematically across seeds/datasets.
* Baseline coverage for interventional/counterfactual settings could be expanded where applicable to strengthen comparative claims.
* Computational overhead from diffusion plus environment inference is non-trivial; clearer wall-clock and memory reporting would help practitioners gauge trade-offs.

### Minor nitpicks

* On the running example, I praise the idea, but for this specific example, I think the direct edge C->X is not clear: is there really an influence of temperature on traffic, that is not a spurious correlation. I think an example where both effects are clear and we want to disentangle them would be more suited.
* There are several small LaTeX errors in the appendix, mainly missing subscripts (e.g l.791)
* I think the paper relies a bit too much on the appendix, some important figures are cited in the text, as if we were supposed to read the text with the figure beside it, but the figure is on the appendix (e.g. Fig 7)

**Questions:**

* If some confounders are observed, can CaTSG directly condition on them instead of learning an environment bank? What changes (if any) to training/sampling are needed?
* How easily can the framework adapt to other causal graphs (mediators, instrumental variables, partial feedback)? What concrete modifications to the guidance rules would that entail?
* What (if any) identifiability guarantees exist for the learned environments with finite $K$? Any recommended heuristics for choosing $K$ (and the guidance scale $\omega$) in practice?
* How do you diagnose and handle limited overlap/positivity in $(C,E)$ (e.g., rare environment–context pairs)? Would you recommend reweighting, trimming, or curriculum strategies?

---

> ### Author Response · Authors · 2025-11-20
> **Author Response (Part 1)**
>
> We sincerely thank the reviewer for the encouraging feedback. We are glad that the derivations, empirical study, clarity of presentation, and our unified treatment of observational, interventional, and counterfactual generation resonated with the reviewer. To respond, we have uploaded a revised version of manuscript, with all changes highlighted in red. Below, we address the reviewer’s concerns point by point.
>
> **[Weaknesses]**
>
> **W1.** Dependence on a fixed SCM can limit validity.
>
> **A1.** We thank the reviewer for raising this point. In our work, given that our primary objective is to establish a clear formulation for causal time series generation, we adopt a fixed and relatively simple SCM to serve as a conceptual abstraction. This structure captures the most common case where latent environments confound the observed variables and allows us to define counterfactual sampling clearly. We agree that this assumption may be restrictive in scenarios involving feedback or more complex dependencies, and extending CaTSG to support dynamic or adaptive causal graphs is an interesting and valuable direction for future work.
>
> **W2.** Finite environment bank and representation quality could leave residual confounding or introduce bias.
>
> **A2.** Thank you for pointing this out. We agree that a finite environment bank may under-represent latent confounders, especially when $K$ or the embedding dimension is small. In our work, we adopt this design as a tractable and interpretable approximation of the latent confounding space. To mitigate under-specification, our framework includes:
>
> - soft environment inference with probabilistic weighting $w_k$ to reduce the effect of hard mis-assignment;
> - diversity-promoting regularization (e.g., orthogonality loss) to encourage diversity; and
> - the swapped loss, which promotes environment consistency under augmentations and avoids collapse.
>
> These mechanisms help ensure that even with moderate $K$, the environment bank captures a sufficiently rich set of latent confounders for effective adjustment.
>
> Nevertheless, we agree with the reviewer that residual confounding may persist when the capacity is too limited. Exploring hierarchical or multi-factor environment models is a promising direction for future work. We appreciate the reviewer for pointing this out, and we have added this discussion in the revision (Appendix H.3). Thanks!
>
> **W3.** Real-data counterfactuals lack ground truth for evaluation.
>
> **A3.** We thank the reviewer for raising this important and well-recognized challenge. Indeed, counterfactuals on real-world datasets inherently lack ground-truth labels, which makes quantitative validation difficult.
>
> In respond, we reflected on how to better substantiate our claims in the absence of counterfactual labels. As an initial step, we added new measurements and introduced three complementary evaluation metrics: E-FTSD, Env-KL, and C-Consis. These metrics respectively assess (1) the alignment of generated and real distributions under marginalized environment variance, (2) the preservation of original environment characteristics, and (3) the content-level consistency after disentangling environmental factors via LDA projection. While these metrics cannot fully replace ground-truth evaluation, they offer concrete signals about the plausibility and stability of counterfactual generations. The details of these metrics’ formulations are added in Appendix G.4 of our revision.
>
> We summarize evaluations on Air Quality in table below, comparing CaTSG against its variants. In addition to the three new metrics, we also report MDD, KL, MMD, and J-FTSD on both the validation and test sets to assess counterfactual generation quality.
>
> |  | CaTSG | RandEnv | w/o SW | FrozenEnv |
> | --- | --- | --- | --- | --- |
> | MDD (val) | 0.064 ± 0.000 | 0.086 ± 0.000 | 0.085 ± 0.000 | 0.089 ± 0.000 |
> | KL  (val) | 0.290 ± 0.001 | 2.549 ± 0.002 | 2.542 ± 0.001 | 2.631 ± 0.002 |
> | MMD (val) | 0.058 ± 0.000 | 0.091 ± 0.001 | 0.086 ± 0.001 | 0.061 ± 0.000 |
> | J-FTSD  (val) | 0.806 ± 0.608 | 1.622 ± 0.552 | 2.868 ± 2.780 | 0.953 ± 0.785 |
> | E-FTSD  (val) | 1.532 ± 0.608 | 2.545 ± 0.552 | 3.237 ± 2.778 | 1.877 ± 0.786 |
> | MDD (test) | 0.064 ± 0.000 | 0.086 ± 0.000 | 0.084 ± 0.000 | 0.089 ± 0.000 |
> | KL  (test) | 0.264 ± 0.002 | 1.907 ± 0.002 | 1.893 ± 0.001 | 1.957 ± 0.001 |
> | MMD (test) | 0.070 ± 0.000 | 0.095 ± 0.001 | 0.090 ± 0.001 | 0.069 ± 0.000 |
> | J-FTSD  (test) | 2.905 ± 0.457 | 4.900 ± 0.313 | 4.757 ± 0.606 | 3.586 ± 0.799 |
> | E-FTSD  (test) | 3.631 ± 0.456 | 5.823 ± 0.314 | 5.128 ± 0.606 | 4.510 ± 0.799 |
> | Env-KL | 3.435 ± 0.103 | 4.175 ± 0.218 | 3.746 ± 0.118 | 4.594 ± 0.233 |
> | C-Consis | 1.754 ± 0.005 | 11.921 ± 0.154 | 30.188 ± 0.355 | 27.520 ± 0.213 |
>
> We see this as an encouraging first step toward more rigorous real-data counterfactual evaluation, and plan to further refine these measurements in future work. We added these discussions in the revision (Table 11, Appendix G.4).

---

> ### Author Response · Authors · 2025-11-20
> **Author Response (Part 2)**
>
> **W4.** Robustness to the guidance scale $\omega$ could be probed.
>
> **A4.** We thank the reviewer for raising the point. Indeed, guidance strength plays a key role in conditional diffusion models. In response to the reviewer’s suggestion, we conducted an additional sensitivity analysis across multiple $ \omega $ values (including {0.1, 0.2, 0.5, 1}), evaluated on Harmonic-VM and Air Quality, shown in the table below.
>
> | $ \omega $  | Harmonic-VM (MDD) | Harmonic-VM (KL) | Air Quality (MDD) | Air Quality (KL) |
> | --- | --- | --- | --- | --- |
> | 0.1 | 0.077 ± 0.001 | 0.011 ± 0.001 | 0.081 ± 0.000 | 0.316 ± 0.003 |
> | 0.2 | 0.098 ± 0.001 | 0.017 ± 0.000 | 0.069 ± 0.000 | 0.301 ± 0.001 |
> | 0.5 | 0.169 ± 0.001 | 0.069 ± 0.002 | 0.092 ± 0.000 | 0.409 ± 0.003 |
> | 1 | 0.243 ± 0.002 | 0.171 ± 0.001 | 0.101 ± 0.000 | 0.512 ± 0.002 |
>
> As shown in the table, small values of $\omega$ (e.g., 0.2) yield the best or near-best scores across datasets, while larger $\omega$ amplifies the conditional signal at the cost of diversity, leading to over-regularized generations. This trend justifies our default choice of $\omega = 0.2$. We also added this analysis in the revision (Table 12, Appendix G.6).
>
> **W5.** Expand baseline comparisons for the interventional and counterfactual settings.
>
> **A5.** We appreciate the reviewer’s suggestion. We would like to clarify that since the generative tasks for time series are, to the best of our knowledge, introduced in this work, existing TSG baselines are not designed to address these tasks. Specifically, prior models do not provide a formal mechanism to incorporate interventional operations, and they lack a generative interface that supports conditioning in a way suitable for counterfactual sampling. As such, existing methods cannot be directly applied to these tasks without significant modifications. One of the central contributions of our work is the proposal of a new model, CaTSG, that supports these tasks. We thank the reviewer for the opportunity to clarify this.
>
> **W6.** Clearer wall-clock and memory reporting would help practitioners gauge trade-offs.
>
> **A6.** Thank you for the comment. We would like to clarify that, though we only included partial results in Section 5.5 due to space constraints in the main text, Appendix G.2 (Tables 7–8, Fig. 14) reports all related experimental results, including parameter size, memory, runtime, and throughput, demonstrating moderate overhead and near-linear scalability.
>
> **W7.** Minor nitpicks.
>
> **A7.** We thank the reviewer for the insightful comment.
>
> - Regarding the direct edge $C \rightarrow X$, this edge in our running example indeed contains two components: (i) a genuine causal relationship (as correctly noted by the reviewer), and (ii) a spurious component driven by contextual factors such as the school calendar, which simultaneously affects $C$ and $X$ (as discussed in our example). The observed dependence is therefore a **mixture** of structural and confounded effects, and our model addresses this issue by introducing the environment $E$, which models the spurious calendar-driven variation, thereby isolating the stable causal component of the $C \rightarrow X$ relationship.  We have added this clarification to our revision (Appendix H.3). In addition, we also provide more diverse examples in Table 13 to make the intuition more general.
> - Regarding LaTeX errors about missing subscripts (Line.791), we have carefully re-checked the equations around that line (Eq. 14) and it appears to be correct on our side. It is possible that some rendering issues occurred in the PDF. If the reviewer could kindly indicate the specific symbol where the issue occurs, we would be very happy to correct it in the revision.
> - Regarding relying too much on the appendix (Figure 7), we sincerely thank the reviewer for pointing this out. There was an incorrect figure reference: the correct one is Figure 2 rather than Figure 7. We apologize for the confusion, and this has been fixed in the revision (Section 4.1). Thank you!
>
> **[Questions]**
>
> **Q1.** If some confounders are observed, can CaTSG directly condition on them instead of learning an environment bank? What changes to training/sampling are needed?
>
> **A1.** We appreciate the insightful question. When certain confounders are observed, CaTSG can indeed condition directly on them by concatenating these variables with $C$ as inputs during both training and sampling. Nevertheless, a more principled approach can be to use the observed confounders **to** **supervise or refine the environment bank**, rather than replacing it. The learned environment embeddings can then represent both observed and latent confounding factors in a unified latent space, enabling backdoor adjustment and counterfactual generation. In this case, no major changes to the training or sampling pipeline are required; only the loss for the environment bank needs to incorporate the observed confounder features as supervision signals.

---

> ### Author Response · Authors · 2025-11-20
> **Author Response (Part 3)**
>
> **Q2.** What modifications to the guidance rules would that entail for other causal graphs?
>
> **A2.** We thank the reviewer for the question. Our guidance rule is obtained by taking $\nabla_x \log$ of the identification formula for $p(x∣do(c))$ and then deriving it to get the final score function (Eqs.25–30). Thus, to handle other causal graphs (e.g., mediators/front-door, IV), we can simply replace the backdoor expression in Eq.25 with the appropriate identification formula and then do the derivation to get the new score function for guidance.
>
> **Q3.** What identifiability guarantees exist for the learned environments with finite $K$? Any recommended heuristics for choosing $K$ and $\omega$?
>
> **A3.** We appreciate the question. Regarding identifiability under finite $K$, please refer to our responses to **W2**. For discussions on the practical selection and robustness of $\omega$, please refer to our responses to **W4**. Thanks!
>
> **Q4.** How do you diagnose and handle limited overlap/positivity in $(C,E)$ ? Would you recommend reweighting, trimming, or curriculum strategies?
>
> **A4.** Thank you for your question. We address this in two parts:
>
> - Regarding diagnosis, Figure 6a visualizes the learned environment posterior across train/val/test splits, showing consistent usage of all $e_k$ without collapse or mode dropping. Figure 6b further confirms that the latent environment representations form well-separated, stable clusters across splits. These suggest sufficient marginal support for each $e_k$. While we do not explicitly enumerate joint $(C, E)$ combinations, the soft posterior $q(e \mid x, c)$ maintains high entropy under diverse contexts, providing indirect evidence of adequate overlap.
> - Regarding strategies, given the above, we did not observe a need to apply additional weighting or scheduling strategies. However, we agree that such techniques could be beneficial in highly imbalanced regimes. If there is an extreme lack of overlap, we can consider pruning the data appropriately to ensure that the basic balance is maintained. If we want to cover all combinations, it can be achieved by trying weighted learning combined with curriculum-based training to balance bias and variance. Thank you.
>
> We thank the reviewer once again for the thoughtful and constructive feedback. We hope our revisions and clarifications address all concerns, and we would be very happy to provide any additional explanations if needed.

---

### Author Response · Authors · 2025-11-20
**Global Response and Summary of Revisions**

We sincerely thank all reviewers for their constructive and insightful feedback. According to reviewers, our work was praised for its originality (R2), clear and well-structured presentation (R1, R3), and strong methodological contribution, including the environment-based conditioning and causal score-guided diffusion (R1–R3). Reviewers also highlighted the thorough experiments across four datasets and the model’s consistent strong performance (R1–R3).

A revised manuscript has been uploaded, with all **modifications highlighted in red**. Below we summarize the revisions made in response to reviewers’ comments, organized by theme:

- **Experiments:** added preliminary real-data counterfactual evaluation (Appendix G.4, Table 11), variance analysis (Appendix G.5, Figure 15), and guidance coefficient sensitivity experiments (Appendix G.6, Table 12).
- **Clarifications:** added intuition behind the SwAV-style loss (Line 1079–1107), detailed environment bank initialization (Line 1124–1125), expanded the discussion on choosing $K$ and $H$ (Line 1558–1560), and added discussion on genuine vs. spurious relationships and the finite environment bank (Appendix H.3).
- **Related Work:** expanded discussion on relevant literature [1–3]
- **Figures & Presentation:** improved readability of Figure 3 and corrected a figure reference error in Section 4.1.

We hope these revisions address the reviewers’ concerns, and we sincerely appreciate the valuable feedback.

*(Here, R1 = Reviewer AVVu, R2 = Reviewer wJr5, R3 = Reviewer zSS3)*

[1] Xia, Tian, et al. "Decoupled Classifier-Free Guidance for Counterfactual Diffusion Models." arXiv preprint arXiv:2506.14399 (2025).

[2] Komanduri, Aneesh, et al. "Causal diffusion autoencoders: Toward counterfactual generation via diffusion probabilistic models." arXiv preprint arXiv:2404.17735 (2024).

[3] Wu, Shenghao, et al. "Counterfactual generative models for time-varying treatments." KDD. 2024.

---

### Author Response · Authors · 2025-12-02

Dear Area Chair,

We would like to sincerely thank you for handling our submission and the reviewers for their constructive feedback. We have carefully revised the manuscript in response to all comments, and a concise summary of the main changes is provided in the **Global Response and Summary of Revisions** section.

Please kindly let us know if any further clarification would be helpful!

Best regards,

Authors

---

### Note · Program_Chairs · 2026-01-07
**Submission Desk Rejected by Program Chairs**

This submission has manipulated the ICLR template to have smaller margins and must be desk rejected.